# From Lazy to Rich: Exact Learning Dynamics in Deep Linear Networks

**Clémentine C. J. Dominé** [†,1]
clementine.domine.20@ucl.ac.uk

**Nicolas Anguita** [†,2]
nicolas.anguita20@imperial.ac.uk

**Alexandra M. Proca** [2]    **Lukas Braun** [3]    **Daniel Kunin** [4]    **Pedro A.M. Mediano** [‡,2,5]

**Andrew M. Saxe** [‡,1,6,7]

## Abstract

Biological and artificial neural networks create internal representations for complex tasks. In artificial networks, the ability to form task-specific representations is shaped by datasets, architectures, initialization strategies, and optimization algorithms. Previous studies show that different initializations lead to either a lazy regime, where representations stay static, or a rich regime, where they evolve dynamically. This work examines how initialization affects learning dynamics in deep linear networks, deriving exact solutions for $\lambda$-balanced initializations, which reflect the weight scaling across layers. These solutions explain how representations and the Neural Tangent Kernel evolve from rich to lazy regimes, with implications for continual, reversal, and transfer learning in neuroscience and practical applications.

## 1  Introduction

Biological and artificial neural networks learn internal representations that enable complex tasks such as categorization, reasoning, and decision-making. Both systems often develop similar representations from comparable stimuli, suggesting shared information processing mechanisms Yamins et al. (2014). Although not yet fully understood, this similarity has garnered significant interest from neuroscience, AI, and cognitive science Haxby et al. (2001); Laakso & Cottrell (2000); Morcos et al. (2018); Kornblith et al. (2019); Moschella et al. (2022). The success of neural models relies on their ability to form these representations and extract relevant features from data to build internal representations, a complex process that in machine learning is defined by two regimes: *lazy* and *rich* Saxe et al. (2014); Pennington et al. (2017); Chizat et al. (2019); Bahri et al. (2020). Despite significant advances, these learning regimes and their characterization are not yet fully understood and would benefit from clearer theoretical predictions, particularly regarding the influence of prior knowledge (initialization) on the learning regime. We discuss related works in the appendix A.

† First authors
1. Gatsby Computational Neuroscience Unit, University College London, London, United Kingdom
2. Department of Computing, Imperial College London, London, United Kingdom
3. Department of Experimental Psychology, University of Oxford, Oxford, United Kingdom
4. Institute for Computational and Mathematical Engineering, Stanford University, Stanford, USA
5. Division of Psychology and Language Sciences, University College London, London, United Kingdom
6. Sainsbury Wellcome Centre, University College London, London, United Kingdom
7. CIFAR Azrieli Global Scholar, CIFAR, Toronto, Canada
‡ Co-senior Author

**Our contributions.** (1) We derive exact solutions for the gradient flow in unequal-input-output two-layer deep linear networks, under a broad range of lambda-balanced initialization conditions (Section 2). (2) We model the full range of learning dynamics from *lazy* to *rich*, showing that this transition is influenced by a complex interaction of architecture, *relative scale*, and *absolute scale*, (Section 3). (3) We present applications relevant to both the neuroscience and machine learning field, providing exact solutions for continual learning dynamics, reversal learning dynamics, and transfer learning (Section 4).

## 2 Exact Learning Dynamics

**Setting** Consider a supervised learning task where input vectors $\mathbf{x}_n \in \mathbb{R}^{N_i}$, from a set of $P$ training pairs $\{(\mathbf{x}_n, \mathbf{y}_n)\}_{n=1}^P$, need to be mapped to their corresponding target output vectors $\mathbf{y}_n \in \mathbb{R}^{N_o}$. We learn this task with a two-layer linear network model that produces the output prediction $\hat{\mathbf{y}}_n = \mathbf{W}_2 \mathbf{W}_1 \mathbf{x}_n$, with weight matrices $\mathbf{W}_1 \in \mathbb{R}^{N_h \times N_i}$ and $\mathbf{W}_2 \in \mathbb{R}^{N_o \times N_h}$, where $N_h$ is the number of hidden units. The network's weights are optimized using full batch gradient descent with learning rate $\eta$ (or respectively time constant $\tau = \frac{1}{\eta}$) on the mean squared error loss $\mathcal{L}(\hat{\mathbf{y}}, \mathbf{y}) = \frac{1}{2} \langle ||\hat{\mathbf{y}} - \mathbf{y}||^2 \rangle$, where $\langle \cdot \rangle$ denotes the average over the dataset. The dynamics are completely determined by the input covariance and input-output correlation matrices of the dataset, defined as $\tilde{\mathbf{\Sigma}}^{xx} = \frac{1}{P} \sum_{n=1}^P \mathbf{x}_n \mathbf{x}_n^T \in \mathbb{R}^{N_i \times N_i}$ and $\tilde{\mathbf{\Sigma}}^{yx} = \frac{1}{P} \sum_{n=1}^P \mathbf{y}_n \mathbf{x}_n^T \in \mathbb{R}^{N_o \times N_i}$, and the initialization $\mathbf{W}_2(0), \mathbf{W}_1(0)$. Our objective is to describe the entire dynamics of the network's output and internal representations based on this initialization and the task statistics. We consider an approach first introduced in the foundational work of Fukumizu Fukumizu (1998) and extended in recent work by Braun et al. (2022), which rather than consider the dynamics of the parameters directly, we consider the dynamics of a matrix of the important statistics. In particular, defining $\mathbf{Q} = \begin{bmatrix} \mathbf{W}_1 & \mathbf{W}_2^T \end{bmatrix}^T \in \mathbb{R}^{(N_i + N_o) \times N_h}$, we consider the $(N_i + N_o) \times (N_i + N_o)$ matrix $\mathbf{QQ}^T(t) = \begin{bmatrix} \mathbf{W}_1^T \mathbf{W}_1(t) & \mathbf{W}_1^T \mathbf{W}_2^T(t) \\ \mathbf{W}_2 \mathbf{W}_1(t) & \mathbf{W}_2 \mathbf{W}_2^T(t) \end{bmatrix}$, which is divided into four quadrants with interpretable meanings. The approach tracks several key statistics collected in the matrix. The off-diagonal blocks contain the network function $\hat{\mathbf{Y}}(t) = \mathbf{W}_2 \mathbf{W}_1(t)\mathbf{X}$, which can be used to evaluate the dynamics of the loss as shown in Fig. 1. The on-diagonal blocks capture the correlation structure of the weight matrices, allowing for the calculation of the temporal evolution of the network's internal representations. This includes the representational similarity matrices (RSM) of the neural representations within the hidden layer, as first defined by Braun et al. (2022),$\text{RSM}_I = \mathbf{X}^T \mathbf{W}_1^T \mathbf{W}_1(t)\mathbf{X}$, $\text{RSM}_O = \mathbf{Y}^T (\mathbf{W}_2 \mathbf{W}_2^T(t))^+ \mathbf{Y}$, where $+$ denotes the pseudoinverse; and the network's finite-width NTK Jacot et al. (2018); Lee et al. (2019); Arora et al. (2019b) $\text{NTK} = \mathbf{I}_{N_o} \otimes \mathbf{X}^T \mathbf{W}_1^T \mathbf{W}_1(t)\mathbf{X} + \mathbf{W}_2 \mathbf{W}_2^T(t) \otimes \mathbf{X}^T \mathbf{X}$, where $\mathbf{I}$ is the identity matrix and $\otimes$ is the Kronecker product. Hence, the dynamics of $\mathbf{QQ}^T$ describes the important aspects of network behaviour.

We extend previous solutions Fukumizu (1998); Braun et al. (2022); Kunin et al. (2024) and derive exact solutions for the dynamics of $\mathbf{QQ}^T$ in unequal-input-output under a broad range of lambda-balanced initialization conditions. See Appendix B.2 for a further discussion of the assumptions and their relation to previous works. The proof of the Theorem and lemma leading to the theorem is in Appendix C. With this solution we can calculate the exact temporal dynamics of the loss, network function, RSMs and NTK (Fig. 1A, C) over a range of lambda-balanced initializations. **Implementation and simulation.** Simulation details are in Appendix F.7.

## 3 Rich and Lazy Learning

In this section we aim to gain a deeper understanding of the transition between the *rich* and *lazy* regimes by examining the dynamics as a function of lambda – the *relative scale* - as it varies between positive and negative infinity.

**Dynamics of the singular values.** Here we examine a *lambda-balanced* linear network initialized with *task-aligned* weights. Previous research Saxe et al. (2019a) has demonstrated that initial weights that are aligned with the task remain aligned throughout training, restricting the learning dynamics to the singular values of the network. As shown in Fig.4 B, as $\lambda$ approaches zero, the dynamics resemble sigmoidal learning curves that traverse between saddle points, characteristic of

the *rich* regime Braun et al. (2022). In this regime the network learns the most salient features first, which can be beneficial for generalization Lampinen & Ganguli (2018). Conversely, as shown in Fig.4 A and C, as the magnitude of $\lambda$ increases, the dynamics become exponential, characteristic of the *lazy* regime. In this regime, all features are treated equally and the network's dynamics resemble that of a shallow network. Overall, our results highlight the critical influence of both the *absolute scale* and the *relative scale* $\lambda$ has in shaping the learning dynamics, from sigmoidal to exponential, steering the network between the *rich* and *lazy* regimes. The proof can be found in Appendix D.1.

**The dynamics of the representations.** We examine how the representations of the parameters $W_1$ and $W_2$ evolve during training. With lambda-balanced initializations, the structure persists throughout training, allowing us to recover the dynamics up to a time-dependent orthogonal transformation. The singular values $S_\lambda$ of the weights are adjusted based on $\lambda$, splitting the representation into two parts (Theorem D.2). Using $\mathbf{Q}\mathbf{Q}^T(t)$, we capture the temporal dynamics of hidden layer activations and analyze whether the network adopts a *rich* or *lazy* representation, depending on $\lambda$. Upon convergence, the internal representation satisfies $\mathbf{W}_1^T\mathbf{W}_1 = \tilde{\mathbf{V}}\tilde{\mathbf{S}}_1^2\tilde{\mathbf{V}}^T$ and $\mathbf{W}_2\mathbf{W}_2^T = \tilde{\mathbf{U}}\tilde{\mathbf{S}}_2^2\tilde{\mathbf{U}}^T$, with detailed proof in Theorem D.3. For a hierarchical semantic task Saxe et al. (2014); Braun et al. (2022), the representational similarity of inputs ($\tilde{\mathbf{V}}\tilde{\mathbf{S}}\tilde{\mathbf{V}}^T$) and targets ($\tilde{\mathbf{U}}\tilde{\mathbf{S}}\tilde{\mathbf{U}}^T$) aligns with the task structure. When training a two-layer network with relative scale $\lambda = 0$, the representational similarity matrices match the task upon convergence (Theorem D.3). As $\lambda$ approaches positive or negative infinity, the network transitions to the *lazy* regime, adopting task-agnostic representations (Theorem D.4, Fig. 2). The NTK be-

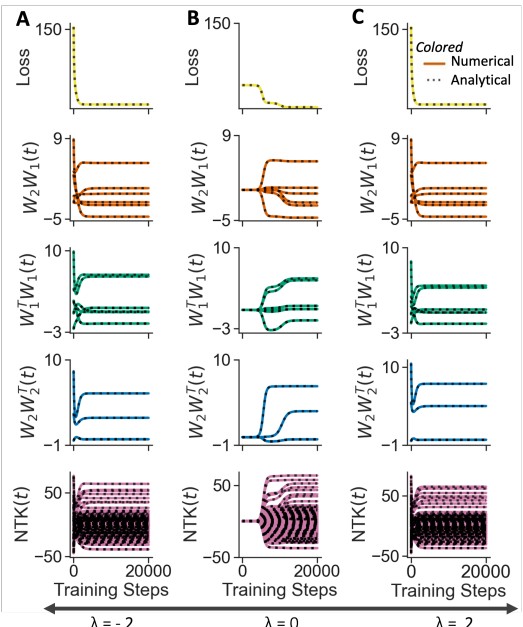

Figure 1: **A** The temporal dynamics of the numerical simulation of the loss, network function, correlation of input and output weights, and the NTK (row 1-5 respectively) are exactly matched by the analytical solution for $\lambda = -2$. **B** $\lambda = 0.001$ Large initial weight values. **C** $\lambda = 2$ initial weight values.

comes static and identity-like, while downscaled representations remain structured. This property can enhance generalization during fine-tuning, as shown in Section 4. In contrast, large Gaussian initializations result in *lazy* learning, lacking structural representation. We propose a new *semi-structured lazy* regime, where initialization determines whether task-specific features reside in input or output layers.

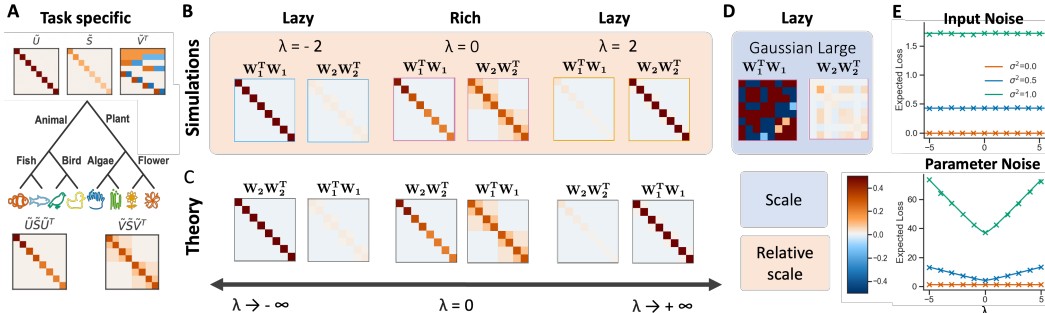

Figure 2: **A** A semantic learning task with the SVD of the input-output correlation matrix of the task. (top) $U$ and $V$ represent the singular vectors, and $S$ contains the singular values. (bottom) The respective RSMs for the input and for the output task. **B** Simulation results and **C** Theoretical input and output representation matrices after training, showing convergence when initialized with varying lambda values, according to the initialization scheme described in F.7. **D** Final RSMs matrices after training converged when initialised from random large weights. **E** After convergence, the network's sensitivity to input noise (top panel) is invariant to $\lambda$, but the sensitivity to parameter noise increases as $\lambda$ becomes smaller (or larger) than zero.

**Representation robustness and sensitivity to noise.** In appendix D.3 we examine the relationship between the learning regime and the robustness of the learned representations to added noise in the inputs and parameters. In practice, parameter noise could be interpreted as the noise occurring within the neurons of a biological network. We find that a rich solution may enable a more robust representation in such systems.

**The impact of the architecture.** Thus far, we have found that the magnitude of the *relative scale* parameter $\lambda$ determines the extent or rich and lazy learning. Here, we explore how a network's learning regime is influenced by the interaction of its architecture and the sign of the *relative scale*. We consider three types of network architectures, depicted in Fig. 3A: *funnel networks*, which narrow from input to output ($N_i > N_h = N_o$); *inverted-funnel networks*, which expand from input to output ($N_i = N_h < N_o$); and *square networks*, where input and output dimensions are equal ($N_i = N_h = N_o$). Our solution, $\mathbf{QQ}^T$, captures the NTK dynamics across these different network architectures. To examine the NTK's evolution under varying $\lambda$ initializations, we compute the kernel distance from initialization, as defined in Fort et al. (2020). As shown in Fig. 3B, we observe that funnel networks consistently enter the *lazy* regime as $\lambda \to \infty$, while inverted-funnel networks do so as $\lambda \to -\infty$. The NTK remains static during the initial phase, rigorously confirming the rank argument first introduced by Kunin et al. (2024) for the multi-output setting. In the opposite limits of $\lambda$, these networks transition from a *lazy* regime to a *rich* regime. During this second alignment phase, the NTK matrix undergoes changes, indicating an initial *lazy* phase followed by a *delayed rich* phase. We further investigate and quantify this *delayed rich* regime, showing the NTK movement over training in Fig. 3C. This behavior is also quantified in Theorem D.6, which describes the rate of learning in this network. For square networks with equal input and output dimensions, this behavior is discussed in Section 3. Across all architectures, as $\lambda \to 0$, the networks consistently transition into the *rich* regime. Altogether, we further characterize the *delayed rich* regime in wide networks.

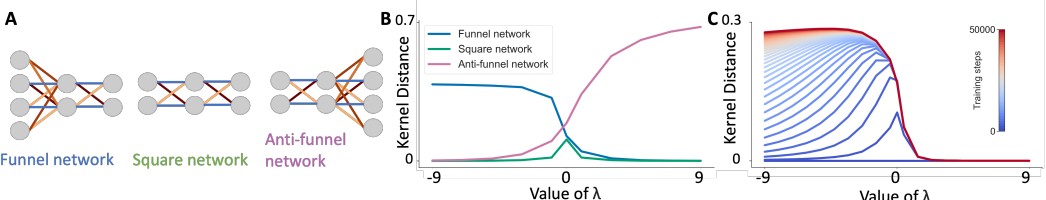

Figure 3: **A.** Schematic representations of the network architectures considered, from left to right: funnel network, square network, and inverted-funnel network. **B.** The plot shows the NTK kernel distance from initialization, as defined in Fort et al. (2020) across the three architecture depicted schematically. **C.** The NTK kernel distance away from initialization over training time.

# 4 Application

**Continual learning.** In line with the framework presented by Braun et al. (2022), our approach describes the exact solutions of the networks dynamics trained across a sequence of tasks. As detailed in Appendix E.1, we demonstrate that, regardless of the chosen value of lambda, training on subsequent tasks can result in the overwriting of previously acquired knowledge, leading to catastrophic forgetting McCloskey & Cohen (1989); Ratcliff (1990); French (1999).

**Reversal learning.** As demonstrated in Braun et al. (2022), reversal learning theoretically does not succeed in deep linear networks as the initalization aligns with the separatrix of a saddle point. While simulations show that the learning dynamics can escape the saddle point due to numerical imprecision, the process is catastrophically slowed in its vicinity. However, when $\lambda$ is non-zero, reversal learning dynamics consistently succeed, as they avoid passing through the saddle point due to the initialization scheme. This is both theoretically proven and numerically illustrated in Appendix E.2. We also present a spectrum of reversal learning behaviors controlled by the *relative scale* $\lambda$, ranging from *rich* to *lazy* learning regimes. This spectrum has the potential to explain the diverse dynamics observed in animal behavior, offering insights into the learning regimes relevant to various neuroscience experiments.

**Transfer learning.** We consider how different $\lambda$ initializations influence generalization to a new feature after being trained on an initial task. We observe in Appendix figure 7 that the task specific structure of the data is effectively transferred to the new feature when the representation is task-specific and $\lambda$ is zero. Conversely, when the output feature representation is *lazy*, meaning the hidden representation lacks adaptation, no task specific generalization is observed. Strikingly, when $\lambda$ is positive, the task specific structure in the input weights remains small but structured, while the output weights exhibit a *lazy* representation and the network generalizes to the task specific features. This suggests that the *lazy* regime structure can be beneficial for transfer learning.

# 5 Discussion

We derive exact solutions to the learning dynamics within a tractable model class: deep linear networks. We examine the transition between the *rich* and *lazy* regimes by analyzing the dynamics as a function of $\lambda$—the *relative scale*—across its full range from positive to negative infinity. Our analysis demonstrates that the *relative scale*, $\lambda$, plays a crucial role in managing the transition between *rich* and *lazy* regimes.

### Acknowledgments

This research was funded in whole, or in part, by the Wellcome Trust [216386/Z/19/Z]. For the purpose of Open Access, the author has applied a CC BY public copyright licence to any Author Accepted Manuscript version arising from this submission. C.D. and A.S. were supported by the Gatsby Charitable Foundation (GAT3755). Further, A.S. was supported by the Sainsbury Wellcome Centre Core Grant (219627/Z/19/Z) and A.S. is a CIFAR Azrieli Global Scholar in the Learning in Machines & Brains program. A.M.P. was supported by the Imperial College London President's PhD Scholarship. L.B. was supported by the Woodward Scholarship awarded by Wadham College, Oxford. and the Medical Research Council [MR/N013468/1]. D.K. thanks the Open Philanthropy AI Fellowship for support. This research was funded in whole, or in part, by the Wellcome Trust [216386/Z/19/Z].

### Author Contributions

Clementine and Nicolas led the work on the equal input-output solution to the exact dynamics, with support from Daniel. Clementine lead the extention to this work to the unequal input-output case from priminilary work from Nicolas. Then she examining the impact of the architecture. Clementine and Nicolas also spearheaded the development of the minimal model for *rich* and *lazy* learning, emphasizing exact solutions and representation convergence. Daniel, Nicolas, and Clementine collaborated on analyzing the dynamics of singular values, while Lukas led the investigation into representation robustness and sensitivity to noise. Clementine and Nicolas primarily conducted the analysis of the continual learning framework, while Alexandra and Clementine focused on transfer

learning in the *rich* and *lazy* settings. Clementine took the lead on the reversal learning analysis. She was primarily responsible for leading the manuscript writing, with assistance from Daniel. Initial work on this project was carried during Nicolas masters supervised by Clementine and Pedro. All authors contributed to writing the appendix and refining the final manuscript.

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

# A   Related Work

**Lazy regime.** Extensive research has identified a fundamental phenomenon in overparameterized neural networks: during training, these networks frequently remain near their linearized form, undergoing minimal changes in the parameter space Chizat et al. (2019). Consequently, they adopt learning dynamics akin to kernel regression, characterized by the Neural Tangent Kernel (NTK) matrix and exhibiting exponential learning behavior Du et al. (2018); Jacot et al. (2018); Du et al. (2019); Allen-Zhu et al. (2019a,b); Zou et al. (2020). This behavior, known as the *lazy* or kernel regime, typically occurs in infinitely wide architectures and can be triggered by large variance initialization at the start of training Jacot et al. (2018); Chizat et al. (2019). While the *lazy* regime offers valuable insights into how networks converge to a global minimum, it does not fully account for the generalization capabilities of neural networks trained with standard initializations. It is, therefore, widely believed that another regime, driven by small or vanishing initializations, underpins some of the successes of neural networks.

**Rich regime.** In contrast, the *rich* feature-learning regime is characterized by a NTK that evolves throughout training, accompanied by non-convex dynamics that navigate saddle points Baldi & Hornik (1989); Saxe et al. (2014, 2019b); Jacot et al. (2021). This regime features sigmoidal learning curves and simplicity biases, such as low-rankness Li et al. (2020) or sparsity Woodworth et al. (2020). Numerous studies have shown that the *absolute scale* of initialization drives the *rich* regime, which typically emerges at small initialization scales Chizat et al. (2019); Geiger et al. (2020). However, it's also been shown that even at small initialization scales, differences in weight magnitudes between layers can induce the *lazy* learning regime Azulay et al. (2021); Kunin et al. (2024). This highlights the significance of both *absolute scale* (initialization variance) and *relative scale* (difference in weight magnitude between layers) in generating diverse learning dynamics. Beyond *absolute scale* and *relative scale*, additional aspects of initialization can profoundly affect feature learning, including the effective rank of the weight matrices Liu et al. (2023), layer-specific initialization variances Yang & Hu (2020); Luo et al. (2021); Yang et al. (2022), and the use of large learning rates Lewkowycz et al. (2020); Ba et al. (2022); Zhu et al. (2023); Cui et al. (2024). These findings illustrate the effect of initialization on inducing complex learning behavior through the resulting dynamics. Here we develop a solvable model which captures these diverse phenomena.

**Rich and lazy regimes in the brain.** The distinction between *rich* and *lazy* learning may also hold implications for neuroscience, where neural representations have been argued to have task-specific or task-agnostic characteristics in different settings Farrell et al. (2023a); Ostojic & Fusi (2024); Tye et al. (2024). The *lazy* regime can be linked to the non-linear mixed selectivity of neurons, where task variables are represented in a high-dimensional space which mixes various potentially relevant variables Raposo et al. (2014); Tang et al. (2019); Rigotti et al. (2013); Bernardi et al. (2020). Conversely, the *rich* regime aligns with linear mixed selectivity Tye et al. (2024) and the manifold learning regime, where the brain encodes tasks on a structured, low-dimensional, task-specific manifold, as observed in grid cells within the entorhinal cortex Chaudhuri et al. (2019); Bernardi et al. (2020); Flesch et al. (2022).

**Linear networks.** Our work builds upon a rich body of research on deep linear networks, which, despite their simplicity, have proven to be valuable models for understanding more complex neural networks Baldi & Hornik (1989); Fukumizu (1998); Saxe et al. (2014). Previous research has extensively analyzed convergence Arora et al. (2018a); Du & Hu (2019), generalization properties Lampinen & Ganguli (2018); Poggio et al. (2018); Huh (2020), and the implicit bias of gradient descent Arora et al. (2019a); Woodworth et al. (2020); Chizat & Bach (2020); Kunin et al. (2022) in linear networks. These studies have also revealed that deep linear networks have intricate fixed point structures and nonlinear learning dynamics in parameter and function space, reminiscent of phenomena observed in nonlinear networks Arora et al. (2018b); Lampinen & Ganguli (2018). Seminal work by Saxe et al. (2014) laid the groundwork by providing exact solutions to gradient flow dynamics under task-aligned initializations, demonstrating that the largest singular values are learned first during training. This analysis has been extended to deep linear networks Arora et al. (2018b, 2019a); Ziyin et al. (2022) with more flexible initialization schemes Gidel et al. (2019); Tarmoun et al. (2021); Gissin et al. (2019). This work directly builds on the matrix Riccati formulation proposed by Fukumizu (1998); Braun et al. (2022) which extends these solutions to wide networks. We extend and refine these results to obtain the dynamics for lambda-balanced initialization dynamics of networks to more clearly demonstrate the impact of initialization on *rich* and *lazy* learning regimes also developed in Tu et al. (2024) for a set of orthogonal initalizations. Our work

extends previous analysis Xu & Ziyin (2024); Kunin et al. (2024) of these regime to wide networks. Previous studies leveraged these solutions primarily to characterize convergence rates; however, our work goes beyond this by providing a comprehensive characterization of the complete dynamics of the system Tarmoun et al. (2021).

**Infinite-width networks.** Recent advances in understanding the *rich* regime have largely stemmed from examining how the initialization variance and layer-wise learning rates must scale in the infinite-width limit to maintain consistent behavior in activations, gradients, and outputs. Several studies have employed statistical mechanics tools to derive analytical solutions for the *rich* population dynamics of two-layer nonlinear neural networks initialized using the *mean field* parameterization Mei et al. (2018); Rotskoff & Vanden-Eijnden (2018); Chizat & Bach (2018); Sirignano & Spiliopoulos (2020); Rotskoff & Vanden-Eijnden (2022); Sirignano & Spiliopoulos (2020). Other methods for analyzing deep network dynamics include the NTK limit, where the network effectively performs kernel regression without feature learning Jacot et al. (2018); Lee et al. (2019); Arora et al. (2019b). Our solution allows us to the study the evolution of the NTK and the influence of *absolute scale* and *relative scale* on the transition between *lazy* and *rich* learning in finite width networks Jacot et al. (2021); Xu & Ziyin (2024); Kunin et al. (2024); Chizat et al. (2019). Furthermore, these approaches typically require numerical integration or operate within a limited learning regime, and are unable to describe the learning dynamics of hidden representations. Instead, our work focuses on the impact of initialization on representation learning dynamics and derives explicit analytical solutions within tractable models.

# B    Preliminaries

## B.1    Appendix: Balanced Condition

**Definition B.1** (Definition of $\lambda$-*balanced* property (Saxe et al. (2013), Marcotte et al. (2023)))**.** The weights $W_1, W_2$ are $\lambda$-*balanced* if and only if there exists a **Balanced Coefficient** $\lambda \in \mathbb{R}$ such that:

$$B(W_1, W_2) = W_2^T W_2 - W_1 W_1^T = \lambda I \tag{1}$$

where $B$ is called the **Balanced Computation**.
For $\lambda = 0$ we have **Zero-Balanced** given as **A1** (). $\mathbf{W}_1(0)\mathbf{W}_1(0)^T = \mathbf{W}_2(0)^T\mathbf{W}_2(0)$.

**Theorem B.2.** *Balanced Condition Persists Through Training*

*Suppose at initialization*

$$W_2(0)^T W_2(0) - W_1(0) W_1(0)^T = \lambda I \tag{2}$$

*Then for all $t \geq 0$*

$$W_2(t)^T W_2(t) - W_1(t) W_1(t)^T = \lambda I \tag{3}$$

*Proof.* Consider:

$$
\begin{aligned}
\tau \frac{d}{dt} \left[ W_2(t) W_2(t)^T - W_1(t) W_1(t)^T \right] &= \left( \tau \frac{d}{dt} W_2(t) \right)^T W_2(t) + W_2(t)^T \left( \tau \frac{d}{dt} W_2(t) \right) \\
&\quad - \left( \tau \frac{d}{dt} W_1(t) \right) W_1(t)^T - W_1(t) \left( \tau \frac{d}{dt} W_1(t) \right)^T \\
&= W_1(t) \left( \tilde{\Sigma}^{yx} - W_2(t) W_1(t) \tilde{\Sigma}^{xx} \right)^T W_2(t) \\
&\quad + W_2(t)^T \left( \tilde{\Sigma}^{yx} - W_2(t) W_1(t) \tilde{\Sigma}^{xx} \right) W_1(t) \\
&\quad - W_2(t)^T \left( \tilde{\Sigma}^{yx} - W_2(t) W_1(t) \tilde{\Sigma}^{xx} \right) W_1(t) \\
&\quad - W_1(t) \left( \tilde{\Sigma}^{yx} - W_2(t) W_1(t) \tilde{\Sigma}^{xx} \right) W_2(t) \\
&= 0
\end{aligned}
$$

Note that $\boldsymbol{W_2}(t)^T\boldsymbol{W_2}(t) - \boldsymbol{W_1}(t)\boldsymbol{W_1}(t)^T$ is conserved for any initial value $\lambda$.  □

## B.2   Discussion Assumptions

- **A2** (*Whitened input*). The input data is whitened, that is $\tilde{\boldsymbol{\Sigma}}^{xx} = \mathbf{I}$.

- **A3** (*Lambda-balanced*). The network's weight matrices are lambda-balanced at the beginning of training, that is $\mathbf{W}_2(0)^T\mathbf{W}_2(0) - \mathbf{W}_1(0)\mathbf{W}_1(0)^T = \lambda\mathbf{I}$. If this condition holds at initialization, it will persist throughout training Saxe et al. (2014); Arora et al. (2018a). For completeness, we prove this in Appendix B.

- **A4** (*Dimensions*). The hidden dimension of the network is defined as $N_h = \min(N_i, N_o)$, ensuring the network is neither bottlenecked ($N_h < \min(N_i, N_o)$) nor overparameterized ($N_h > \min(N_i, N_o)$).

- **A5** (*Full-rank*). The input-output correlation of the task and the initial state of the network function have full rank, that is $\mathrm{rank}(\tilde{\boldsymbol{\Sigma}}^{xy}) = \mathrm{rank}(\mathbf{W}_2(0)\mathbf{W}_1(0)) = \min(N_i, N_o)$.

**Whittened Inputs.**    Although the whitened input assumption is quite strong, it is commonly used in analytical work to obtain exact solutions, and much of the existing literature relies on these solutions Fukumizu (1998); Braun et al. (2022); Kunin et al. (2024) . Kunin et al. (2024) goes further by exploring the implicit bias of the trajectory without relying on exact solutions. When $X^\intercal X$ is low-rank, they can only predict the trajectories in the limit as $\lambda \to \pm\infty$. If the interpolating manifold is one-dimensional, the solution can be solved exactly in terms of $\lambda$ (black dots).

**Dimension.**    Fukumizu assumed equal input and output dimensions $N_i = N_o$, but allowed for a bottleneck in the hidden dimension of the network $N_h \leq N_i = N_o$. The work by Braun et al. (2022) extended Fukumizu (1998) solutions to cases with unequal input and output dimensions $N_i \neq N_o$, but to so did not allow a bottleneck $N_h = \min\{N_i, N_o\}$ and added an assumption on the invertibility of a statistic of the singular vector overlap between the model and the input-output statistics. In our work we allow for unequal input and output $N_i \neq N_o$ and do not introduce an additional invertibility assumption.

**Balancedness.**    The main distinction between our work and prior works is that both Fukumizu (1998) and Braun et al. (2022) assumed zero-balanced $\mathbf{W}_1(0)\mathbf{W}_1(0)^T = \mathbf{W}_2(0)^T\mathbf{W}_2(0)$, while we relax this assumption to $\lambda$-balanced. The zero-balanced condition restricts the networks to a *rich* setting. We develop solutions to explore the continuum between the *rich* and the *lazy* regime. While some works, such as Tarmoun et al. (2021), have considered removing this constraint, their solutions remain in an unstable and mixed form. Our work, in its form enable the understanding of different learning regimes by exploring initialization properties beyond just *absolute scale* and demonstrate that this transition can be accessed and controlled by adjusting a key parameter: the *relative scale*. Other studies, such as Kunin et al. (2024) and Xu & Zheng (2024), have similarly relaxed the balancedness assumption but were limited to single-output neuron settings.

## C   Appendix: Exact learning dynamics with prior knowledge

### C.1   Appendix: Fukumizu Approach

**Lemma C.1.** *We introduce the variables*

$$\mathbf{Q} = \begin{bmatrix} \mathbf{W}_1^T \\ \mathbf{W}_2 \end{bmatrix} \quad and \quad \mathbf{Q}\mathbf{Q}^T = \begin{bmatrix} \mathbf{W}_1^T\mathbf{W}_1 & \mathbf{W}_1^T\mathbf{W}_2^T \\ \mathbf{W}_2\mathbf{W}_1 & \mathbf{W}_2\mathbf{W}_2^T \end{bmatrix}. \tag{4}$$

*Defining*

$$\mathbf{F} = \begin{bmatrix} -\frac{\lambda}{2}I & (\tilde{\Sigma}^{yx})^T \\ \tilde{\Sigma}^{yx} & \frac{\lambda}{2}I \end{bmatrix}, \tag{5}$$

*the gradient flow dynamics of $\mathbf{Q}\mathbf{Q}^T(t)$ can be written as a differential matrix Riccati equation*

$$\tau\frac{d}{dt}(\mathbf{Q}\mathbf{Q}^T) = \mathbf{F}\mathbf{Q}\mathbf{Q}^T + \mathbf{Q}\mathbf{Q}^T\mathbf{F} - (\mathbf{Q}\mathbf{Q}^T)^2. \tag{6}$$

*Proof.* We introduce the variables

$$\mathbf{Q} = \begin{bmatrix} \mathbf{W}_1^T \\ \mathbf{W}_2 \end{bmatrix} \quad \text{and} \quad \mathbf{Q}\mathbf{Q}^T = \begin{bmatrix} \mathbf{W}_1^T\mathbf{W}_1 & \mathbf{W}_1^T\mathbf{W}_2^T \\ \mathbf{W}_2\mathbf{W}_1 & \mathbf{W}_2\mathbf{W}_2^T \end{bmatrix}. \tag{7}$$

We compute the time derivative

$$\tau\frac{d}{dt}(\mathbf{Q}\mathbf{Q}^T) = \tau \begin{bmatrix} \frac{d\mathbf{W}_1^T}{dt}\mathbf{W}_1 + \mathbf{W}_1^T\frac{d\mathbf{W}_1}{dt} & \frac{d\mathbf{W}_1^T}{dt}\mathbf{W}_2 + \mathbf{W}_1^T\frac{d\mathbf{W}_2}{dt} \\ \frac{d\mathbf{W}_2}{dt}\mathbf{W}_1 + \mathbf{W}_2\frac{d\mathbf{W}_1}{dt} & \frac{d\mathbf{W}_2^T}{dt}\mathbf{W}_2 + \mathbf{W}_2^T\frac{d\mathbf{W}_2}{dt} \end{bmatrix}. \tag{8}$$

Using equations 18 and 19, we compute the four quadrants separately giving

$$\tau\left(\frac{d\mathbf{W}_1^T}{dt}\mathbf{W}_1 + \mathbf{W}_1^T\frac{d\mathbf{W}_1}{dt}\right) = \tag{9}$$

$$= (\Sigma^{yx} - \mathbf{W}_2\mathbf{W}_1)^T\mathbf{W}_2\mathbf{W}_1 + \mathbf{W}_1^T\mathbf{W}_2^T(\Sigma^{yx} - \mathbf{W}_2\mathbf{W}_1) \tag{10}$$

$$= (\Sigma^{yx})^T\mathbf{W}_2\mathbf{W}_1 + \mathbf{W}_1^T\mathbf{W}_2^T\Sigma^{yx} - \mathbf{W}_1^T\mathbf{W}_2^T\mathbf{W}_2\mathbf{W}_1 - (\mathbf{W}_2\mathbf{W}_1)^T\mathbf{W}_2\mathbf{W}_1 \tag{11}$$

$$= (\Sigma^{yx})^T\mathbf{W}_2\mathbf{W}_1 + \mathbf{W}_1^T\mathbf{W}_2^T\Sigma^{yx} - \mathbf{W}_1^T\mathbf{W}_2^T\mathbf{W}_2\mathbf{W}_1 - \mathbf{W}_1^T\mathbf{W}_1\mathbf{W}_1^T\mathbf{W}_1 - \lambda\mathbf{W}_1^T\mathbf{W}_1, \tag{12}$$

$$\tau\left(\frac{d\mathbf{W}_1^T}{dt}\mathbf{W}_2^T + \mathbf{W}_1^T\frac{d\mathbf{W}_2^T}{dt}\right) = \tag{13}$$

$$= (\Sigma^{yx} - \mathbf{W}_2\mathbf{W}_1)^T\mathbf{W}_2\mathbf{W}_2^T + \mathbf{W}_1^T\mathbf{W}_1(\Sigma^{yx} - \mathbf{W}_2\mathbf{W}_1)^T \tag{14}$$

$$= (\Sigma^{yx})^T\mathbf{W}_2\mathbf{W}_2^T + \mathbf{W}_1^T\mathbf{W}_1(\Sigma^{yx})^T - \mathbf{W}_1^T\mathbf{W}_1\mathbf{W}_1^T\mathbf{W}_2^T - \mathbf{W}_1^T\mathbf{W}_2^T\mathbf{W}_2\mathbf{W}_2^T, \tag{15}$$

$$\tau\left(\frac{d\mathbf{W}_2}{dt}\mathbf{W}_1 + \mathbf{W}_2\frac{d\mathbf{W}_1}{dt}\right) = \tag{16}$$

$$= (\Sigma^{yx} - \mathbf{W}_2\mathbf{W}_1)\mathbf{W}_1^T\mathbf{W}_1 + \mathbf{W}_2\mathbf{W}_2^T(\Sigma^{yx} - \mathbf{W}_2\mathbf{W}_1) \tag{17}$$

$$= \Sigma^{yx}\mathbf{W}_1^T\mathbf{W}_1 + \mathbf{W}_2\mathbf{W}_2^T\Sigma^{yx} - \mathbf{W}_2\mathbf{W}_2^T\mathbf{W}_2\mathbf{W}_1 - \mathbf{W}_2\mathbf{W}_1\mathbf{W}_1^T\mathbf{W}_1, \tag{18}$$

$$\tau\left(\frac{d\mathbf{W}_2}{dt}\mathbf{W}_2^T + \mathbf{W}_2\frac{d\mathbf{W}_2^T}{dt}\right) = \tag{19}$$

$$(\tilde{\Sigma}^{yx} - \mathbf{W}_2\mathbf{W}_1)\mathbf{W}_1^T\mathbf{W}_2^T + \mathbf{W}_2\mathbf{W}_1(\tilde{\Sigma}^{yx} - \mathbf{W}_2\mathbf{W}_1)^T \tag{20}$$

$$= \tilde{\Sigma}^{yx}\mathbf{W}_1^T\mathbf{W}_2^T + \mathbf{W}_2\mathbf{W}_1(\tilde{\Sigma}^{yx})^T - \mathbf{W}_2\mathbf{W}_1\mathbf{W}_1^T\mathbf{W}_2^T - \mathbf{W}_2\mathbf{W}_1(\mathbf{W}_2\mathbf{W}_1)^T \tag{21}$$

$$= \tilde{\Sigma}^{yx}\mathbf{W}_1^T\mathbf{W}_2^T + \mathbf{W}_2\mathbf{W}_1(\tilde{\Sigma}^{yx})^T - \mathbf{W}_2\mathbf{W}_1\mathbf{W}_1^T\mathbf{W}_2^T - \mathbf{W}_2\mathbf{W}_1\mathbf{W}_1^T\mathbf{W}_2^T \tag{22}$$

$$= \tilde{\Sigma}^{yx}\mathbf{W}_1^T\mathbf{W}_2^T + \mathbf{W}_2\mathbf{W}_1(\tilde{\Sigma}^{yx})^T - \mathbf{W}_2\mathbf{W}_1\mathbf{W}_1^T\mathbf{W}_2^T - \mathbf{W}_2\mathbf{W}_2^T\mathbf{W}_2\mathbf{W}_2^T + \lambda\mathbf{W}_2\mathbf{W}_2^T. \tag{23}$$

Defining

$$\mathbf{F} = \begin{bmatrix} -\frac{\lambda}{2}I & (\tilde{\Sigma}^{yx})^T \\ \tilde{\Sigma}^{yx} & \frac{\lambda}{2}I \end{bmatrix}, \tag{24}$$

the gradient flow dynamics of $\mathbf{Q}\mathbf{Q}^T(t)$ can be written as a differential matrix Riccati equation

$$\tau\frac{d}{dt}(\mathbf{Q}\mathbf{Q}^T) = \mathbf{F}\mathbf{Q}\mathbf{Q}^T + \mathbf{Q}\mathbf{Q}^T\mathbf{F} - (\mathbf{Q}\mathbf{Q}^T)^2. \tag{25}$$

We write $\tau\frac{d}{dt}(\mathbf{Q}\mathbf{Q}^T)$ for completeness

$$\tau\frac{d}{dt}(\mathbf{Q}\mathbf{Q}^T) = \begin{bmatrix} -\frac{\lambda}{2} & (\tilde{\Sigma}^{yx})^T \\ \tilde{\Sigma}^{yx} & \frac{\lambda}{2} \end{bmatrix}\begin{bmatrix} \mathbf{W}_1^T\mathbf{W}_1 & \mathbf{W}_1^T\mathbf{W}_2 \\ \mathbf{W}_2\mathbf{W}_1 & \mathbf{W}_2\mathbf{W}_2^T \end{bmatrix} + \begin{bmatrix} \mathbf{W}_1^T\mathbf{W}_1 & \mathbf{W}_1^T\mathbf{W}_2 \\ \mathbf{W}_2\mathbf{W}_1 & \mathbf{W}_2\mathbf{W}_2^T \end{bmatrix}^T\begin{bmatrix} -\frac{\lambda}{2} & (\tilde{\Sigma}^{yx})^T \\ \tilde{\Sigma}^{yx} & \frac{\lambda}{2} \end{bmatrix}$$
$$- \begin{bmatrix} \mathbf{W}_1^T\mathbf{W}_1 & \mathbf{W}_1^T\mathbf{W}_2 \\ \mathbf{W}_2\mathbf{W}_1 & \mathbf{W}_2\mathbf{W}_2^T \end{bmatrix}^2$$

$$\tag{26}$$

$$
= \begin{bmatrix} -\frac{\lambda}{2} & (\tilde{\Sigma}^{yx})^T \\ \tilde{\Sigma}^{yx} & \frac{\lambda}{2} \end{bmatrix} \begin{bmatrix} \mathbf{W}_1^T \mathbf{W}_1 & \mathbf{W}_1^T \mathbf{W}_2 \\ \mathbf{W}_2 \mathbf{W}_1 & \mathbf{W}_2 \mathbf{W}_2^T \end{bmatrix} + \begin{bmatrix} \mathbf{W}_1^T \mathbf{W}_1 & \mathbf{W}_1^T \mathbf{W}_2 \\ \mathbf{W}_2 \mathbf{W}_1 & \mathbf{W}_2 \mathbf{W}_2^T \end{bmatrix}^T \begin{bmatrix} -\frac{\lambda}{2} & (\tilde{\Sigma}^{yx})^T \\ \tilde{\Sigma}^{yx} & \frac{\lambda}{2} \end{bmatrix}
$$
$$
- \begin{bmatrix} \mathbf{W}_1^T \mathbf{W}_1 & \mathbf{W}_1^T \mathbf{W}_2 \\ \mathbf{W}_2 \mathbf{W}_1 & \mathbf{W}_2 \mathbf{W}_2^T \end{bmatrix} \begin{bmatrix} \mathbf{W}_1^T \mathbf{W}_1 & \mathbf{W}_1^T \mathbf{W}_2 \\ \mathbf{W}_2 \mathbf{W}_1 & \mathbf{W}_2 \mathbf{W}_2^T \end{bmatrix} \tag{27}
$$

$$
= \begin{bmatrix} -\frac{\lambda}{2}\mathbf{W}_1^T\mathbf{W}_1 + (\tilde{\Sigma}^{yx})^T\mathbf{W}_2\mathbf{W}_1 & -\frac{\lambda}{2}\mathbf{W}_1^T\mathbf{W}_2 + (\tilde{\Sigma}^{yx})^T\mathbf{W}_2\mathbf{W}_2^T \\ \tilde{\Sigma}^{yx}\mathbf{W}_1^T\mathbf{W}_1 + \frac{\lambda}{2}\mathbf{W}_2\mathbf{W}_1 & \tilde{\Sigma}^{yx}\mathbf{W}_1^T\mathbf{W}_2^T + \frac{\lambda}{2}\mathbf{W}_2\mathbf{W}_2^T \end{bmatrix}
$$
$$
+ \begin{bmatrix} -\frac{\lambda}{2}\mathbf{W}_1^T\mathbf{W}_1 + \mathbf{W}_1^T\mathbf{W}_1(\tilde{\Sigma}^{yx})^T & \frac{\lambda}{2}\mathbf{W}_1^T\mathbf{W}_2 + \mathbf{W}_1^T\mathbf{W}_2(\tilde{\Sigma}^{yx})^T \\ -\frac{\lambda}{2}\mathbf{W}_2^T\mathbf{W}_1 + \mathbf{W}_2\mathbf{W}_1(\tilde{\Sigma}^{yx})^T & \frac{\lambda}{2}\mathbf{W}_2\mathbf{W}_2^T + \mathbf{W}_2\mathbf{W}_2^T(\tilde{\Sigma}^{yx})^T \end{bmatrix}
$$
$$
- \begin{bmatrix} \mathbf{W}_1^T \mathbf{W}_1 & \mathbf{W}_1^T \mathbf{W}_2 \\ \mathbf{W}_2 \mathbf{W}_1 & \mathbf{W}_2 \mathbf{W}_2^T \end{bmatrix} \begin{bmatrix} \mathbf{W}_1^T \mathbf{W}_1 & \mathbf{W}_1^T \mathbf{W}_2 \\ \mathbf{W}_2 \mathbf{W}_1 & \mathbf{W}_2 \mathbf{W}_2^T \end{bmatrix} \tag{28}
$$

$$
= \begin{bmatrix} -\frac{\lambda}{2}\mathbf{W}_1^T\mathbf{W}_1 + (\tilde{\Sigma}^{yx})^T\mathbf{W}_2\mathbf{W}_1 & -\frac{\lambda}{2}\mathbf{W}_1^T\mathbf{W}_2 + (\tilde{\Sigma}^{yx})^T\mathbf{W}_2\mathbf{W}_2^T \\ \tilde{\Sigma}^{yx}\mathbf{W}_1^T\mathbf{W}_1 + \frac{\lambda}{2}\mathbf{W}_2\mathbf{W}_1 & \tilde{\Sigma}^{yx}\mathbf{W}_1^T\mathbf{W}_2^T + \frac{\lambda}{2}\mathbf{W}_2\mathbf{W}_2^T \end{bmatrix}
$$
$$
+ \begin{bmatrix} -\frac{\lambda}{2}\mathbf{W}_1^T\mathbf{W}_1 + \mathbf{W}_1^T\mathbf{W}_1(\tilde{\Sigma}^{yx})^T & \frac{\lambda}{2}\mathbf{W}_1^T\mathbf{W}_2 + \mathbf{W}_1^T\mathbf{W}_2(\tilde{\Sigma}^{yx})^T \\ -\frac{\lambda}{2}\mathbf{W}_2^T\mathbf{W}_1 + \mathbf{W}_2\mathbf{W}_1(\tilde{\Sigma}^{yx})^T & \frac{\lambda}{2}\mathbf{W}_2\mathbf{W}_2^T + \mathbf{W}_2\mathbf{W}_2^T(\tilde{\Sigma}^{yx})^T \end{bmatrix}
$$
$$
- \begin{bmatrix} \mathbf{W}_1^T\mathbf{W}_1\mathbf{W}_1^T\mathbf{W}_1 + \mathbf{W}_1^T\mathbf{W}_2\mathbf{W}_2^T\mathbf{W}_1 & \mathbf{W}_1^T\mathbf{W}_1\mathbf{W}_1^T\mathbf{W}_2 + \mathbf{W}_1^T\mathbf{W}_2\mathbf{W}_2^T\mathbf{W}_2 \\ \mathbf{W}_2\mathbf{W}_1\mathbf{W}_1^T\mathbf{W}_1 + \mathbf{W}_2\mathbf{W}_2^T\mathbf{W}_2\mathbf{W}_1 & \mathbf{W}_2\mathbf{W}_1\mathbf{W}_1^T\mathbf{W}_2 + \mathbf{W}_2\mathbf{W}_2^T\mathbf{W}_2\mathbf{W}_2^T \end{bmatrix} \tag{45}
$$
$$
\square
$$

The four quadrants of 8 are equivalent to equations 12, 15, 18, and 23 respectively.

## C.2 $\mathbf{QQ}^T$ Diagonalisation

**Lemma C.2.** *If $\boldsymbol{F} = \boldsymbol{P}\boldsymbol{\Lambda}\boldsymbol{P}^T$ is symmetric and diagonalizable, then the matrix Riccati differential equation $\tau\frac{d}{dt}(\mathbf{QQ}^T) = \mathbf{F}\mathbf{QQ}^T + \mathbf{QQ}^T\mathbf{F} - (\mathbf{QQ}^T)^2$ with initialization $\mathbf{QQ}^T(0) = \mathbf{Q}(0)\mathbf{Q}(0)^T$ has a unique solution for all $t \geq 0$, and the solution is given by*

$$
\mathbf{QQ}^T(t) = e^{\mathbf{F}\frac{t}{\tau}}\mathbf{Q}(0)\left[\mathbf{I} + \mathbf{Q}(0)^T\boldsymbol{P}\left(\frac{e^{2\boldsymbol{\Lambda}\frac{t}{\tau}} - \mathbf{I}}{2\boldsymbol{\Lambda}}\right)\boldsymbol{P}^T\mathbf{Q}(0)\right]^{-1}\mathbf{Q}(0)^T e^{\mathbf{F}\frac{t}{\tau}}. \tag{29}
$$

*This is true even when there exists $\boldsymbol{\Lambda}_i = 0$.*

*Proof.* First we show that there exists a unique solution to the initial value problem stated. This is true by Picard-Lindelöf theorem. Now we show that the provided solution satisfies the ODE. Let $\boldsymbol{L} = e^{\mathbf{F}\frac{t}{\tau}}\mathbf{Q}(0)$ and $\boldsymbol{C} = \mathbf{I} + \mathbf{Q}(0)^T\boldsymbol{P}\left(\frac{e^{2\boldsymbol{\Lambda}\frac{t}{\tau}} - \mathbf{I}}{2\boldsymbol{\Lambda}}\right)\boldsymbol{P}^T\mathbf{Q}(0)$ such that solution $\mathbf{QQ}^T(t) = \boldsymbol{L}\boldsymbol{C}^{-1}\boldsymbol{L}^T$. The time derivative of $\mathbf{QQ}^T$ is then given by

$$
\tau\frac{d}{dt}(\mathbf{QQ}^T) = \tau\left(\frac{d}{dt}(\boldsymbol{L})\boldsymbol{C}^{-1}\boldsymbol{L}^T + \boldsymbol{L}\frac{d}{dt}(\boldsymbol{C}^{-1})\boldsymbol{L}^T + \boldsymbol{L}\boldsymbol{C}^{-1}\frac{d}{dt}(\boldsymbol{L}^T)\right) \tag{30}
$$

Solving for these derivatives individually, we find

$$
\tau\frac{d}{dt}(\boldsymbol{L}) = \tau\frac{d}{dt}e^{\mathbf{F}\frac{t}{\tau}}\mathbf{Q}(0) = \boldsymbol{F}e^{\mathbf{F}\frac{t}{\tau}}\mathbf{Q}(0) = \boldsymbol{F}\boldsymbol{L} \tag{31}
$$

$$
\tau\frac{d}{dt}(\boldsymbol{C}^{-1}) = -\tau\boldsymbol{C}^{-1}\frac{d}{dt}(\boldsymbol{C})\boldsymbol{C}^{-1} = -\tau\boldsymbol{C}^{-1}\mathbf{Q}(0)^T\boldsymbol{P}\frac{d}{dt}\left(\frac{e^{2\boldsymbol{\Lambda}\frac{t}{\tau}} - \mathbf{I}}{2\boldsymbol{\Lambda}}\right)\boldsymbol{P}^T\mathbf{Q}(0)\boldsymbol{C}^{-1} \tag{32}
$$

We consider the derivative of the fraction serpately,

$$
\tau\frac{d}{dt}\left(\frac{e^{2\boldsymbol{\Lambda}\frac{t}{\tau}} - \mathbf{I}}{2\boldsymbol{\Lambda}}\right) = e^{2\boldsymbol{\Lambda}\frac{t}{\tau}} \tag{33}
$$

this is true even in the limit as $\lambda_i \to 0$. Plugging these derivatives back in we see that the solution satisfies the ODE. Lastly, let $t = 0$, we see that the the solution satisfies the initial conditions. $\square$

In Appendix C.2 we prove that this equation is the unique solution to the initial value problem derived in Lemma C.2 no matter the value of $\Lambda$. However, as discussed in Braun et al. (2022), the solution in this form is not very useable or interpretable due to the matrix inverse mixing the blocks of $\mathbf{QQ}^T$. Additionally, we need to diagonalize $\tilde{F}$. To do so we consider the compact singular value decomposition $\text{SVD}(\tilde{\boldsymbol{\Sigma}}^{yx}) = \tilde{\mathbf{U}}\tilde{\mathbf{S}}\tilde{\mathbf{V}}^T$. Here, $\tilde{\mathbf{U}} \in \mathbb{R}^{N_o \times N_h}$ denote the left singular vectors, $\tilde{\mathbf{S}} \in \mathbb{R}^{N_h \times N_h}$ the square matrix with ordered, non-zero eigenvalues on its diagonal, and $\tilde{\mathbf{V}} \in \mathbb{R}^{N_i \times N_h}$ the corresponding right singular vectors. For unequal input-output dimensions ($N_i \neq N_o$), the right and left singular vectors are not square. Accordingly, for the case $N_i > N_h = N_o$, we define $\tilde{\mathbf{U}}^{\perp} \in \mathbb{R}^{N_o \times |N_o - N_i|}$ as a matrix containing orthogonal column vectors that complete the basis for $\tilde{\mathbf{U}}$, i.e., make $[\tilde{\mathbf{U}}\ \tilde{\mathbf{U}}^{\perp}]$ orthonormal, and $\tilde{\mathbf{V}}^{\perp} \in \mathbb{R}^{N_i \times |N_o - N_i|}$ as a matrix of zeros. Conversely, when $N_i = N_h < N_o$, then $\tilde{\mathbf{V}}^{\perp}$ is a matrix containing orthogonal column vectors that complete the basis for $\tilde{V}$ and $\tilde{\mathbf{U}}^{\perp}$ is a matrix of zeros. Using this SVD structure we can now describe the eigendecomposition of $\mathbf{F}$.

## C.3 $F$ Diagonalization

**Lemma C.3.** *Under assumptions of full-rank 5, the eigendecomposition of* $\mathbf{F} = \mathbf{P}\boldsymbol{\Lambda}\mathbf{P}^T$ *where*

$$\mathbf{P} = \frac{1}{\sqrt{2}}\begin{pmatrix} \tilde{\boldsymbol{V}}(\tilde{\boldsymbol{G}} - \tilde{\boldsymbol{H}}\tilde{\boldsymbol{G}}) & \tilde{\boldsymbol{V}}(\tilde{\boldsymbol{G}} + \tilde{\boldsymbol{H}}\tilde{\boldsymbol{G}}) & \sqrt{2}\tilde{\boldsymbol{V}}_{\perp} \\ \tilde{\boldsymbol{U}}(\tilde{\boldsymbol{G}} + \tilde{\boldsymbol{H}}\tilde{\boldsymbol{G}}) & -\tilde{\boldsymbol{U}}(\tilde{\boldsymbol{G}} - \tilde{\boldsymbol{H}}\tilde{\boldsymbol{G}}) & \sqrt{2}\tilde{\boldsymbol{U}}_{\perp} \end{pmatrix}, \quad \boldsymbol{\Lambda} = \begin{pmatrix} \tilde{\boldsymbol{S}}_{\lambda} & 0 & 0 \\ 0 & -\tilde{\boldsymbol{S}}_{\lambda} & 0 \\ 0 & 0 & \boldsymbol{\lambda}_{\perp} \end{pmatrix} \quad (34)$$

*and the matrices* $\tilde{\boldsymbol{S}}_{\lambda}$, $\boldsymbol{\lambda}_{\perp}$, $\tilde{\boldsymbol{H}}$, *and* $\tilde{\boldsymbol{G}}$ *are the diagonal matrices defined as:*

$$\tilde{\boldsymbol{S}}_{\lambda} = \sqrt{\tilde{\boldsymbol{S}}^2 + \frac{\lambda^2}{4}\mathbf{I}}, \quad \boldsymbol{\lambda}_{\perp} = \text{sgn}(N_o - N_i)\frac{\lambda}{2}\mathbf{I}, \quad \tilde{\boldsymbol{H}} = \text{sgn}(\lambda)\sqrt{\frac{\tilde{\boldsymbol{S}}_{\lambda} - \tilde{\boldsymbol{S}}}{\tilde{\boldsymbol{S}}_{\lambda} + \tilde{\boldsymbol{S}}}}, \quad \tilde{\boldsymbol{G}} = \frac{1}{\sqrt{\mathbf{I} + \tilde{\boldsymbol{H}}^2}}. \quad (35)$$

Beyond the invertibility of $F$, notice from the equation (Fukumizu solution) we need to understand the relationship between $F$ and $Q(0)$. To do this the following lemma relates the structure between the SVD of the model with the SVD structure of the individual parameters.

*Proof.* We leave for the reader by computing

$$\boldsymbol{F} = \boldsymbol{P}\boldsymbol{\Lambda}\boldsymbol{P}^T \quad (36)$$

$\square$

## C.4 Solution Unequal-Input-Output

**Theorem C.4.** *Under the assumptions of whitened inputs, 2, lambda-balanced weights 3, no bottleneck 4, and full rank 5, the temporal dynamics of* $\mathbf{QQ}^T$ *are*

$$\mathbf{QQ}^T(t) = \begin{pmatrix} \boldsymbol{Z_1}\boldsymbol{A}^{-1}\boldsymbol{Z_1}^T & \boldsymbol{Z_1}\boldsymbol{A}^{-1}\boldsymbol{Z_2}^T \\ \boldsymbol{Z_2}\boldsymbol{A}^{-1}\boldsymbol{Z_1}^T & \boldsymbol{Z_2}\boldsymbol{A}^{-1}\boldsymbol{Z_2}^T \end{pmatrix},$$

*where the variables* $\boldsymbol{Z_1} \in \mathbb{R}^{N_i \times N_h}$, $\boldsymbol{Z_2} \in \mathbb{R}^{N_o \times N_h}$, *and* $\boldsymbol{A} \in \mathbb{R}^{N_h \times N_h}$ *are defined as*

$$\boldsymbol{Z_1}(t) = \frac{1}{2}\tilde{\boldsymbol{V}}(\tilde{\boldsymbol{G}} - \tilde{\boldsymbol{H}}\tilde{\boldsymbol{G}})e^{\tilde{\boldsymbol{S}}_{\lambda}\frac{t}{\tau}}\boldsymbol{B}^T - \frac{1}{2}\tilde{\boldsymbol{V}}(\tilde{\boldsymbol{G}} + \tilde{\boldsymbol{H}}\tilde{\boldsymbol{G}})e^{-\tilde{\boldsymbol{S}}_{\lambda}\frac{t}{\tau}}\boldsymbol{C}^T + \tilde{\boldsymbol{V}}_{\perp}e^{\boldsymbol{\lambda}_{\perp}\frac{t}{\tau}}\tilde{\boldsymbol{V}}_{\perp}^T\mathbf{W}_1(0)^T \quad (37)$$

$$\boldsymbol{Z_2}(t) = \frac{1}{2}\tilde{\boldsymbol{U}}(\tilde{\boldsymbol{G}} + \tilde{\boldsymbol{H}}\tilde{\boldsymbol{G}})e^{\tilde{\boldsymbol{S}}_{\lambda}\frac{t}{\tau}}\boldsymbol{B}^T + \frac{1}{2}\tilde{\boldsymbol{U}}(\tilde{\boldsymbol{G}} - \tilde{\boldsymbol{H}}\tilde{\boldsymbol{G}})e^{-\tilde{\boldsymbol{S}}_{\lambda}\frac{t}{\tau}}\boldsymbol{C}^T + \tilde{\boldsymbol{U}}_{\perp}e^{\boldsymbol{\lambda}_{\perp}\frac{t}{\tau}}\tilde{\boldsymbol{U}}_{\perp}^T\mathbf{W}_2(0) \quad (38)$$

$$\boldsymbol{A}(t) = \mathbf{I} + \boldsymbol{B}\left(\frac{e^{2\tilde{\boldsymbol{S}}_{\lambda}\frac{t}{\tau}} - \mathbf{I}}{4\tilde{\boldsymbol{S}}_{\lambda}}\right)\boldsymbol{B}^T - \boldsymbol{C}\left(\frac{e^{-2\tilde{\boldsymbol{S}}_{\lambda}\frac{t}{\tau}} - \mathbf{I}}{4\tilde{\boldsymbol{S}}_{\lambda}}\right)\boldsymbol{C}^T + \mathbf{W}_2(0)^T\tilde{\boldsymbol{U}}_{\perp}\left(\frac{e^{\boldsymbol{\lambda}_{\perp}\frac{t}{\tau}} - \mathbf{I}}{\boldsymbol{\lambda}_{\perp}}\right)\tilde{\boldsymbol{U}}_{\perp}^T\mathbf{W}_2(0)$$

$$+ \mathbf{W}_1(0)\tilde{\boldsymbol{V}}_{\perp}\left(\frac{e^{\boldsymbol{\lambda}_{\perp}\frac{t}{\tau}} - \mathbf{I}}{\boldsymbol{\lambda}_{\perp}}\right)\tilde{\boldsymbol{V}}_{\perp}^T\mathbf{W}_1(0)^T \quad (39)$$

*Proof.* We start and use the diagonalization of $\mathbf{F}$ to rewrite the matrix exponential of $\boldsymbol{F}$ and $\boldsymbol{F}$. Note that $\mathbf{P}^T\mathbf{P} = \mathbf{PP}^T = \mathbf{I}$ and therefore $\mathbf{P}^T = \mathbf{P}^{-1}$.

$$e^{\mathbf{F}\frac{t}{\tau}} = \mathbf{P}e^{\mathbf{\Gamma}}\mathbf{P}^T$$

$$= \frac{1}{\sqrt{2}}\begin{bmatrix} \tilde{\mathbf{V}}(\tilde{G}-\check{H}\tilde{G}) & \tilde{\mathbf{V}}(\tilde{G}+\check{H}\tilde{G}) & \sqrt{2}\mathbf{V}_{\perp} \\ \tilde{\mathbf{U}}(\tilde{G}+\check{H}\tilde{G}) & -\tilde{\mathbf{U}}(\tilde{G}-\check{H}\tilde{G}) & \sqrt{2}\mathbf{U}_{\perp} \end{bmatrix} \begin{bmatrix} e^{\tilde{S}_\lambda \frac{t}{\tau}} & 0 & 0 \\ 0 & e^{-\tilde{S}_\lambda \frac{t}{\tau}} & 0 \\ 0 & 0 & e^{\lambda_{\perp}\frac{t}{\tau}} \end{bmatrix} \frac{1}{\sqrt{2}}\begin{bmatrix} \tilde{\mathbf{V}}(\tilde{G}-\check{H}\tilde{G}) & \tilde{\mathbf{V}}(\tilde{G}+\check{H}\tilde{G}) & \sqrt{2}\mathbf{V}_{\perp} \\ \tilde{\mathbf{U}}(\tilde{G}+\check{H}\tilde{G}) & -\tilde{\mathbf{U}}(\tilde{G}-\check{H}\tilde{G}) & \sqrt{2}\mathbf{U}_{\perp} \end{bmatrix}^T$$

$$= \frac{1}{\sqrt{2}}\begin{bmatrix} \tilde{\mathbf{V}}(\tilde{G}-\check{H}\tilde{G}) & \tilde{\mathbf{V}}(\tilde{G}+\check{H}\tilde{G}) \\ \tilde{\mathbf{U}}(\tilde{G}-\check{H}\tilde{G}) & -\tilde{\mathbf{U}}(\tilde{G}+\check{H}\tilde{G}) \end{bmatrix} \begin{bmatrix} e^{\tilde{S}_\lambda \frac{t}{\tau}} & 0 \\ 0 & e^{-\tilde{S}_\lambda \frac{t}{\tau}} \end{bmatrix} \frac{1}{\sqrt{2}}\begin{bmatrix} \tilde{\mathbf{V}}(\tilde{G}-\check{H}\tilde{G}) & \tilde{\mathbf{V}}(\tilde{G}+\check{H}\tilde{G}) \\ \tilde{\mathbf{U}}(\tilde{G}+\check{H}\tilde{G}) & -\tilde{\mathbf{U}}(\tilde{G}-\check{H}\tilde{G}) \end{bmatrix}^T + 2\frac{1}{\sqrt{2}}\begin{bmatrix} \tilde{\mathbf{V}}_{\perp} \\ \tilde{\mathbf{U}}_{\perp} \end{bmatrix} e^{\lambda_{\perp}\frac{t}{\tau}} \frac{1}{\sqrt{2}}\begin{bmatrix} \tilde{\mathbf{V}}_{\perp} \\ \tilde{\mathbf{U}}_{\perp} \end{bmatrix}^T$$

$$= \mathbf{O}e^{\mathbf{\Lambda}\frac{t}{\tau}}\mathbf{O} + 2\mathbf{M}e^{\lambda_{\perp}\frac{t}{\tau}}\mathbf{M}^T. \tag{40}$$

$$e^{\mathbf{F}\frac{t}{\tau}}\mathbf{F}^{-1}e^{\mathbf{F}\frac{t}{\tau}} - \mathbf{F}^{-1} = \mathbf{O}e^{\mathbf{\Lambda}\frac{t}{\tau}}\mathbf{O}^T\mathbf{O}\mathbf{\Lambda}^{-1}\mathbf{O}^T\mathbf{O}e^{\mathbf{\Lambda}\frac{t}{\tau}}\mathbf{O}^T - \mathbf{O}\mathbf{\Lambda}^{-1}\mathbf{O}^T + \mathbf{M}(e^{\lambda_{\perp}\frac{t}{\tau}} - \mathbf{I})(\lambda_{\perp})^{-1}\mathbf{M}^T. \tag{41}$$

$$\mathbf{F} = \mathbf{O}\mathbf{\Lambda}\mathbf{O}^T + 2\mathbf{M}\lambda_{\perp}\mathbf{M}^T \tag{42}$$

Where $\boldsymbol{M} = \frac{1}{\sqrt{2}}\begin{bmatrix} \tilde{\mathbf{V}}_{\perp} \\ \tilde{\mathbf{U}}_{\perp} \end{bmatrix}^T$. Placing these expressions into equation 29 gives

$$\mathbf{QQ}^T(t) = \left[\mathbf{O}e^{\mathbf{\Lambda}\frac{t}{\tau}}\mathbf{O}^T + 2\mathbf{M}e^{\lambda_{\perp}\frac{t}{\tau}}\mathbf{M}^T\right]\mathbf{Q}(0)$$

$$\left[\mathbf{I} + \frac{1}{2}\mathbf{Q}(0)^T\left(\mathbf{O}\left(e^{2\mathbf{\Lambda}\frac{t}{\tau}} - \mathbf{I}\right)\mathbf{\Lambda}^{-1}\mathbf{O}^T + \mathbf{M}(e^{\lambda_{\perp}\frac{t}{\tau}} - \mathbf{I})\lambda_{\perp}^{-1}\mathbf{M}^T\right)\mathbf{Q}(0)\right]^{-1} \tag{43}$$

$$\mathbf{Q}(0)^T\left[\mathbf{O}e^{\mathbf{\Lambda}\frac{t}{\tau}}\mathbf{O}^T + 2\mathbf{M}e^{\lambda_{\perp}\frac{t}{\tau}}\mathbf{M}^T\right]^T$$

$$\mathbf{O}^T\mathbf{Q}(0) = \frac{1}{\sqrt{2}}\begin{pmatrix} \tilde{\mathbf{V}}(\tilde{G}-\check{H}\tilde{G}) & \tilde{\mathbf{V}}(\tilde{G}+\check{H}\tilde{G}) \\ \tilde{\mathbf{U}}(\tilde{G}+\check{H}\tilde{G}) & -\tilde{\mathbf{U}}(\tilde{G}-\check{H}\tilde{G}) \end{pmatrix}^T \begin{pmatrix} \mathbf{W}_1^T(0) \\ \mathbf{W}_2(0) \end{pmatrix}$$

$$= \frac{1}{\sqrt{2}}\begin{pmatrix} (\tilde{G}-\check{H}\tilde{G})\tilde{\mathbf{V}}^T\mathbf{W}_1^T(0) + (\tilde{G}+\check{H}\tilde{G})\tilde{\mathbf{U}}^T\mathbf{W}_2(0) \\ (\tilde{G}+\check{H}\tilde{G})\tilde{\mathbf{V}}^T\mathbf{W}_1^T(0) - (\tilde{G}-\check{H}\tilde{G})\tilde{\mathbf{U}}^T\mathbf{W}_2(0) \end{pmatrix}$$

$$= \frac{1}{\sqrt{2}}\begin{pmatrix} \boldsymbol{B}^T \\ -\boldsymbol{C}^T \end{pmatrix} \tag{44}$$

where

$$\mathbf{B} = \mathbf{W}_2(0)^T\tilde{\mathbf{U}}(\tilde{G}+\check{H}\tilde{G}) + \mathbf{W}_1(0)\tilde{\mathbf{V}}(\tilde{G}-\check{H}\tilde{G}) \in \mathbb{R}^{N_h \times N_h} \tag{45}$$

$$\mathbf{C} = \mathbf{W}_2(0)^T\tilde{\mathbf{U}}(\tilde{G}-\check{H}\tilde{G}) - \mathbf{W}_1(0)\tilde{\mathbf{V}}(\tilde{G}+\check{H}\tilde{G}) \in \mathbb{R}^{N_h \times N_h} \tag{46}$$

$$\mathbf{O}e^{\mathbf{\Lambda}t/\tau} = \frac{1}{\sqrt{2}}\begin{pmatrix} \tilde{\mathbf{V}}(\tilde{G}-\check{H}\tilde{G}) & \tilde{\mathbf{V}}(\tilde{G}+\check{H}\tilde{G}) \\ \tilde{\mathbf{U}}(\tilde{G}+\check{H}\tilde{G}) & -\tilde{\mathbf{U}}(\tilde{G}-\check{H}\tilde{G}) \end{pmatrix} \begin{pmatrix} e^{\tilde{S}_\lambda \frac{t}{\tau}} & 0 \\ 0 & e^{-\tilde{S}_\lambda \frac{t}{\tau}} \end{pmatrix}$$

$$= \frac{1}{\sqrt{2}}\begin{pmatrix} \tilde{\mathbf{V}}(\tilde{G}-\check{H}\tilde{G})e^{\tilde{S}_\lambda \frac{t}{\tau}} & \tilde{\mathbf{V}}(\tilde{G}+\check{H}\tilde{G})e^{-\tilde{S}_\lambda \frac{t}{\tau}} \\ \tilde{\mathbf{U}}(\tilde{G}+\check{H}\tilde{G})e^{\tilde{S}_\lambda \frac{t}{\tau}} & -\tilde{\mathbf{U}}(\tilde{G}-\check{H}\tilde{G})e^{-\tilde{S}_\lambda \frac{t}{\tau}} \end{pmatrix} \tag{47}$$

$$\mathbf{O}e^{\mathbf{\Lambda}t/\tau}\mathbf{O}^T\mathbf{Q}(0) = \frac{1}{2}\begin{pmatrix} \tilde{\mathbf{V}}(\tilde{G}-\check{H}\tilde{G})e^{\tilde{S}_\lambda \frac{t}{\tau}} & \tilde{\mathbf{V}}(\tilde{G}+\check{H}\tilde{G})e^{-\tilde{S}_\lambda \frac{t}{\tau}} \\ \tilde{\mathbf{U}}(\tilde{G}+\check{H}\tilde{G})e^{\tilde{S}_\lambda \frac{t}{\tau}} & -\tilde{\mathbf{U}}(\tilde{G}-\check{H}\tilde{G})e^{-\tilde{S}_\lambda \frac{t}{\tau}} \end{pmatrix}\begin{pmatrix} \boldsymbol{B}^T \\ -\boldsymbol{C}^T \end{pmatrix}$$

$$= \frac{1}{2}\begin{pmatrix} \tilde{\mathbf{V}}(\tilde{G}-\check{H}\tilde{G})e^{\tilde{S}_\lambda \frac{t}{\tau}}\boldsymbol{B}^T - \tilde{\mathbf{V}}(\tilde{G}+\check{H}\tilde{G})e^{-\tilde{S}_\lambda \frac{t}{\tau}}\boldsymbol{C}^T \\ \tilde{\mathbf{U}}(\tilde{G}+\check{H}\tilde{G})e^{\tilde{S}_\lambda \frac{t}{\tau}}\boldsymbol{B}^T + \tilde{\mathbf{U}}(\tilde{G}-\check{H}\tilde{G})e^{-\tilde{S}_\lambda \frac{t}{\tau}}\boldsymbol{C}^T \end{pmatrix} \tag{48}$$

$$2\mathbf{M}e^{\boldsymbol{\lambda}_\perp \frac{t}{\tau}}\mathbf{M}^T\mathbf{Q}(0) = 2\frac{1}{\sqrt{2}}\begin{bmatrix}\tilde{\mathbf{V}}_\perp \\ \tilde{\mathbf{U}}_\perp\end{bmatrix}\begin{bmatrix}e^{\boldsymbol{\lambda}_\perp \frac{t}{\tau}} & 0 \\ 0 & e^{\boldsymbol{\lambda}_\perp \frac{t}{\tau}}\end{bmatrix}\frac{1}{\sqrt{2}}\begin{bmatrix}\tilde{\mathbf{V}}_\perp \\ \tilde{\mathbf{U}}_\perp\end{bmatrix}^T\begin{bmatrix}\mathbf{W}_1(0)^T \\ \mathbf{W}_2(0)\end{bmatrix}$$

$$= \begin{bmatrix}\tilde{\mathbf{V}}_\perp e^{\boldsymbol{\lambda}_\perp \frac{t}{\tau}}\tilde{\mathbf{V}}_\perp^T & 0 \\ 0 & \tilde{\mathbf{U}}_\perp e^{\boldsymbol{\lambda}_\perp \frac{t}{\tau}}\tilde{\mathbf{U}}_\perp^T\end{bmatrix}\begin{bmatrix}\mathbf{W}_1(0)^T \\ \mathbf{W}_2(0)\end{bmatrix}$$

$$= \begin{bmatrix}\tilde{\mathbf{V}}_\perp e^{\boldsymbol{\lambda}_\perp \frac{t}{\tau}}\tilde{\mathbf{V}}_\perp^T\mathbf{W}_1(0)^T \\ \tilde{\mathbf{U}}_\perp e^{\boldsymbol{\lambda}_\perp \frac{t}{\tau}}\tilde{\mathbf{U}}_\perp^T\mathbf{W}_2(0)\end{bmatrix}$$

$$(49)$$

Putting it together we get the expressions for $\boldsymbol{Z_1}(t)$ and $\boldsymbol{Z_2}(t)$

$$\left[\mathbf{O}e^{\boldsymbol{\Lambda}\frac{t}{\tau}}\mathbf{O}^T + 2\mathbf{M}e^{\boldsymbol{\lambda}_\perp \frac{t}{\tau}}\mathbf{M}^T\right]\mathbf{Q}(0) =$$

$$= \frac{1}{2}\begin{pmatrix}\tilde{\boldsymbol{V}}(\tilde{\boldsymbol{G}} - \tilde{\boldsymbol{H}}\tilde{\boldsymbol{G}})e^{\tilde{\boldsymbol{S}}_\lambda \frac{t}{\tau}}\boldsymbol{B}^T - \tilde{\boldsymbol{V}}(\tilde{\boldsymbol{G}} + \tilde{\boldsymbol{H}}\tilde{\boldsymbol{G}})e^{-\tilde{\boldsymbol{S}}_\lambda \frac{t}{\tau}}\boldsymbol{C}^T \\ \tilde{\boldsymbol{U}}(\tilde{\boldsymbol{G}} + \tilde{\boldsymbol{H}}\tilde{\boldsymbol{G}})e^{\tilde{\boldsymbol{S}}_\lambda \frac{t}{\tau}}\boldsymbol{B}^T + \tilde{\boldsymbol{U}}(\tilde{\boldsymbol{G}} - \tilde{\boldsymbol{H}}\tilde{\boldsymbol{G}})e^{-\tilde{\boldsymbol{S}}_\lambda \frac{t}{\tau}}\boldsymbol{C}^T\end{pmatrix} + \begin{bmatrix}\tilde{\mathbf{V}}_\perp e^{\boldsymbol{\lambda}_\perp \frac{t}{\tau}}\tilde{\mathbf{V}}_\perp^T\mathbf{W}_1(0)^T \\ \tilde{\mathbf{U}}_\perp e^{\boldsymbol{\lambda}_\perp \frac{t}{\tau}}\tilde{\mathbf{U}}_\perp^T\mathbf{W}_2(0)\end{bmatrix} \quad (50)$$

$$\boldsymbol{Z_1}(t) = \frac{1}{2}\tilde{\boldsymbol{V}}(\tilde{\boldsymbol{G}} - \tilde{\boldsymbol{H}}\tilde{\boldsymbol{G}})e^{\tilde{\boldsymbol{S}}_\lambda \frac{t}{\tau}}\boldsymbol{B}^T - \frac{1}{2}\tilde{\boldsymbol{V}}(\tilde{\boldsymbol{G}} + \tilde{\boldsymbol{H}}\tilde{\boldsymbol{G}})e^{-\tilde{\boldsymbol{S}}_\lambda \frac{t}{\tau}}\boldsymbol{C}^T + \tilde{\mathbf{V}}_\perp e^{\boldsymbol{\lambda}_\perp \frac{t}{\tau}}\tilde{\mathbf{V}}_\perp^T\mathbf{W}_1(0)^T \quad (51)$$

$$\boldsymbol{Z_2}(t) = \frac{1}{2}\tilde{\boldsymbol{U}}(\tilde{\boldsymbol{G}} + \tilde{\boldsymbol{H}}\tilde{\boldsymbol{G}})e^{\tilde{\boldsymbol{S}}_\lambda \frac{t}{\tau}}\boldsymbol{B}^T + \frac{1}{2}\tilde{\boldsymbol{U}}(\tilde{\boldsymbol{G}} - \tilde{\boldsymbol{H}}\tilde{\boldsymbol{G}})e^{-\tilde{\boldsymbol{S}}_\lambda \frac{t}{\tau}}\boldsymbol{C}^T + \tilde{\mathbf{U}}_\perp e^{\boldsymbol{\lambda}_\perp \frac{t}{\tau}}\tilde{\mathbf{U}}_\perp^T\mathbf{W}_2(0) \quad (52)$$

We now compute the terms inside the inverse

$$\mathbf{Q}(0)^T\mathbf{M}(e^{\boldsymbol{\lambda}_\perp \frac{t}{\tau}})\boldsymbol{\lambda}_\perp^{-1}\mathbf{M}^T\mathbf{Q}(0)$$

$$= \begin{bmatrix}\mathbf{W}_1(0) & \mathbf{W}_2(0)^T\end{bmatrix}\frac{1}{\sqrt{2}}\begin{bmatrix}\tilde{\mathbf{V}}_\perp \\ \tilde{\mathbf{U}}_\perp\end{bmatrix}\begin{bmatrix}e^{\boldsymbol{\lambda}_\perp \frac{t}{\tau}} & 0 \\ 0 & e^{\boldsymbol{\lambda}_\perp \frac{t}{\tau}}\end{bmatrix}\begin{bmatrix}\boldsymbol{\lambda}_\perp & 0 \\ 0 & \lambda_\perp\end{bmatrix}^{-1}\frac{1}{\sqrt{2}}\begin{bmatrix}\tilde{\mathbf{V}}_\perp \\ \tilde{\mathbf{U}}_\perp\end{bmatrix}^T\begin{bmatrix}\mathbf{W}_1(0)^T \\ \mathbf{W}_2(0)\end{bmatrix}$$

$$= \begin{bmatrix}\mathbf{W}_1(0) & \mathbf{W}_2(0)^T\end{bmatrix}\begin{bmatrix}e^{\boldsymbol{\lambda}_\perp \frac{t}{\tau}}\boldsymbol{\lambda}_\perp^{-1}\tilde{\mathbf{V}}_\perp\tilde{\mathbf{V}}_\perp^T\mathbf{W}_1(0)^T \\ e^{\boldsymbol{\lambda}_\perp \frac{t}{\tau}}\boldsymbol{\lambda}_\perp^{-1}\tilde{\mathbf{U}}_\perp\tilde{\mathbf{U}}_\perp^T\mathbf{W}_2(0)\end{bmatrix}$$

$$= \left[\left(\mathbf{W}_1(0)\tilde{\mathbf{V}}_\perp e^{\boldsymbol{\lambda}_\perp \frac{t}{\tau}}\boldsymbol{\lambda}_\perp^{-1}\tilde{\mathbf{V}}_\perp^T\mathbf{W}_1(0)^T + \mathbf{W}_2(0)^T\tilde{\mathbf{U}}_\perp e^{\boldsymbol{\lambda}_\perp \frac{t}{\tau}}\boldsymbol{\lambda}_\perp^{-1}\tilde{\mathbf{U}}_\perp^T\mathbf{W}_2(0)\right)\right] \quad (53)$$

$$\mathbf{Q}(0)^T\mathbf{M}\boldsymbol{\lambda}_\perp^{-1}\mathbf{M}^T\mathbf{Q}(0) = 2\begin{bmatrix}\mathbf{W}_1(0) & \mathbf{W}_2(0)^T\end{bmatrix}\frac{1}{\sqrt{2}}\begin{bmatrix}\tilde{\mathbf{V}}_\perp \\ \tilde{\mathbf{U}}_\perp\end{bmatrix}\begin{bmatrix}\boldsymbol{\lambda}_\perp & 0 \\ 0 & \lambda_\perp\end{bmatrix}^{-1}\frac{1}{\sqrt{2}}\begin{bmatrix}\tilde{\mathbf{V}}_\perp \\ \tilde{\mathbf{U}}_\perp\end{bmatrix}^T\begin{bmatrix}\mathbf{W}_1(0)^T \\ \mathbf{W}_2(0)\end{bmatrix}$$

$$= \begin{bmatrix}\mathbf{W}_1(0) & \mathbf{W}_2(0)^T\end{bmatrix}\begin{bmatrix}\tilde{\mathbf{V}}_\perp \\ \tilde{\mathbf{U}}_\perp\end{bmatrix}\begin{bmatrix}\boldsymbol{\lambda}_\perp^{-1}\tilde{\mathbf{V}}_\perp\tilde{\mathbf{V}}_\perp^T\mathbf{W}_1(0)^T \\ \boldsymbol{\lambda}_\perp^{-1}\tilde{\mathbf{U}}_\perp\tilde{\mathbf{U}}_\perp^T\mathbf{W}_2(0)\end{bmatrix}$$

$$= \begin{bmatrix}\mathbf{W}_1(0)\tilde{\mathbf{V}}_\perp\boldsymbol{\lambda}_\perp^{-1}\tilde{\mathbf{V}}_\perp^T\mathbf{W}_1(0)^T + \mathbf{W}_2(0)^T\tilde{\mathbf{U}}_\perp\boldsymbol{\lambda}_\perp^{-1}\tilde{\mathbf{U}}_\perp^T\mathbf{W}_2(0)\end{bmatrix} \quad (54)$$

Now

$$\frac{1}{2}\mathbf{Q}(0)^T\mathbf{O}\left(e^{2\boldsymbol{\Lambda}\frac{t}{\tau}} - \mathbf{I}\right)\boldsymbol{\Lambda}^{-1}\mathbf{O}^T = \frac{1}{4}\begin{bmatrix}\boldsymbol{B} - \boldsymbol{C}\end{bmatrix}\left(e^{\boldsymbol{\Lambda}\frac{t}{\tau}} - \mathbf{I}\right)\boldsymbol{\Lambda}^{-1}\begin{pmatrix}\boldsymbol{B}^T \\ -\boldsymbol{C}^T\end{pmatrix}$$

$$= \frac{1}{4}\left(\boldsymbol{B}\left(e^{2\tilde{\boldsymbol{S}}_\lambda \frac{t}{\tau}} - \mathbf{I}\right)(\tilde{\boldsymbol{S}}_\lambda)^{-1}\boldsymbol{B}^T - \boldsymbol{C}\left(e^{-2\tilde{\boldsymbol{S}}_\lambda \frac{t}{\tau}} - \mathbf{I}\right)(\tilde{\boldsymbol{S}}_\lambda)^{-1}\boldsymbol{C}^T\right)$$

$$(55)$$

Putting it all together

$$\boldsymbol{A}(t) = \mathbf{I} + \boldsymbol{B}\left(\frac{e^{2\tilde{\boldsymbol{S}}_\lambda \frac{t}{\tau}} - \mathbf{I}}{4\tilde{\boldsymbol{S}}_\lambda}\right)\boldsymbol{B}^T - \boldsymbol{C}\left(\frac{e^{-2\tilde{\boldsymbol{S}}_\lambda \frac{t}{\tau}} - \mathbf{I}}{4\tilde{\boldsymbol{S}}_\lambda}\right)\boldsymbol{C}^T + \mathbf{W}_2(0)^T\tilde{\mathbf{U}}_\perp\left(\frac{e^{\boldsymbol{\lambda}_\perp \frac{t}{\tau}} - \mathbf{I}}{\boldsymbol{\lambda}_\perp}\right)\tilde{\mathbf{U}}_\perp^T\mathbf{W}_2(0)$$

$$+ \mathbf{W}_1(0)\tilde{\mathbf{V}}_\perp\left(\frac{e^{\boldsymbol{\lambda}_\perp \frac{t}{\tau}} - \mathbf{I}}{\boldsymbol{\lambda}_\perp}\right)\tilde{\mathbf{V}}_\perp^T\mathbf{W}_1(0)^T \quad (56)$$

So, final form:

$$\mathbf{QQ}^T(t) =$$

$$
\left[\begin{pmatrix} \frac{1}{2}\tilde{V}(\tilde{G} - \tilde{H}\tilde{G})e^{\tilde{S}_\lambda \frac{t}{\tau}}B^T - \frac{1}{2}\tilde{V}(\tilde{G} + \tilde{H}\tilde{G})e^{-\tilde{S}_\lambda \frac{t}{\tau}}C^T + \tilde{V}_\perp e^{\lambda_\perp \frac{t}{\tau}}\tilde{V}_\perp^T \mathbf{W}_1(0)^T \\ \frac{1}{2}\tilde{U}(\tilde{G} + \tilde{H}\tilde{G})e^{\tilde{S}_\lambda \frac{t}{\tau}}B^T + \frac{1}{2}\tilde{U}(\tilde{G} - \tilde{H}\tilde{G})e^{-\tilde{S}_\lambda \frac{t}{\tau}}C^T + \tilde{U}_\perp e^{\lambda_\perp \frac{t}{\tau}}\tilde{U}_\perp^T \mathbf{W}_2(0) \end{pmatrix}\right]
$$

$$
\left[\mathbf{I} + \frac{1}{4}\left(B\left(\frac{e^{2\tilde{S}_\lambda \frac{t}{\tau}} - \mathbf{I}}{\tilde{S}_\lambda}\right)B^T - C\left(\frac{e^{-2\tilde{S}_\lambda \frac{t}{\tau}} - \mathbf{I}}{\tilde{S}_\lambda}\right)C^T\right)\right.
$$

$$
\left. + \mathbf{W}_2(0)^T\tilde{U}_\perp\left(\frac{e^{\lambda_\perp \frac{t}{\tau}} - \mathbf{I}}{\lambda_\perp}\right)\tilde{U}_\perp^T\mathbf{W}_2(0) + \mathbf{W}_1(0)\tilde{V}_\perp\left(\frac{e^{\lambda_\perp \frac{t}{\tau}} - \mathbf{I}}{\lambda_\perp}\right)\tilde{V}_\perp^T\mathbf{W}_1(0)^T\right]^{-1}
$$

$$
\left[\begin{pmatrix} \frac{1}{2}\tilde{V}(\tilde{G} - \tilde{H}\tilde{G})e^{\tilde{S}_\lambda \frac{t}{\tau}}B^T - \frac{1}{2}\tilde{V}(\tilde{G} + \tilde{H}\tilde{G})e^{-\tilde{S}_\lambda \frac{t}{\tau}}C^T + \tilde{V}_\perp e^{\lambda_\perp \frac{t}{\tau}}\tilde{V}_\perp^T \mathbf{W}_1(0)^T \\ \frac{1}{2}\tilde{U}(\tilde{G} + \tilde{H}\tilde{G})e^{\tilde{S}_\lambda \frac{t}{\tau}}B^T + \frac{1}{2}\tilde{U}(\tilde{G} - \tilde{H}\tilde{G})e^{-\tilde{S}_\lambda \frac{t}{\tau}}C^T + \tilde{U}_\perp e^{\lambda_\perp \frac{t}{\tau}}\tilde{U}_\perp^T \mathbf{W}_2(0) \end{pmatrix}\right]^T
$$

(57)

$\square$

## C.5 Stable solution Unequal-Input-Output

**Theorem C.5.** *Given the assumptions of Theorem C.4 further assuming that $\mathbf{B}$ is invertible and defining $e^{\lambda_\perp \frac{t}{\tau}} = \operatorname{sgn}(N_o - N_i)\frac{\lambda}{2}$, the temporal evolution of $\mathbf{QQ}^T$ is described as follows:*

$$\mathbf{QQ}^T(t) = \mathbf{Z}\left[e^{-\tilde{S}_\lambda \frac{t}{\tau}}\mathbf{B}^{-1}\mathbf{B}^{-T}e^{-\tilde{S}_\lambda \frac{t}{\tau}}\right. \tag{58}$$

$$
+ \left(\frac{\mathbf{I} - e^{-2\tilde{S}_\lambda \frac{t}{\tau}}}{4\tilde{S}_\lambda}\right) - e^{-\tilde{S}_\lambda \frac{t}{\tau}}\mathbf{B}^{-1}C\left(\frac{e^{-\tilde{S}_\lambda \frac{t}{\tau}} - \mathbf{I}}{4\tilde{S}_\lambda}\right)C^T\mathbf{B}^{-T}e^{-\tilde{S}_\lambda \frac{t}{\tau}}
$$

$$
- e^{-\tilde{S}_\lambda \frac{t}{\tau}}\mathbf{B}^{-1}\mathbf{W}_2(0)^T\tilde{U}_\perp\lambda_\perp^{-1}\tilde{U}_\perp^T\mathbf{W}_2(0)\mathbf{B}^{-T}e^{-\tilde{S}_\lambda \frac{t}{\tau}}
$$

$$
e^{-\tilde{S}_\lambda \frac{t}{\tau}}e^{\frac{\lambda_\perp}{2}\frac{t}{\tau}}\mathbf{B}^{-1}\mathbf{W}_2(0)^T\tilde{U}_\perp\lambda_\perp^{-1}\tilde{U}_\perp^T\mathbf{W}_2(0)\mathbf{B}^{-T}e^{-\tilde{S}_\lambda \frac{t}{\tau}}
$$

$$
+ e^{-\tilde{S}_\lambda \frac{t}{\tau}}e^{\frac{\lambda}{2}\frac{t}{\tau}}\mathbf{B}^{-1}\mathbf{W}_1(0)\tilde{V}_\perp\lambda_\perp^{-1}\tilde{V}_\perp^T\mathbf{W}_1(0)^T\mathbf{B}^{-T}e^{-\tilde{S}_\lambda \frac{t}{\tau}}
$$

$$
\left. - e^{-\tilde{S}_\lambda \frac{t}{\tau}}\mathbf{B}^{-1}\mathbf{W}_1(0)\tilde{V}_\perp\lambda_\perp^{-1}\tilde{V}_\perp^T\mathbf{W}_1(0)^T\mathbf{B}^{-T}e^{-\tilde{S}_\lambda \frac{t}{\tau}}\right]^{-1}\mathbf{Z}^T
$$

$$
\mathbf{Z} = \begin{pmatrix} \frac{1}{2}\tilde{V}\left[(\tilde{G} - \tilde{H}\tilde{G}) - (\tilde{G} + \tilde{H}\tilde{G})e^{-\tilde{S}_\lambda \frac{t}{\tau}}C^T\mathbf{B}^{-T}e^{-\tilde{S}_\lambda \frac{t}{\tau}}\right] + \tilde{V}_\perp\tilde{V}_\perp^T\mathbf{W}_1(0)\mathbf{B}^{-T}e^{\lambda_\perp \frac{t}{\tau}}e^{-\tilde{S}_\lambda \frac{t}{\tau}} \\ \frac{1}{2}\tilde{U}\left[(\tilde{G} + \tilde{H}\tilde{G}) + (\tilde{G} - \tilde{H}\tilde{G})e^{-\tilde{S}_\lambda \frac{t}{\tau}}C^T\mathbf{B}^{-T}e^{-\tilde{S}_\lambda \frac{t}{\tau}}\right] + \tilde{U}_\perp\tilde{U}_\perp^T\mathbf{W}_2(0)^T\mathbf{B}^{-T}e^{\lambda_\perp \frac{t}{\tau}}e^{-\tilde{S}_\lambda \frac{t}{\tau}} \end{pmatrix}
$$

(59)

*Proof.* We start from

$$\mathbf{QQ}^T(t) =$$

$$
\left[\begin{pmatrix} \frac{1}{2}\tilde{V}(\tilde{G} - \tilde{H}\tilde{G})e^{\tilde{S}_\lambda \frac{t}{\tau}}B^T - \frac{1}{2}\tilde{V}(\tilde{G} + \tilde{H}\tilde{G})e^{-\tilde{S}_\lambda \frac{t}{\tau}}C^T + \tilde{V}_\perp e^{\lambda_\perp \frac{t}{\tau}}\tilde{V}_\perp^T \mathbf{W}_1(0)^T \\ \frac{1}{2}\tilde{U}(\tilde{G} + \tilde{H}\tilde{G})e^{\tilde{S}_\lambda \frac{t}{\tau}}B^T + \frac{1}{2}\tilde{U}(\tilde{G} - \tilde{H}\tilde{G})e^{-\tilde{S}_\lambda \frac{t}{\tau}}C^T + \tilde{U}_\perp e^{\lambda_\perp \frac{t}{\tau}}\tilde{U}_\perp^T \mathbf{W}_2(0) \end{pmatrix}\right]
$$

$$
\left[\mathbf{I} + \frac{1}{4}\left(B\left(\frac{e^{2\tilde{S}_\lambda \frac{t}{\tau}} - \mathbf{I}}{\tilde{S}_\lambda}\right)B^T - C\left(\frac{e^{-2\tilde{S}_\lambda \frac{t}{\tau}} - \mathbf{I}}{\tilde{S}_\lambda}\right)C^T\right)\right.
$$

$$
\left. + \mathbf{W}_2(0)^T\tilde{U}_\perp\left(\frac{e^{\lambda_\perp \frac{t}{\tau}} - \mathbf{I}}{\lambda_\perp}\right)\tilde{U}_\perp^T\mathbf{W}_2(0) + \mathbf{W}_1(0)\tilde{V}_\perp\left(\frac{e^{\lambda_\perp \frac{t}{\tau}} - \mathbf{I}}{\lambda_\perp}\right)\tilde{V}_\perp^T\mathbf{W}_1(0)^T\right]^{-1}
$$

$$
\left[\begin{pmatrix} \frac{1}{2}\tilde{V}(\tilde{G} - \tilde{H}\tilde{G})e^{\tilde{S}_\lambda \frac{t}{\tau}}B^T - \frac{1}{2}\tilde{V}(\tilde{G} + \tilde{H}\tilde{G})e^{-\tilde{S}_\lambda \frac{t}{\tau}}C^T + \tilde{V}_\perp e^{\lambda_\perp \frac{t}{\tau}}\tilde{V}_\perp^T \mathbf{W}_1(0)^T \\ \frac{1}{2}\tilde{U}(\tilde{G} + \tilde{H}\tilde{G})e^{\tilde{S}_\lambda \frac{t}{\tau}}B^T + \frac{1}{2}\tilde{U}(\tilde{G} - \tilde{H}\tilde{G})e^{-\tilde{S}_\lambda \frac{t}{\tau}}C^T + \tilde{U}_\perp e^{\lambda_\perp \frac{t}{\tau}}\tilde{U}_\perp^T \mathbf{W}_2(0) \end{pmatrix}\right]^T
$$

(60)

We extract $\boldsymbol{B}^{-T}e^{-\tilde{\boldsymbol{S}}_\lambda \frac{t}{\tau}}$ from all terms as exemplified bellow

$$\boldsymbol{O}e^{\boldsymbol{\Lambda}t/\tau}\boldsymbol{O}^T\boldsymbol{Q}(0) = \frac{1}{2}\begin{pmatrix}\tilde{\boldsymbol{V}}\left[(\tilde{\boldsymbol{G}}-\tilde{\boldsymbol{H}}\tilde{\boldsymbol{G}})-(\tilde{\boldsymbol{G}}+\tilde{\boldsymbol{H}}\tilde{\boldsymbol{G}})e^{-\tilde{\boldsymbol{S}}_\lambda\frac{t}{\tau}}\boldsymbol{C}^T\boldsymbol{B}^{-T}e^{-\tilde{\boldsymbol{S}}_\lambda\frac{t}{\tau}}\right]\\ \tilde{\boldsymbol{U}}\left[(\tilde{\boldsymbol{G}}+\tilde{\boldsymbol{H}}\tilde{\boldsymbol{G}})+(\tilde{\boldsymbol{G}}-\tilde{\boldsymbol{H}}\tilde{\boldsymbol{G}})e^{-\tilde{\boldsymbol{S}}_\lambda\frac{t}{\tau}}\boldsymbol{C}^T\boldsymbol{B}^{-T}e^{-\tilde{\boldsymbol{S}}_\lambda\frac{t}{\tau}}\right]\end{pmatrix}\boldsymbol{B}^T e^{\tilde{\boldsymbol{S}}_\lambda\frac{t}{\tau}}\quad(61)$$

and rewrite the dynamis as

$$\mathbf{QQ}^T(t) =$$
$$\left[\begin{pmatrix}\frac{1}{2}\tilde{\boldsymbol{V}}(\tilde{\boldsymbol{G}}-\tilde{\boldsymbol{H}}\tilde{\boldsymbol{G}})-\frac{1}{2}\tilde{\boldsymbol{V}}(\tilde{\boldsymbol{G}}+\tilde{\boldsymbol{H}}\tilde{\boldsymbol{G}})e^{-\tilde{\boldsymbol{S}}_\lambda\frac{t}{\tau}}\boldsymbol{C}^T\boldsymbol{B}^{-T}e^{-\tilde{\boldsymbol{S}}_\lambda\frac{t}{\tau}}+\tilde{\mathbf{V}}_\perp e^{\boldsymbol{\lambda}_\perp\frac{t}{\tau}}\tilde{\mathbf{V}}_\perp^T\mathbf{W}_1(0)^T\boldsymbol{B}^{-T}e^{-\tilde{\boldsymbol{S}}_\lambda\frac{t}{\tau}}\\ \frac{1}{2}\tilde{\boldsymbol{U}}(\tilde{\boldsymbol{G}}+\tilde{\boldsymbol{H}}\tilde{\boldsymbol{G}})+\frac{1}{2}\tilde{\boldsymbol{U}}(\tilde{\boldsymbol{G}}-\tilde{\boldsymbol{H}}\tilde{\boldsymbol{G}})e^{-\tilde{\boldsymbol{S}}_\lambda\frac{t}{\tau}}\boldsymbol{C}^T\boldsymbol{B}^{-T}e^{-\tilde{\boldsymbol{S}}_\lambda\frac{t}{\tau}}+\tilde{\mathbf{U}}_\perp e^{\boldsymbol{\lambda}_\perp\frac{t}{\tau}}\tilde{\mathbf{U}}_\perp^T\mathbf{W}_2(0)\boldsymbol{B}^{-T}e^{-\tilde{\boldsymbol{S}}_\lambda\frac{t}{\tau}}\end{pmatrix}\right]$$
$$\left[e^{-\tilde{\boldsymbol{S}}_\lambda\frac{t}{\tau}}\boldsymbol{B}^{-1}\boldsymbol{B}^{-T}e^{-\tilde{\boldsymbol{S}}_\lambda\frac{t}{\tau}}+\frac{1}{4}\left(\left(\frac{\mathbf{I}-e^{-2\tilde{\boldsymbol{S}}_\lambda\frac{t}{\tau}}}{\tilde{\boldsymbol{S}}_\lambda}\right)-e^{-\tilde{\boldsymbol{S}}_\lambda\frac{t}{\tau}}\boldsymbol{B}^{-1}\boldsymbol{C}\left(\frac{e^{-2\tilde{\boldsymbol{S}}_\lambda\frac{t}{\tau}}-\mathbf{I}}{\tilde{\boldsymbol{S}}_\lambda}\right)\boldsymbol{C}^T\boldsymbol{B}^{-T}e^{-\tilde{\boldsymbol{S}}_\lambda\frac{t}{\tau}}\right)\right.$$
$$+e^{-\tilde{\boldsymbol{S}}_\lambda\frac{t}{\tau}}\boldsymbol{B}^{-1}\mathbf{W}_2(0)^T\tilde{\mathbf{U}}_\perp\left(\frac{e^{\boldsymbol{\lambda}_\perp\frac{t}{\tau}}-\mathbf{I}}{\boldsymbol{\lambda}_\perp}\right)\tilde{\mathbf{U}}_\perp^T\mathbf{W}_2(0)\boldsymbol{B}^{-T}e^{-\tilde{\boldsymbol{S}}_\lambda\frac{t}{\tau}}$$
$$\left.+e^{-\tilde{\boldsymbol{S}}_\lambda\frac{t}{\tau}}\boldsymbol{B}^{-1}\mathbf{W}_1(0)\tilde{\mathbf{V}}_\perp\left(\frac{e^{\boldsymbol{\lambda}_\perp\frac{t}{\tau}}-\mathbf{I}}{\boldsymbol{\lambda}_\perp}\right)\tilde{\mathbf{V}}_\perp^T\mathbf{W}_1(0)^T\boldsymbol{B}^{-T}e^{-\tilde{\boldsymbol{S}}_\lambda\frac{t}{\tau}}\right]^{-1}$$
$$\left[\begin{pmatrix}\frac{1}{2}\tilde{\boldsymbol{V}}(\tilde{\boldsymbol{G}}-\tilde{\boldsymbol{H}}\tilde{\boldsymbol{G}})-\frac{1}{2}\tilde{\boldsymbol{V}}(\tilde{\boldsymbol{G}}+\tilde{\boldsymbol{H}}\tilde{\boldsymbol{G}})e^{-\tilde{\boldsymbol{S}}_\lambda\frac{t}{\tau}}\boldsymbol{C}^T\boldsymbol{B}^{-T}e^{-\tilde{\boldsymbol{S}}_\lambda\frac{t}{\tau}}+\tilde{\mathbf{V}}_\perp e^{\boldsymbol{\lambda}_\perp\frac{t}{\tau}}\tilde{\mathbf{V}}_\perp^T\mathbf{W}_1(0)^T\boldsymbol{B}^{-T}e^{-\tilde{\boldsymbol{S}}_\lambda\frac{t}{\tau}}\\ \frac{1}{2}\tilde{\boldsymbol{U}}(\tilde{\boldsymbol{G}}+\tilde{\boldsymbol{H}}\tilde{\boldsymbol{G}})+\frac{1}{2}\tilde{\boldsymbol{U}}(\tilde{\boldsymbol{G}}-\tilde{\boldsymbol{H}}\tilde{\boldsymbol{G}})e^{-\tilde{\boldsymbol{S}}_\lambda\frac{t}{\tau}}\boldsymbol{C}^T\boldsymbol{B}^{-T}e^{-\tilde{\boldsymbol{S}}_\lambda\frac{t}{\tau}}+\tilde{\mathbf{U}}_\perp e^{\boldsymbol{\lambda}_\perp\frac{t}{\tau}}\tilde{\mathbf{U}}_\perp^T\mathbf{W}_2(0)\boldsymbol{B}^{-T}e^{-\tilde{\boldsymbol{S}}_\lambda\frac{t}{\tau}}\end{pmatrix}\right]^T$$
$$(62)$$

$$\mathbf{QQ}^T(t) =$$
$$\begin{pmatrix}\frac{1}{2}\tilde{\boldsymbol{V}}\left[(\tilde{\boldsymbol{G}}-\tilde{\boldsymbol{H}}\tilde{\boldsymbol{G}})-(\tilde{\boldsymbol{G}}+\tilde{\boldsymbol{H}}\tilde{\boldsymbol{G}})e^{-\tilde{\boldsymbol{S}}_\lambda\frac{t}{\tau}}\boldsymbol{C}^T\boldsymbol{B}^{-T}e^{-\tilde{\boldsymbol{S}}_\lambda\frac{t}{\tau}}\right]+\tilde{\mathbf{V}}_\perp\tilde{\mathbf{V}}_\perp^T\mathbf{W}_1(0)\boldsymbol{B}^{-T}e^{\boldsymbol{\lambda}_\perp\frac{t}{\tau}}e^{-\tilde{\boldsymbol{S}}_\lambda\frac{t}{\tau}}\\ \frac{1}{2}\tilde{\boldsymbol{U}}\left[(\tilde{\boldsymbol{G}}+\tilde{\boldsymbol{H}}\tilde{\boldsymbol{G}})+(\tilde{\boldsymbol{G}}-\tilde{\boldsymbol{H}}\tilde{\boldsymbol{G}})e^{-\tilde{\boldsymbol{S}}_\lambda\frac{t}{\tau}}\boldsymbol{C}^T\boldsymbol{B}^{-T}e^{-\tilde{\boldsymbol{S}}_\lambda\frac{t}{\tau}}\right]+\tilde{\mathbf{U}}_\perp\tilde{\mathbf{U}}_\perp^T\mathbf{W}_2(0)^T\boldsymbol{B}^{-T}e^{\boldsymbol{\lambda}_\perp\frac{t}{\tau}}e^{-\tilde{\boldsymbol{S}}_\lambda\frac{t}{\tau}}\end{pmatrix}$$
$$\left[e^{-\tilde{\boldsymbol{S}}_\lambda\frac{t}{\tau}}\boldsymbol{B}^{-1}\boldsymbol{B}^{-T}e^{-\tilde{\boldsymbol{S}}_\lambda\frac{t}{\tau}}\right.$$
$$+\left(\frac{\mathbf{I}-e^{-2\tilde{\boldsymbol{S}}_\lambda\frac{t}{\tau}}}{4\tilde{\boldsymbol{S}}_\lambda}\right)-e^{-\tilde{\boldsymbol{S}}_\lambda\frac{t}{\tau}}\boldsymbol{B}^{-1}\boldsymbol{C}\left(\frac{e^{-\tilde{\boldsymbol{S}}_\lambda\frac{t}{\tau}}-\mathbf{I}}{4\tilde{\boldsymbol{S}}_\lambda}\right)\boldsymbol{C}^T\boldsymbol{B}^{-T}e^{-\tilde{\boldsymbol{S}}_\lambda\frac{t}{\tau}}$$
$$-e^{-\tilde{\boldsymbol{S}}_\lambda\frac{t}{\tau}}\boldsymbol{B}^{-1}\mathbf{W}_2(0)^T\tilde{\mathbf{U}}_\perp\boldsymbol{\lambda}_\perp^{-1}\tilde{\mathbf{U}}_\perp^T\mathbf{W}_2(0)\mathbf{B}^{-T}e^{-\tilde{\boldsymbol{S}}_\lambda\frac{t}{\tau}}$$
$$e^{-\tilde{\boldsymbol{S}}_\lambda\frac{t}{\tau}}e^{\frac{\boldsymbol{\lambda}_\perp}{2}\frac{t}{\tau}}\boldsymbol{B}^{-1}\mathbf{W}_2(0)^T\tilde{\mathbf{U}}_\perp\boldsymbol{\lambda}_\perp^{-1}\tilde{\mathbf{U}}_\perp^T\mathbf{W}_2(0)\mathbf{B}^{-T}e^{-\tilde{\boldsymbol{S}}_\lambda\frac{t}{\tau}}$$
$$+e^{-\tilde{\boldsymbol{S}}_\lambda\frac{t}{\tau}}e^{\frac{\boldsymbol{\lambda}}{2}\frac{t}{\tau}}\boldsymbol{B}^{-1}\mathbf{W}_1(0)\tilde{\mathbf{V}}_\perp\boldsymbol{\lambda}_\perp^{-1}\tilde{\mathbf{V}}_\perp^T\mathbf{W}_1(0)^T\mathbf{B}^{-T}e^{-\tilde{\boldsymbol{S}}_\lambda\frac{t}{\tau}}$$
$$\left.-e^{-\tilde{\boldsymbol{S}}_\lambda\frac{t}{\tau}}\boldsymbol{B}^{-1}\mathbf{W}_1(0)\tilde{\mathbf{V}}_\perp\boldsymbol{\lambda}_\perp^{-1}\tilde{\mathbf{V}}_\perp^T\mathbf{W}_1(0)^T\mathbf{B}^{-T}e^{-\tilde{\boldsymbol{S}}_\lambda\frac{t}{\tau}}\right]^{-1}$$
$$\begin{pmatrix}\tilde{\boldsymbol{V}}\left[(\tilde{\boldsymbol{G}}-\tilde{\boldsymbol{H}}\tilde{\boldsymbol{G}})-(\tilde{\boldsymbol{G}}+\tilde{\boldsymbol{H}}\tilde{\boldsymbol{G}})e^{-\tilde{\boldsymbol{S}}_\lambda\frac{t}{\tau}}\boldsymbol{C}^T\boldsymbol{B}^{-T}e^{-\tilde{\boldsymbol{S}}_\lambda\frac{t}{\tau}}\right]+\tilde{\mathbf{V}}_\perp\tilde{\mathbf{V}}_\perp^T\mathbf{W}_1(0)\boldsymbol{B}^{-T}e^{\boldsymbol{\lambda}_\perp\frac{t}{\tau}}e^{-\tilde{\boldsymbol{S}}_\lambda\frac{t}{\tau}}\\ \tilde{\boldsymbol{U}}\left[(\tilde{\boldsymbol{G}}+\tilde{\boldsymbol{H}}\tilde{\boldsymbol{G}})+(\tilde{\boldsymbol{G}}-\tilde{\boldsymbol{H}}\tilde{\boldsymbol{G}})e^{-\tilde{\boldsymbol{S}}_\lambda\frac{t}{\tau}}\boldsymbol{C}^T\boldsymbol{B}^{-T}e^{-\tilde{\boldsymbol{S}}_\lambda\frac{t}{\tau}}\right]+\tilde{\mathbf{U}}_\perp\tilde{\mathbf{U}}_\perp^T\mathbf{W}_2(0)^T\boldsymbol{B}^{-T}e^{\boldsymbol{\lambda}_\perp\frac{t}{\tau}}e^{-\tilde{\boldsymbol{S}}_\lambda\frac{t}{\tau}}\end{pmatrix}^T$$
$$(63)$$

where $e^{\boldsymbol{\lambda}_\perp\frac{t}{\tau}}=\mathrm{sgn}(N_o-N_i)\frac{\lambda}{2}$ is a scalar $\qquad\square$

### C.5.1 Proof Exact learning dynamics with prior knowledge unequal dimension

We follow a similar derivation presented in Braun et al. (2022) and start with the following equation

$$\mathbf{QQ}^T(t) = \underbrace{\left[\mathbf{O}e^{\mathbf{\Lambda}\frac{t}{\tau}}\mathbf{O}^T + 2\mathbf{M}e^{\boldsymbol{\lambda}_\perp \frac{t}{\tau}}\mathbf{M}^T\right]\mathbf{Q}(0)}_{\mathbf{L}}$$

$$\underbrace{\left[\mathbf{I} + \frac{1}{2}\mathbf{Q}(0)^T\left(\mathbf{O}\left(e^{2\mathbf{\Lambda}\frac{t}{\tau}} - \mathbf{I}\right)\mathbf{\Lambda}^{-1}\mathbf{O}^T + \mathbf{M}(e^{\boldsymbol{\lambda}_\perp \frac{t}{\tau}} - \mathbf{I})\boldsymbol{\lambda}_\perp^{-1}\mathbf{M}^T\right)\mathbf{Q}(0)\right]^{-1}}_{\mathbf{C}^{-1}} \quad (64)$$

$$\underbrace{\mathbf{Q}(0)^T\left[\mathbf{O}e^{\mathbf{\Lambda}\frac{t}{\tau}}\mathbf{O}^T + 2\mathbf{M}e^{\boldsymbol{\lambda}_\perp \frac{t}{\tau}}\mathbf{M}^T\right]}_{\mathbf{R}}$$

$$=\mathbf{L}\mathbf{C}^{-1}\mathbf{R}, \quad (65)$$

Substituting our solution into the matrix Riccati equation then yields

$$\tau\frac{d}{dt}\mathbf{QQ}^T = \mathbf{F}\mathbf{QQ}^T + \mathbf{QQ}^T\mathbf{F} - (\mathbf{QQ}^T)^2 \quad (66)$$

$$\Rightarrow \tau\frac{d}{dt}\mathbf{L}\mathbf{C}^{-1}\mathbf{R} \stackrel{?}{=} \mathbf{F}\mathbf{L}\mathbf{C}^{-1}\mathbf{R} + \mathbf{L}\mathbf{C}^{-1}\mathbf{R}\mathbf{F} - \mathbf{L}\mathbf{C}^{-1}\mathbf{R}\mathbf{L}\mathbf{C}^{-1}\mathbf{R}. \quad (67)$$

Using the chain rule $\partial(\mathbf{AB}) = (\partial\mathbf{A})\mathbf{B} + \mathbf{A}(\partial\mathbf{B})$ and the identities

$$\frac{d}{dt}(\mathbf{A}^{-1}) = \mathbf{A}^{-1}(\frac{d}{dt}\mathbf{A})\mathbf{A}^{-1} \quad \text{and} \quad \frac{d}{dt}(e^{t\mathbf{A}}) = \mathbf{A}e^{t\mathbf{A}} = e^{t\mathbf{A}}\mathbf{A} \quad (68)$$

$$\tau\frac{d}{dt}\mathbf{QQ}^T = \tau\frac{d}{dt}\mathbf{L}\mathbf{C}^{-1}\mathbf{R} \quad (69)$$

$$= \tau\left(\frac{d}{dt}\mathbf{L}\right)\mathbf{C}^{-1}\mathbf{R} + \tau\mathbf{L}\left(\frac{d}{dt}C^{-1}\mathbf{R}\right) \quad (70)$$

$$= \tau\left(\frac{d}{dt}\mathbf{L}\right)\mathbf{C}^{-1}\mathbf{R} + \tau\mathbf{L}\mathbf{C}^{-1}\left(\frac{d}{dt}\mathbf{R}\right) + \tau\mathbf{L}\left(\frac{d}{dt}\mathbf{C}^{-1}\right)\mathbf{R}, \quad (71)$$

Next, we note that

$$\mathbf{O} = \frac{1}{\sqrt{2}}\begin{pmatrix} \tilde{\mathbf{V}}(\tilde{\mathbf{G}} - \tilde{\mathbf{H}}\tilde{\mathbf{G}}) & \tilde{\mathbf{V}}(\tilde{\mathbf{G}} + \tilde{\mathbf{H}}\tilde{\mathbf{G}}) \\ \tilde{\mathbf{U}}(\tilde{\mathbf{G}} + \tilde{\mathbf{H}}\tilde{\mathbf{G}}) & -\tilde{\mathbf{U}}(\tilde{\mathbf{G}} - \tilde{\mathbf{H}}\tilde{\mathbf{G}}) \end{pmatrix}^T \quad (72)$$

$$\mathbf{O}^T\mathbf{O} = \frac{1}{\sqrt{2}}\begin{pmatrix} \tilde{\mathbf{V}}(\tilde{\mathbf{G}} - \tilde{\mathbf{H}}\tilde{\mathbf{G}}) & \tilde{\mathbf{V}}(\tilde{\mathbf{G}} + \tilde{\mathbf{H}}\tilde{\mathbf{G}}) \\ \tilde{\mathbf{U}}(\tilde{\mathbf{G}} + \tilde{\mathbf{H}}\tilde{\mathbf{G}}) & -\tilde{\mathbf{U}}(\tilde{\mathbf{G}} - \tilde{\mathbf{H}}\tilde{\mathbf{G}}) \end{pmatrix}^T \frac{1}{\sqrt{2}}\begin{pmatrix} \tilde{\mathbf{V}}(\tilde{\mathbf{G}} - \tilde{\mathbf{H}}\tilde{\mathbf{G}}) & \tilde{\mathbf{V}}(\tilde{\mathbf{G}} + \tilde{\mathbf{H}}\tilde{\mathbf{G}}) \\ \tilde{\mathbf{U}}(\tilde{\mathbf{G}} + \tilde{\mathbf{H}}\tilde{\mathbf{G}}) & -\tilde{\mathbf{U}}(\tilde{\mathbf{G}} - \tilde{\mathbf{H}}\tilde{\mathbf{G}}) \end{pmatrix}$$

$$\quad (73)$$

$$= \mathbf{I} \quad (74)$$

$$\mathbf{O}^T\mathbf{M} = \frac{1}{\sqrt{2}}\begin{bmatrix} \tilde{\mathbf{V}}(\tilde{\mathbf{G}} - \tilde{\mathbf{H}}\tilde{\mathbf{G}}) & \tilde{\mathbf{V}}(\tilde{\mathbf{G}} + \tilde{\mathbf{H}}\tilde{\mathbf{G}}) \\ \tilde{\mathbf{U}}(\tilde{\mathbf{G}} + \tilde{\mathbf{H}}\tilde{\mathbf{G}}) & -\tilde{\mathbf{U}}(\tilde{\mathbf{G}} - \tilde{\mathbf{H}}\tilde{\mathbf{G}}) \end{bmatrix}\frac{1}{\sqrt{2}}\begin{bmatrix} \tilde{\mathbf{V}}_\perp \\ \tilde{\mathbf{U}}_\perp \end{bmatrix} \quad (75)$$

$$= \frac{1}{2}\begin{bmatrix} (\tilde{\mathbf{G}} - \tilde{\mathbf{H}}\tilde{\mathbf{G}})^T\tilde{\mathbf{V}}^T\tilde{\mathbf{V}}_\perp + (\tilde{\mathbf{G}} + \tilde{\mathbf{H}}\tilde{\mathbf{G}})^T\tilde{\mathbf{U}}^T\tilde{\mathbf{U}}_\perp \\ (\tilde{\mathbf{G}} + \tilde{\mathbf{H}}\tilde{\mathbf{G}})^T\tilde{\mathbf{V}}^T\tilde{\mathbf{V}}_\perp - (\tilde{\mathbf{G}} - \tilde{\mathbf{H}}\tilde{\mathbf{G}})^T\tilde{\mathbf{U}}^T\tilde{\mathbf{U}}_\perp \end{bmatrix} \quad (76)$$

$$= \mathbf{0} \quad (77)$$

and

$$\mathbf{M}^T\mathbf{O} = \frac{1}{\sqrt{2}} \begin{bmatrix} \tilde{\mathbf{V}}_\perp^T & \tilde{\mathbf{U}}_\perp^T \end{bmatrix} \begin{pmatrix} \tilde{V}(\tilde{G}-\tilde{H}\tilde{G}) & \tilde{V}(\tilde{G}+\tilde{H}\tilde{G}) \\ \tilde{U}(\tilde{G}+\tilde{H}\tilde{G}) & -\tilde{U}(\tilde{G}-\tilde{H}\tilde{G}) \end{pmatrix} \tag{78}$$

$$= \frac{1}{2} \begin{bmatrix} \tilde{\mathbf{V}}_\perp^T \tilde{V}(\tilde{G}-\tilde{H}\tilde{G}) + \tilde{\mathbf{U}}_\perp^T \tilde{U}(\tilde{G}+\tilde{H}\tilde{G}) \\ \tilde{\mathbf{V}}_\perp^T \tilde{V}(\tilde{G}+\tilde{H}\tilde{G}) - \tilde{\mathbf{U}}_\perp^T \tilde{U}(\tilde{G}-\tilde{H}\tilde{G}) \end{bmatrix} \tag{79}$$

$$= \mathbf{0}. \tag{80}$$

we get

$$\tau \frac{d}{dt}\mathbf{Q}\mathbf{Q}^T = \tau \frac{d}{dt}\left(\mathbf{L}\mathbf{C}^{-1}\mathbf{R}\right) \tag{81}$$

$$= \tau\left(\frac{d}{dt}\mathbf{L}\right)\mathbf{C}^{-1}\mathbf{R} + \tau\mathbf{L}\left(\frac{d}{dt}C^{-1}\mathbf{R}\right) \tag{82}$$

$$= \tau\left(\frac{d}{dt}\mathbf{L}\right)\mathbf{C}^{-1}\mathbf{R} + \tau\mathbf{L}\mathbf{C}^{-1}\left(\frac{d}{dt}\mathbf{R}\right) + \tau\mathbf{L}\left(\frac{d}{dt}\mathbf{C}^{-1}\right)\mathbf{R}, \tag{83}$$

with

$$\tau\left(\frac{d}{dt}\mathbf{L}\right)\mathbf{C}^{-1}\mathbf{R} = \tau\left(\mathbf{O}\frac{1}{\tau}\mathbf{\Lambda}e^{\mathbf{\Lambda}\frac{t}{\tau}}\mathbf{O}^T + 2\mathbf{M}\frac{\lambda_\perp\mathbf{I}}{2\tau}e^{\lambda_\perp\frac{t}{\tau}}\mathbf{M}^T\right)\mathbf{Q}(0)\mathbf{C}^{-1}\mathbf{R} \tag{84}$$

$$= \left(\mathbf{O}\mathbf{\Lambda}e^{\mathbf{\Lambda}\frac{t}{\tau}}\mathbf{O}^T + \mathbf{M}\lambda_\perp\mathbf{I}e^{\lambda_\perp\frac{t}{\tau}}\mathbf{M}^T\right)\mathbf{Q}(0)\mathbf{C}^{-1}\mathbf{R} \tag{85}$$

$$= \left(\mathbf{O}\lambda_\perp\mathbf{O}^T + 2\mathbf{M}\lambda_\perp\mathbf{M}^T\right)\left(\mathbf{O}e^{\mathbf{\Lambda}\frac{t}{\tau}}\mathbf{O}^T + 2\mathbf{M}e^{\lambda_\perp\frac{t}{\tau}}\mathbf{M}^T\right)\mathbf{Q}(0)\mathbf{C}^{-1}\mathbf{R} \tag{86}$$

$$= \mathbf{F}\mathbf{L}\mathbf{C}^{-1}\mathbf{R}, \tag{87}$$

$$\tau\mathbf{L}\mathbf{C}^{-1}\left(\frac{d}{dt}\mathbf{R}\right) = \tau\mathbf{L}\mathbf{C}^{-1}\mathbf{Q}(0)^T\left(\mathbf{O}\frac{1}{\tau}e^{\mathbf{\Lambda}\frac{t}{\tau}}\mathbf{\Lambda}\mathbf{O}^T + 2\mathbf{M}e^{\lambda_\perp\frac{t}{\tau}}\frac{\lambda_\perp\mathbf{I}}{2\tau}\mathbf{M}^T\right) \tag{88}$$

$$= \mathbf{L}\mathbf{C}^{-1}\mathbf{Q}(0)^T\left(\mathbf{O}\frac{1}{\tau}e^{\mathbf{\Lambda}\frac{t}{\tau}}\mathbf{\Lambda}\mathbf{O}^T + 2\mathbf{M}e^{\lambda_\perp\frac{t}{\tau}}\frac{\lambda_\perp\mathbf{I}}{2\tau}\mathbf{M}^T\right) \tag{89}$$

$$= \mathbf{L}\mathbf{C}^{-1}\mathbf{R}\mathbf{F} \tag{90}$$

and

$$\tau\mathbf{L}\left(\frac{d}{dt}\mathbf{C}^{-1}\right)\mathbf{R} = -\tau\mathbf{L}\mathbf{C}^{-1}\left(\frac{d}{dt}\mathbf{C}\right)\mathbf{C}^{-1}\mathbf{R} \tag{91}$$

$$= -\mathbf{L}\mathbf{C}^{-1}\Big[\tau\frac{1}{2}\mathbf{Q}(0)^T\mathbf{O}2\frac{1}{\tau}e^{2\mathbf{\Lambda}\frac{t}{\tau}}\mathbf{\Lambda}\mathbf{\Lambda}^{-1}\mathbf{O}^T\mathbf{Q}(0) \tag{92}$$

$$\qquad\qquad + \tau\frac{1}{2}\mathbf{Q}(0)^T 4\frac{1}{\tau}\mathbf{M}e^{\lambda_\perp\frac{t}{\tau}}\lambda_\perp\left(\lambda_\perp\right)^{-1}\mathbf{M}^T\mathbf{Q}(0)\Big]\mathbf{C}^{-1}\mathbf{R}$$

$$= -\mathbf{L}\mathbf{C}^{-1}\Big[\mathbf{Q}(0)^T\mathbf{O}e^{2\mathbf{\Lambda}\frac{t}{\tau}}\mathbf{O}^T\mathbf{Q}(0) + 2\mathbf{Q}(0)^T\mathbf{M}e^{\lambda_\perp\frac{t}{\tau}}\mathbf{M}^T\mathbf{Q}(0)\Big]\mathbf{C}^{-1}\mathbf{R} \tag{93}$$

$$= -\mathbf{L}\mathbf{C}^{-1}\Big[\mathbf{Q}(0)^T\mathbf{O}e^{\mathbf{\Lambda}\frac{t}{\tau}}\mathbf{O}^T\mathbf{O}e^{\mathbf{\Lambda}\frac{t}{\tau}}\mathbf{O}^T\mathbf{Q}(0)$$

$$\qquad\qquad + 2\mathbf{Q}(0)^T\mathbf{O}e^{\mathbf{\Lambda}\frac{t}{\tau}}\underbrace{\mathbf{O}^T\mathbf{M}}_{\mathbf{0}}e^{\lambda_\perp\frac{t}{\tau}}\mathbf{M}^T\mathbf{Q}(0) \tag{94}$$

$$\qquad\qquad + 2\mathbf{Q}(0)^T\mathbf{M}e^{\lambda_\perp\frac{t}{\tau}}\underbrace{\mathbf{M}^T\mathbf{O}}_{\mathbf{0}}e^{\mathbf{\Lambda}\frac{t}{\tau}}\mathbf{O}^T\mathbf{Q}(0)$$

$$\qquad\qquad + 4\mathbf{Q}(0)^T\mathbf{M}e^{\lambda_\perp\frac{t}{\tau}}\mathbf{M}^T\mathbf{M}e^{\lambda_\perp\frac{t}{\tau}}\mathbf{M}^T\mathbf{Q}(0)\Big]\mathbf{C}^{-1}\mathbf{R}$$

$$= -\mathbf{L}\mathbf{C}^{-1}\mathbf{R}\mathbf{L}\mathbf{C}^{-1}\mathbf{R}. \tag{95}$$

Finally, substituting equations 84, 88 and 91 into the left hand side of equation 67 proves equality.
□

## D Rich-Lazy

### D.1 Dynamics of the Singular Values

**Theorem D.1.** *Under the assumptions of Theorem C.4 and with a task-aligned initialization given by $\boldsymbol{W}_1(0) = \boldsymbol{R}\boldsymbol{S}_1\tilde{\boldsymbol{V}}^T$ and $\boldsymbol{W}_2(0) = \tilde{\boldsymbol{U}}\boldsymbol{S}_2\boldsymbol{R}^T$, where $\boldsymbol{R} \in \mathbb{R}^{N_h \times N_h}$ is an orthonormal matrix, then the network function is given by the expression $\boldsymbol{W}_2\boldsymbol{W}_1(t) = \tilde{\boldsymbol{U}}\boldsymbol{S}(t)\tilde{\boldsymbol{V}}^T$ where $\boldsymbol{S}(t) \in \mathbb{R}^{N_h \times N_h}$ is a diagonal matrix of singular values with elements $s_\alpha(t)$ that evolve according to the equation,*

$$s_\alpha(t) = s_\alpha(0) + \gamma_\alpha(t; \lambda)\left(\tilde{s}_\alpha - s_\alpha(0)\right), \tag{96}$$

*where $\tilde{s}_\alpha$ is the $\alpha$ singular value of $\tilde{\boldsymbol{S}}$ and $\gamma_\alpha(t; \lambda)$ is a $\lambda$-dependent monotonic transition function for each singular value that increases from $\gamma_\alpha(0; \lambda) = 0$ to $\lim_{t\to\infty} \gamma_\alpha(t; \lambda) = 1$ defined as*

$$\gamma_\alpha(t; \lambda) = \frac{\tilde{s}_{\lambda,\alpha}s_{\lambda,\alpha}\sinh\left(2\tilde{s}_{\lambda,\alpha}\frac{t}{\tau}\right) + \left(\tilde{s}_\alpha s_\alpha + \frac{\lambda^2}{4}\right)\cosh\left(2\tilde{s}_{\lambda,\alpha}\frac{t}{\tau}\right) - \left(\tilde{s}_\alpha s_\alpha + \frac{\lambda^2}{4}\right)}{\tilde{s}_{\lambda,\alpha}s_{\lambda,\alpha}\sinh\left(2\tilde{s}_{\lambda,\alpha}\frac{t}{\tau}\right) + \left(\tilde{s}_\alpha s_\alpha + \frac{\lambda^2}{4}\right)\cosh\left(2\tilde{s}_{\lambda,\alpha}\frac{t}{\tau}\right) + \tilde{s}_\alpha\left(\tilde{s}_\alpha - s_\alpha\right)}, \tag{97}$$

*where $\tilde{s}_{\lambda,\alpha} = \sqrt{\tilde{s}_\alpha^2 + \frac{\lambda^2}{4}}$, $s_{\lambda,\alpha} = \sqrt{s_\alpha(0)^2 + \frac{\lambda^2}{4}}$, and $s_\alpha = s_\alpha(0)$. We find that under different limits of $\lambda$, the transition function converges pointwise to the sigmoidal ($\lambda \to 0$) and exponential ($\lambda \to \pm\infty$) transition functions,*

$$\gamma_\alpha(t; \lambda) \to \begin{cases} \frac{e^{2\tilde{s}_\alpha\frac{t}{\tau}} - 1}{e^{2\tilde{s}_\alpha\frac{t}{\tau}} - 1 + \frac{\tilde{s}_\alpha}{s_\alpha(0)}} & as \ \lambda \to 0, \\ 1 - e^{-|\lambda|\frac{t}{\tau}} & as \ \lambda \to \pm\infty \end{cases} . \tag{98}$$

*Proof.* According to Theorem C.4, the network function is given by the equation

$$\boldsymbol{W}_2\boldsymbol{W}_1(t) = \boldsymbol{Z}_2(t)\boldsymbol{A}^{-1}(t)\boldsymbol{Z}_1^T(t), \tag{99}$$

which depends on the variables of the initialization **B** and **C**. Plugging the expressions for a task-aligned initialization $\boldsymbol{W}_1(0)$ and $\boldsymbol{W}_2(0)$ into these variables we get the following simplified expressions,

$$\mathbf{B} = \boldsymbol{R}\underbrace{\left(\boldsymbol{S}_2(\tilde{\boldsymbol{G}} + \tilde{\boldsymbol{H}}\tilde{\boldsymbol{G}}) + \boldsymbol{S}_1(\tilde{\boldsymbol{G}} - \tilde{\boldsymbol{H}}\tilde{\boldsymbol{G}})\right)}_{\boldsymbol{D}_B}, \tag{100}$$

$$\mathbf{C} = \boldsymbol{R}\underbrace{\left(\boldsymbol{S}_2(\tilde{\boldsymbol{G}} - \tilde{\boldsymbol{H}}\tilde{\boldsymbol{G}}) - \boldsymbol{S}_1(\tilde{\boldsymbol{G}} + \tilde{\boldsymbol{H}}\tilde{\boldsymbol{G}})\right)}_{\boldsymbol{D}_C}, \tag{101}$$

where we define the diagonal matrices $\boldsymbol{D}_B$ and $\boldsymbol{D}_C$ for ease of notation. Using these expressions, we now get the following time-dependent expressions for $\boldsymbol{Z}_2(t)$, $\boldsymbol{A}^{-1}(t)$, and $\boldsymbol{Z}_1(t)$,

$$\boldsymbol{Z}_1(t) = \frac{1}{2}\tilde{\boldsymbol{V}}\left((\tilde{\boldsymbol{G}} - \tilde{\boldsymbol{H}}\tilde{\boldsymbol{G}})e^{\tilde{\boldsymbol{S}}_\lambda\frac{t}{\tau}}\boldsymbol{D}_B - (\tilde{\boldsymbol{G}} + \tilde{\boldsymbol{H}}\tilde{\boldsymbol{G}})e^{-\tilde{\boldsymbol{S}}_\lambda\frac{t}{\tau}}\boldsymbol{D}_C\right)\boldsymbol{R}^T \tag{102}$$

$$\boldsymbol{Z}_2(t) = \frac{1}{2}\tilde{\boldsymbol{U}}\left((\tilde{\boldsymbol{G}} + \tilde{\boldsymbol{H}}\tilde{\boldsymbol{G}})e^{\tilde{\boldsymbol{S}}_\lambda\frac{t}{\tau}}\boldsymbol{D}_B + (\tilde{\boldsymbol{G}} - \tilde{\boldsymbol{H}}\tilde{\boldsymbol{G}})e^{-\tilde{\boldsymbol{S}}_\lambda\frac{t}{\tau}}\boldsymbol{D}_C\right)\boldsymbol{R}^T \tag{103}$$

$$\boldsymbol{A}(t) = \boldsymbol{R}\left(\mathbf{I} + \left(\frac{e^{2\tilde{\boldsymbol{S}}_\lambda\frac{t}{\tau}} - \mathbf{I}}{4\tilde{\boldsymbol{S}}_\lambda}\right)\boldsymbol{D}_B^2 - \left(\frac{e^{-2\tilde{\boldsymbol{S}}_\lambda\frac{t}{\tau}} - \mathbf{I}}{4\tilde{\boldsymbol{S}}_\lambda}\right)\boldsymbol{D}_C^2\right)\boldsymbol{R}^T \tag{104}$$

Plugging these expressions into the expression for the network function, notice that the $\boldsymbol{R}$ terms cancel each other resulting in following equation

$$\boldsymbol{W}_2\boldsymbol{W}_1(t) = \tilde{\boldsymbol{U}}\underbrace{\left(\frac{\left((\tilde{\boldsymbol{G}} - \tilde{\boldsymbol{H}}\tilde{\boldsymbol{G}})e^{\tilde{\boldsymbol{S}}_\lambda\frac{t}{\tau}}\boldsymbol{D}_B - (\tilde{\boldsymbol{G}} + \tilde{\boldsymbol{H}}\tilde{\boldsymbol{G}})e^{-\tilde{\boldsymbol{S}}_\lambda\frac{t}{\tau}}\boldsymbol{D}_C\right)\left((\tilde{\boldsymbol{G}} + \tilde{\boldsymbol{H}}\tilde{\boldsymbol{G}})e^{\tilde{\boldsymbol{S}}_\lambda\frac{t}{\tau}}\boldsymbol{D}_B + (\tilde{\boldsymbol{G}} - \tilde{\boldsymbol{H}}\tilde{\boldsymbol{G}})e^{-\tilde{\boldsymbol{S}}_\lambda\frac{t}{\tau}}\boldsymbol{D}_C\right)}{4\mathbf{I} + \left(\frac{e^{2\tilde{\boldsymbol{S}}_\lambda\frac{t}{\tau}} - \mathbf{I}}{\tilde{\boldsymbol{S}}_\lambda}\right)\boldsymbol{D}_B^2 - \left(\frac{e^{-2\tilde{\boldsymbol{S}}_\lambda\frac{t}{\tau}} - \mathbf{I}}{\tilde{\boldsymbol{S}}_\lambda}\right)\boldsymbol{D}_C^2}\right)}_{\boldsymbol{S}(t)}\tilde{\boldsymbol{V}}^T,$$

$$\tag{105}$$

Notice that the middle term is simply a product of diagonal matrices. We can factor the numerator of this expressions as,

$$(\tilde{G}^2 - \tilde{H}^2\tilde{G}^2)e^{2\tilde{S}_\lambda\frac{t}{\tau}}D_B^2 + \left((\tilde{G} - \tilde{H}\tilde{G})^2 - (\tilde{G} + \tilde{H}\tilde{G})^2\right)D_BD_C - (\tilde{G}^2 - \tilde{H}^2\tilde{G}^2)e^{-2\tilde{S}_\lambda\frac{t}{\tau}}D_C^2 \tag{106}$$

We can further factor this expression as,

$$\tilde{G}^2(\mathbf{I} - \tilde{H}^2)\left(e^{2\tilde{S}_\lambda\frac{t}{\tau}}D_B^2 - e^{-2\tilde{S}_\lambda\frac{t}{\tau}}D_C^2\right) - 4\tilde{G}^2\tilde{H}D_BD_C. \tag{107}$$

Putting it all together we find that $S(t)$ can be expressed as,

$$S(t) = \frac{\tilde{G}^2(\mathbf{I} - \tilde{H}^2)\left(e^{2\tilde{S}_\lambda\frac{t}{\tau}}D_B^2 - e^{-2\tilde{S}_\lambda\frac{t}{\tau}}D_C^2\right) - 4\tilde{G}^2\tilde{H}D_BD_C}{4\mathbf{I} + \left(\frac{e^{2\tilde{S}_\lambda\frac{t}{\tau}}-\mathbf{I}}{\tilde{S}_\lambda}\right)D_B^2 - \left(\frac{e^{-2\tilde{S}_\lambda\frac{t}{\tau}}-\mathbf{I}}{\tilde{S}_\lambda}\right)D_C^2}. \tag{108}$$

Now using the relationship between $\tilde{H}$ and $\tilde{G}$ we use the following two identities:

$$\tilde{G}^2(\mathbf{I} - \tilde{H}^2) = \frac{\tilde{S}}{\tilde{S}_\lambda}, \qquad 4\tilde{G}^2\tilde{H} = \frac{\lambda}{\tilde{S}_\lambda} \tag{109}$$

Plugging these identities into the previous expression and multiplying the numerator and denominator by $\tilde{S}_\lambda$ gives,

$$S(t) = \frac{\tilde{S}\left(e^{2\tilde{S}_\lambda\frac{t}{\tau}}D_B^2 - e^{-2\tilde{S}_\lambda\frac{t}{\tau}}D_C^2\right) - \lambda D_BD_C}{4\tilde{S}_\lambda + e^{2\tilde{S}_\lambda\frac{t}{\tau}}D_B^2 - e^{-2\tilde{S}_\lambda\frac{t}{\tau}}D_C^2 + D_C^2 - D_B^2}. \tag{110}$$

Add and subtract $\tilde{S}\left(4\tilde{S}_\lambda + D_C^2 - D_B^2\right)$ from the numerator such that

$$S(t) = \tilde{S} - \frac{\tilde{S}\left(4\tilde{S}_\lambda + D_C^2 - D_B^2\right) + \lambda D_BD_C}{4\tilde{S}_\lambda + e^{2\tilde{S}_\lambda\frac{t}{\tau}}D_B^2 - e^{-2\tilde{S}_\lambda\frac{t}{\tau}}D_C^2 + D_C^2 - D_B^2}. \tag{111}$$

Using the form of $D_B$ and $D_C$ notice the following two identities:

$$D_BD_C = \frac{\lambda}{\tilde{S}_\lambda}\left(\tilde{S} - S_2S_1\right), \qquad D_C^2 - D_B^2 = -\frac{4}{\tilde{S}_\lambda}\left(\tilde{S}S_2S_1 + \frac{\lambda^2}{4}\mathbf{I}\right) \tag{112}$$

From the second identity we can derive a third identity,

$$4\tilde{S}_\lambda + D_C^2 - D_B^2 = 4\frac{\tilde{S}}{\tilde{S}_\lambda}\left(\tilde{S} - S_2S_1\right) \tag{113}$$

Plugging the first and third identities into the numerator for the previous expression gives,

$$S(t) = \tilde{S} - \frac{\frac{(4\tilde{S}^2 + \lambda^2\mathbf{I})}{\tilde{S}_\lambda}\left(\tilde{S} - S_2S_1\right)}{4\tilde{S}_\lambda + e^{2\tilde{S}_\lambda\frac{t}{\tau}}D_B^2 - e^{-2\tilde{S}_\lambda\frac{t}{\tau}}D_C^2 + D_C^2 - D_B^2}. \tag{114}$$

Multiply numerator and denominator by $\frac{\tilde{S}_\lambda}{4}$ and simplify terms gives the expression,

$$S(t) = \tilde{S} - \frac{\tilde{S}_\lambda^2}{\tilde{S}_\lambda^2 + \frac{\tilde{S}_\lambda}{4}\left(e^{2\tilde{S}_\lambda\frac{t}{\tau}}D_B^2 - e^{-2\tilde{S}_\lambda\frac{t}{\tau}}D_C^2\right) - \frac{\tilde{S}_\lambda}{4}\left(D_B^2 - D_C^2\right)}\left(\tilde{S} - S_2S_1\right). \tag{115}$$

Thus we have found the transition function,

$$\gamma(t;\lambda) = \frac{\frac{\tilde{S}_\lambda}{4}\left(e^{2\tilde{S}_\lambda\frac{t}{\tau}}D_B^2 - e^{-2\tilde{S}_\lambda\frac{t}{\tau}}D_C^2\right) + \frac{\tilde{S}_\lambda}{4}\left(D_C^2 - D_B^2\right)}{\frac{\tilde{S}_\lambda}{4}\left(e^{2\tilde{S}_\lambda\frac{t}{\tau}}D_B^2 - e^{-2\tilde{S}_\lambda\frac{t}{\tau}}D_C^2\right) + \frac{\tilde{S}_\lambda}{4}\left(4\tilde{S}_\lambda + D_C^2 - D_B^2\right)}. \tag{116}$$

We will use our previous identities and the definitions of $\boldsymbol{D}_B^2$ and $\boldsymbol{D}_C^2$ to simplify this expression. Notice the following identity,

$$\frac{\tilde{\boldsymbol{S}}_\lambda}{4}\left(e^{2\tilde{\boldsymbol{S}}_\lambda\frac{t}{\tau}}\boldsymbol{D}_B^2 - e^{-2\tilde{\boldsymbol{S}}_\lambda\frac{t}{\tau}}\boldsymbol{D}_C^2\right) = \tilde{\boldsymbol{S}}_\lambda\boldsymbol{S}_\lambda\sinh\left(2\tilde{\boldsymbol{S}}_\lambda\frac{t}{\tau}\right) + \left(\tilde{\boldsymbol{S}}\boldsymbol{S}(0) + \frac{\lambda^2}{4}\mathbf{I}\right)\cosh\left(2\tilde{\boldsymbol{S}}_\lambda\frac{t}{\tau}\right) \tag{117}$$

Putting it all together we get

$$\gamma(t;\lambda) = \frac{\tilde{\boldsymbol{S}}_\lambda\boldsymbol{S}_\lambda\sinh\left(2\tilde{\boldsymbol{S}}_\lambda\frac{t}{\tau}\right) + \left(\tilde{\boldsymbol{S}}\boldsymbol{S}(0) + \frac{\lambda^2}{4}\mathbf{I}\right)\cosh\left(2\tilde{\boldsymbol{S}}_\lambda\frac{t}{\tau}\right) - \left(\tilde{\boldsymbol{S}}\boldsymbol{S}(0) + \frac{\lambda^2}{4}\mathbf{I}\right)}{\tilde{\boldsymbol{S}}_\lambda\boldsymbol{S}_\lambda\sinh\left(2\tilde{\boldsymbol{S}}_\lambda\frac{t}{\tau}\right) + \left(\tilde{\boldsymbol{S}}\boldsymbol{S}(0) + \frac{\lambda^2}{4}\mathbf{I}\right)\cosh\left(2\tilde{\boldsymbol{S}}_\lambda\frac{t}{\tau}\right) + \tilde{\boldsymbol{S}}\left(\tilde{\boldsymbol{S}} - \boldsymbol{S}(0)\right)} \tag{118}$$

We will now show why under certain limits of $\lambda$ this expression simplifies to the sigmoidal and exponential dynamics discussed in the previous section.

**Sigmoidal dynamics.** When $\lambda = 0$, then $\tilde{\boldsymbol{S}}_\lambda = \tilde{\boldsymbol{S}}$ and $\boldsymbol{S}_\lambda = \boldsymbol{S}(0)$. Notice, that the coefficients for the hyperbolic functions all simplify to $\tilde{\boldsymbol{S}}\boldsymbol{S}(0)$. Using the hyperbolic identity $\sinh(x) + \cosh(x) = e^x$, we can simplify the expression for the transition function to

$$\gamma(t;\lambda) = \frac{\tilde{\boldsymbol{S}}\boldsymbol{S}(0)e^{2\tilde{\boldsymbol{S}}\frac{t}{\tau}} - \tilde{\boldsymbol{S}}\boldsymbol{S}(0)}{\tilde{\boldsymbol{S}}\boldsymbol{S}(0)e^{2\tilde{\boldsymbol{S}}\frac{t}{\tau}} - \tilde{\boldsymbol{S}}\boldsymbol{S}(0) + \tilde{\boldsymbol{S}}^2}. \tag{119}$$

Dividing the numerator and denominator by $\tilde{\boldsymbol{S}}\boldsymbol{S}(0)$ gives the final expression.

**Exponential dynamics.** In the limit as $\lambda \to \pm\infty$ the expressions $\tilde{\boldsymbol{S}}_\lambda \to \frac{|\lambda|}{2}$ and $\boldsymbol{S}_\lambda \to \frac{|\lambda|}{2}$. Additionally, in these limits because $\frac{\lambda^2}{4}\mathbf{I} \gg \tilde{\boldsymbol{S}}\boldsymbol{S}(0)$ then $\left(\tilde{\boldsymbol{S}}\boldsymbol{S}(0) + \frac{\lambda^2}{4}\mathbf{I}\right) \to \frac{\lambda^2}{4}\mathbf{I}$. As a result of these simplifications the coefficients for the hyperbolic functions all simplify to $\frac{\lambda^2}{4}\mathbf{I}$. As a result we can again use the hyperbolic identity $\sinh(x) + \cosh(x) = e^x$ to simplify the expression as

$$\gamma(t;\lambda) = \frac{\frac{\lambda^2}{4}e^{|\lambda|\frac{t}{\tau}} - \frac{\lambda^2}{4}\mathbf{I}}{\frac{\lambda^2}{4}e^{|\lambda|\frac{t}{\tau}} + \tilde{\boldsymbol{S}}\left(\tilde{\boldsymbol{S}} - \boldsymbol{S}(0)\right)}. \tag{120}$$

Dividing the numerator and denominator by $\frac{\lambda^2}{4}$ results in all terms without a coefficient proportional to $\lambda^2$ vanishing, which simplifying further gives the final expression. $\qquad\square$

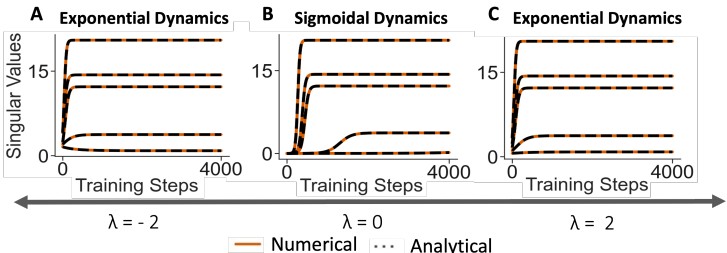

Figure 4: Simulated and analytical dynamics of the singular values of the network function with *relative scale* lambda **A** $\lambda = -2$ **B** $\lambda = 0$ **C** $\lambda = 2$ initialized as described in F.7.

### D.2 Dynamics of the representation from the Lazy to the Rich Regime

The *lazy* and *rich* regimes are defined by the dynamics of the NTK of the network. *Lazy* learning occurs when the NTK is constant, *rich* learning occurs when it is not. (Farrell et al. (2023b))
The NTK intuitively measures the movement of the network representations through training. As shown in (Braun et al. (2022)), in specific experimental setup, we can calculate the NTK of the network in terms of the internal representations in a straightforward way:

$$\text{NTK} = \mathbf{I}_{N_o} \otimes \mathbf{X}^T\mathbf{W}_1^T\mathbf{W}_1(t)\mathbf{X} + \mathbf{W}_2\mathbf{W}_2^T(t) \otimes \mathbf{X}^T\mathbf{X} \tag{121}$$

In order to better understand the effect of $\lambda$ on NTK dynamics, we first prove some theorems involving the Singular Values of the $\lambda$-*balanced* weights, and the representations of a $\lambda$-*balanced* network.

### D.2.1 Lambda-balanced singular value

**Theorem D.2.** *Under a $\lambda$-Balanced initialization 3, if the network function $W_2 W_1(t) = U(t)S(t)V^T(t)$ is full-rank 5 and we define $S_\lambda(t) = \sqrt{S^2(t) + \frac{\lambda^2}{4}\mathbf{I}}$ , then we can recover the parameters $W_2(t) = U(t)S_2(t)R^T(t)$, $W_1(t) = R(t)S_1(t)V^T(t)$ up to time-dependent orthogonal transformation R(t) of size $N_h \times N_h$, where*

$$S_1(t) = \left(\left(S_\lambda(t) - \tfrac{\lambda \mathbf{I}}{2}\right)^{\frac{1}{2}} \quad 0_{\max(0, N_i - N_o)}\right) \qquad S_2(t) = \left(\left(S_\lambda(t) + \tfrac{\lambda \mathbf{I}}{2}\right)^{\frac{1}{2}} 0_{\max(0, N_o - N_i)}\right)$$
(122)

*Proof.* We prove the case $N_i \le N_o$ and $N_h = min(N_i, N_o)$. The proof for $N_o \le N_i$ follows the same structure. Let $USV^T = W_2(t)W_1(t)$ be the Singular Value Decomposition of the product of the weights at training step $t$. We will use $W_2 = W_2(t), W_1 = W_1(t)$ as a shorthand.

By properties of Singular Value Decomposition, we can write $W_2 = US_2R^T, W_1 = RS_1V^T$, where $R$ is an orthonormal matrix and $S_2, S_1$ are diagonal (possibly rectangular) matrices.

The Balanced property states that $W_2^T W_2 - W_1 W_1^T = \lambda\mathbf{I}$. We know this holds for any $t$ since this is a conserved quantity in linear networks.

Hence

$$RS_2^T S_2 R^T - RS_1 S_1 R^T = \lambda\mathbf{I} \tag{123}$$

$$S_2^T S_2 - S_1 S_1 = \lambda\mathbf{I} \tag{124}$$

The matrices $S_1, S_2$, have shapes $(N_h, N_i), (N_o, N_h)$ respectively. We introduce the diagonal matrices $\hat{S}_1$ of shape $(N_h, N_i)$, $\hat{S}_2$ of shape $(N_i, N_h)$ such that the zero matrix has size $(N_o - N_i, N_h)$ :

$$S_1 = \left(\hat{S}_1\right), \quad S_2 = \begin{pmatrix} \hat{S}_2 \\ 0 \end{pmatrix} \tag{125}$$

Hence

$$S_2^T S_2 - S_1 S_1 = \lambda\mathbf{I} \tag{126}$$

From the equation above and the fact that $\hat{S}_1 \hat{S}_2 = S$ we derive that:

$$\hat{S}_2 = \left(\frac{\sqrt{\lambda^2\mathbf{I} + 4S^2} + \lambda\mathbf{I}}{2}\right)^{\frac{1}{2}}, \quad \hat{S}_1 = \left(\frac{\sqrt{\lambda^2\mathbf{I} + 4S^2} - \lambda\mathbf{I}}{2}\right)^{\frac{1}{2}}, \tag{127}$$

Hence

$$W_2 = U\begin{pmatrix} \left(\frac{\sqrt{\lambda^2\mathbf{I}+4S^2}+\lambda\mathbf{I}}{2}\right)^{\frac{1}{2}} \\ 0_{\max(0, N_o - N_i)} \end{pmatrix}, R^T, \quad W_1 = R\left(\left(\frac{\sqrt{\lambda^2\mathbf{I}+4S^2}-\lambda\mathbf{I}}{2}\right)^{\frac{1}{2}} \quad 0_{\max(0, N_i - N_o)}\right)V^T$$
(128)

$\square$

### D.2.2 Convergence proof

With our solution, $\mathbf{QQ}^T(t)$, which captures the temporal dynamics of the similarity between hidden layer activations, we can analyze the network's internal representations in relation to the task. This allows us to determine whether the network adopts a *rich* or *lazy* representation, depending on the value of $\lambda$. Consider a $\lambda$-Balanced network training on data $\mathbf{\Sigma}^{yx} = \tilde{U}\tilde{S}\tilde{V}^T$. We assume that the convergence is toward global minima and B is invertible

**Theorem D.3.** *Under the assumptions of Theorem C.5, the network function converges to $\tilde{U}\tilde{S}\tilde{V}^T$ and acquires the internal representation, that is $\mathbf{W}_1^T\mathbf{W}_1 = \tilde{V}\tilde{S}_1^2\tilde{V}^T$ and $\mathbf{W}_2\mathbf{W}_2^T = \tilde{U}\tilde{S}_2^2\tilde{U}^T$*

*Proof.* As training time increases, all terms including a matrix exponential with negative exponent in Equation 58 vanish to zero, as $\boldsymbol{S_\lambda} = \tilde{\boldsymbol{S}}_\lambda$ is a diagonal matrix with entries larger zero

As training time increases, all terms in the equations vanish to zero. Terms in Equation 58 decay as

$$\lim_{t\to\infty} e^{-\sqrt{\tilde{S}^2 + \frac{\lambda^2 \mathbf{I}}{4}}\frac{t}{\tau}} = \mathbf{0}. \tag{129}$$

and

$$\lim_{t\to\infty} e^{\lambda_\perp \frac{t}{\tau}} e^{-\sqrt{\tilde{S}^2 + \frac{\lambda^2}{4}\mathbf{I}}\frac{t}{\tau}} = \mathbf{0}. \tag{130}$$

where $\tilde{\boldsymbol{S_\lambda}} = \tilde{\boldsymbol{S}}_\lambda$ is a diagonal matrix with entries larger zero

Therefore, in the temporal limit, eq. 58 reduces to

$$\lim_{t\to\infty} \mathbf{QQ}^T(t) = \lim_{t\to\infty} \begin{bmatrix} \mathbf{W}_1^T\mathbf{W}_1(t) & \mathbf{W}_1^T\mathbf{W}_2^T(t) \\ \mathbf{W}_2\mathbf{W}_1(t) & \mathbf{W}_2^T\mathbf{W}_2(t) \end{bmatrix} \tag{131}$$

$$= \begin{bmatrix} \tilde{V}(\tilde{G} - \tilde{H}\tilde{G}) \\ \tilde{U}(\tilde{H}\tilde{G} + \tilde{G}) \end{bmatrix} \left[ \tilde{S}_\lambda^{-1} \right]^{-1} \left[ (\tilde{V}(\tilde{G} - \tilde{H}\tilde{G}))^T \quad (\tilde{U}(\tilde{H}\tilde{G} + \tilde{G}))^T \right] \tag{132}$$

$$= \begin{bmatrix} \tilde{V}(\tilde{G} - \tilde{H}\tilde{G})\tilde{S}_\lambda(\tilde{G} - \tilde{H}\tilde{G})^T\tilde{V}^T & \tilde{V}(\tilde{G} - \tilde{H}\tilde{G})\tilde{S}_\lambda(\tilde{H}\tilde{G} + \tilde{G})^T\tilde{U}^T \\ \tilde{U}(\tilde{H}\tilde{G} + \tilde{G})\tilde{S}_\lambda(\tilde{G} - \tilde{H}\tilde{G})^T\tilde{V}^T & \tilde{U}(\tilde{H}\tilde{G} + \tilde{G})\tilde{S}_\lambda(\tilde{H}\tilde{G} + \tilde{G})^T\tilde{U}^T \end{bmatrix}. \tag{133}$$

$$(\tilde{G} - \tilde{H}\tilde{G})\tilde{S}_\lambda(\tilde{G} + \tilde{H}\tilde{G}) = \frac{\boldsymbol{S_\lambda}(1 - \tilde{H}^2)}{1 + \tilde{H}^2} = \tilde{S} \tag{134}$$

$$\tilde{S}_\lambda(\tilde{G} - \tilde{H}\tilde{G})^2 = \frac{\tilde{S}_\lambda(1 + \tilde{H}^2)}{1 + \tilde{H}^2} - \frac{\tilde{S}_\lambda(2\tilde{H})}{1 + \tilde{H}^2} = \frac{\sqrt{4\tilde{S}^2 + \lambda^2\mathbf{I}} - \lambda\mathbf{I}}{2} \tag{135}$$

$$\tilde{S}_\lambda(\tilde{G} + \tilde{H}\tilde{G})^2 = \frac{\tilde{S}_\lambda(1 + \tilde{H}^2)}{1 + \tilde{H}^2} + \frac{\tilde{S}_\lambda(2\tilde{H})}{1 + \tilde{H}^2} = \frac{\sqrt{4\tilde{S}^2 + \lambda^2\mathbf{I}} + \lambda\mathbf{I}}{2} \tag{136}$$

$$\lim_{t\to\infty} \mathbf{QQ}^T(t) = \lim_{t\to\infty} \begin{bmatrix} \mathbf{W}_1^T\mathbf{W}_1(t) & \mathbf{W}_1^T\mathbf{W}_2^T(t) \\ \mathbf{W}_2\mathbf{W}_1(t) & \mathbf{W}_2^T\mathbf{W}_2(t) \end{bmatrix} \tag{137}$$

$$= \begin{bmatrix} \tilde{V}\boldsymbol{S}_1^2\tilde{V}^T & \tilde{V}\tilde{S}\tilde{U}^T \\ \tilde{U}\tilde{S}\tilde{V}^T & \tilde{U}\boldsymbol{S}_2^2\tilde{U}^T \end{bmatrix}. \tag{138}$$

$\square$

### D.2.3 Representation in the limit

**Theorem D.4.** *Under the assumptions of Theorem C.5, training on data $\mathbf{\Sigma}^{yx} = \tilde{U}\tilde{S}\tilde{V}^T$, as $\lambda \to \infty$ the representation tends to*

$$W_2 W_2^T = \tilde{U} \begin{pmatrix} \lambda \mathbf{I} & 0_{\max(0,N_o-N_i)} \\ 0_{\max(0,N_o-N_i)} & 0 \end{pmatrix} \tilde{U}^T \quad W_1^T W_1 = \frac{1}{\lambda} \tilde{V} \begin{pmatrix} \tilde{S}^2 & 0_{max(0,N_i-N_o)} \\ 0_{\max(0,N_i-N_o)} & 0 \end{pmatrix} \tilde{V}^T$$

*As $\lambda \to -\infty$*

$$W_2 W_2^T = -\frac{1}{\lambda}\tilde{U} \begin{pmatrix} \tilde{S}^2 & 0_{\max(0,N_o-N_i)} \\ 0_{\max(0,N_o-N_i)} & 0 \end{pmatrix} \tilde{U}^T, \quad W_1^T W_1 = \tilde{V} \begin{pmatrix} -\lambda \mathbf{I} & 0_{\max(0,N_i-N_o)} \\ 0_{\max(0,N_i-N_o)} & 0 \end{pmatrix} \tilde{V}^T$$

*As $\lambda \to -\infty$*

$$W_2 W_2^T = -\frac{1}{\lambda}\tilde{U} \begin{pmatrix} \tilde{S}^2 & 0_{\max(0,N_o-N_i)} \\ 0_{\max(0,N_o-N_i)} & 0 \end{pmatrix} \tilde{U}^T, \quad W_1^T W_1 = \tilde{V} \begin{pmatrix} -\lambda \mathbf{I} & 0_{\max(0,N_i-N_o)} \\ 0_{\max(0,N_i-N_o)} & 0 \end{pmatrix} \tilde{V}^T$$

*Proof.* We start from the representation derived in D.3 and using the Taylor expansion of $f(x) = \sqrt{1+x^2}$, we compute

$$\frac{\sqrt{\lambda^2 \mathbf{I} + 4\tilde{S}^2} + \lambda \mathbf{I}}{2} = \frac{|\lambda|\sqrt{1 + \left(\frac{2\tilde{S}}{\lambda}\right)^2} + \lambda \mathbf{I}}{2} \tag{139}$$

$$\frac{|\lambda|\left(1 + \left(\frac{2\tilde{S}}{\lambda}\right)^2 + O(\lambda^{-4})\right) + \lambda \mathbf{I}}{2} = \frac{|\lambda| + \lambda}{2} + \frac{\tilde{S}^2}{|\lambda|} + O(\lambda^{-3}) \tag{140}$$

Hence

$$\lim_{\lambda \to \infty} \frac{\sqrt{\lambda^2 \mathbf{I} + 4\tilde{S}^2} + \lambda \mathbf{I}}{2} = \lambda \mathbf{I}, \quad \lim_{\lambda \to -\infty} \frac{\sqrt{\lambda^2 \mathbf{I} + 4\tilde{S}^2} + \lambda \mathbf{I}}{2} = \frac{\tilde{S}^2}{|\lambda|} = -\frac{\tilde{S}^2}{\lambda} \tag{141}$$

Similarly,

$$\frac{\sqrt{\lambda^2 \mathbf{I} + 4\tilde{S}^2} - \lambda \mathbf{I}}{2} = \frac{|\lambda| - \lambda}{2} + \frac{\tilde{S}^2}{|\lambda|} + O(\lambda^{-3}) \tag{142}$$

$$\lim_{\lambda \to \infty} \frac{\sqrt{\lambda^2 \mathbf{I} + 4\tilde{S}^2} - \lambda \mathbf{I}}{2} = \frac{\tilde{S}^2}{\lambda}, \quad \lim_{\lambda \to -\infty} \frac{\sqrt{\lambda^2 \mathbf{I} + 4\tilde{S}^2} - \lambda \mathbf{I}}{2} = \frac{\tilde{S}^2}{|\lambda|} = -\lambda \mathbf{I} \tag{143}$$

Since $\tilde{U}, \tilde{V}$ are independent of $\lambda$:

$$\lim_{\lambda \to \pm\infty} W_2 W_2^T = \tilde{U} \left( \lim_{\lambda \to \pm\infty} S_2 \right) \tilde{U}^T \tag{144}$$

$$\lim_{\lambda \to \pm\infty} W_1^T W_1 = \tilde{V} \left( \lim_{\lambda \to \pm\infty} S_1 \right) \tilde{V}^T \tag{145}$$

$\square$

As $|\lambda| \to \infty$, one of the network representations approaches a scaled identity matrix, while the other tends toward zero. Intuitively, this suggests that the representations shift less and less as $|\lambda|$ increases. Next, we demonstrate that the NTK becomes progressively less variable as $|\lambda|$ grows and ultimately converges to zero.

### D.2.4 NTK movement

Relationship between $\lambda$ and the NTK of the network

**Theorem D.5.** *Under the assumptions of Theorem C.5, consider a linear network training on data* $\mathbf{\Sigma}^{yx} = \tilde{\boldsymbol{U}}\tilde{\boldsymbol{S}}\tilde{\boldsymbol{V}}^T$. *At any arbitrary training time* $t \geq 0$, *let* $\boldsymbol{W}_2(t)\boldsymbol{W}_1(t) = \boldsymbol{U}^*\boldsymbol{S}^*\boldsymbol{V}^{*T}$. *Then,*

*1. For any $\lambda \in \mathbf{R}$:*

$$
\begin{aligned}
NTK(0) = \mathbf{I}_{N_o} \otimes \boldsymbol{X}^T \boldsymbol{V} \begin{pmatrix} \frac{\sqrt{\lambda^2\mathbf{I}+4\boldsymbol{S}^{*2}}-\lambda\mathbf{I}}{2} & 0 \\ 0 & 0 \end{pmatrix} \boldsymbol{V}^T \boldsymbol{X} \\
+ \boldsymbol{U} \begin{pmatrix} \frac{\sqrt{\lambda^2\mathbf{I}+4\boldsymbol{S}^{*2}}+\lambda\mathbf{I}}{2} & 0 \\ 0 & 0 \end{pmatrix} \boldsymbol{U}^T \otimes \boldsymbol{X}^T \boldsymbol{X}
\end{aligned}
\tag{146}
$$

$$
\begin{aligned}
NTK(t) = \mathbf{I}_{N_o} \otimes \boldsymbol{X}^T \boldsymbol{V}^* \begin{pmatrix} \frac{\sqrt{\lambda^2\mathbf{I}+4\boldsymbol{S}^{*2}}-\lambda\mathbf{I}}{2} & 0 \\ 0 & 0 \end{pmatrix} \boldsymbol{V}^{*T} \\
+ \boldsymbol{U}^* \begin{pmatrix} \frac{\sqrt{\lambda^2\mathbf{I}+4\boldsymbol{S}^{*2}}+\lambda\mathbf{I}}{2} & 0 \\ 0 & 0 \end{pmatrix} \boldsymbol{U}^{*T} \otimes \boldsymbol{X}^T \boldsymbol{X}
\end{aligned}
\tag{147}
$$

*2. As $\lambda \to \infty$:*

$$
NTK(t) - NTK(0) \to \frac{1}{\lambda} \left( \mathbf{I}_{N_o} \otimes \boldsymbol{X}^T \boldsymbol{V}^* \tilde{\boldsymbol{S}}^{*2} \boldsymbol{V}^{*T} \boldsymbol{X} - \mathbf{I}_{N_o} \otimes \boldsymbol{X}^T \boldsymbol{V} \tilde{\boldsymbol{S}}^2 \boldsymbol{V}^T \boldsymbol{X} \right) \to 0
\tag{148}
$$

*3. As $\lambda \to -\infty$:*

$$
NTK(t) - NTK(0) \to \frac{1}{\lambda} \left( \boldsymbol{U}\tilde{\boldsymbol{S}}^2 \boldsymbol{U}^T \otimes \boldsymbol{X}^T \boldsymbol{X} - \boldsymbol{U}^* \tilde{\boldsymbol{S}}^{*2} \boldsymbol{U}^{*T} \otimes \boldsymbol{X}^T \boldsymbol{X} \right) \to 0
\tag{149}
$$

***Proof.*** Follows by substituting the expressions for the network representations in terms of $\lambda$ from (Braun et al. (2022))'s expression for the NTK of a linear network. Similarly, follows from substituting the limit expressions for the network representations and the fact that the Kronecker product is linear in both arguments. $\square$

The theorem above demonstrates that as $|\lambda| \to \infty$, the NTK of a $\lambda$-Balanced network remains constant. This indicates that the network operates in the *lazy* regime throughout all training steps. This finding is significant as it highlights the impact of weight initialization on learning regimes.

### D.3 Representation robustness and sensitivity to noise

As derived in (Braun et al., 2024), the expected mean squared error under additive, independent and identically distributed input noise with mean $\mu = 0$ and variance $\sigma_{\mathbf{x}}^2$ is

$$
\left\langle \frac{1}{2P} \sum_{i=1}^{P} ||\mathbf{W}_2 \mathbf{W}_1 (\mathbf{x}_{\mathbf{x}} + \xi_i) - \mathbf{y}_i||_2^2 \right\rangle_{\xi_{\mathbf{x}}} = \sigma_{\mathbf{x}}^2 ||\mathbf{W}_2 \mathbf{W}_1||_F^2 + c,
\tag{150}
$$

where $c = \frac{1}{2}\text{Tr}(\tilde{\mathbf{\Sigma}}^{yy}) - \frac{1}{2}\text{Tr}(\tilde{\mathbf{\Sigma}}^{yx}\tilde{\mathbf{\Sigma}}^{yxT})$ is a noise independent constant that only depends on the statistics of the training data. In Theorem D.3 we show that the network function converges to $\tilde{\mathbf{U}}\tilde{\mathbf{S}}\tilde{\mathbf{V}}^T$ and therefore

$$
\begin{aligned}
\sigma_{\mathbf{x}}^2 ||\mathbf{W}_2 \mathbf{W}_1||_F^2 &= \sigma_{\mathbf{x}}^2 ||\tilde{\mathbf{U}}\tilde{\mathbf{S}}\tilde{\mathbf{V}}^T||_F^2 \\
&= \sigma_{\mathbf{x}}^2 ||\tilde{\mathbf{S}}||_F^2 \\
&= \sigma_{\mathbf{x}}^2 \sum_{i=1}^{N_h} \tilde{\mathbf{S}}_i^2
\end{aligned}
\tag{151}
$$

As derived in (Braun et al., 2024), under the assumption of whitened inputs (Assumption 2), in the case of additive parameter noise with $\mu = 0$ and variance $\sigma_\mathbf{W}^2$, the expected mean squared error is

$$\left\langle \frac{1}{2P} \sum_{i=1}^{P} ||\left(\mathbf{W}_2 + \xi_{\mathbf{W}_2}\right)\left(\mathbf{W}_1 + \xi_{\mathbf{W}_1}\right)\mathbf{x}_i - \mathbf{y}_i||_2^2 \right\rangle_{\xi_{\mathbf{W}_1}, \xi_{\mathbf{W}_2}} \tag{152}$$
$$= \frac{1}{2} N_i \sigma_\mathbf{W}^2 ||\mathbf{W}_2||_F^2 + \frac{1}{2} N_o \sigma_\mathbf{W}^2 ||\mathbf{W}_1||_F^2 + \frac{1}{2} N_i N_h N_o \sigma^4 + c.$$

Using Theorem D.3, we have

$$\begin{aligned}
||\mathbf{W}_1||_F^2 &= \text{Tr}(\mathbf{W}_1^T \mathbf{W}_1) \\
&= \text{Tr}\left(\frac{\sqrt{\lambda^2 \mathbf{I} + 4\tilde{\mathbf{S}}^2} + \lambda \mathbf{I}}{2}\right) \\
&= \frac{1}{2}\left(\sum_{i=1}^{N_h} \sqrt{\lambda^2 + 4\tilde{\mathbf{S}}_i^2} + \lambda\right)
\end{aligned} \tag{153}$$

and

$$\begin{aligned}
||\mathbf{W}_2||_F^2 &= \text{Tr}(\mathbf{W}_2 \mathbf{W}_2^T) \\
&= \text{Tr}\left(\frac{\sqrt{\lambda^2 \mathbf{I} + 4\tilde{\mathbf{S}}^2} - \lambda \mathbf{I}}{2}\right) \\
&= \frac{1}{2}\left(\sum_{i=1}^{N_h} \sqrt{\lambda^2 + 4\tilde{\mathbf{S}}_i^2} - \lambda\right).
\end{aligned} \tag{154}$$

To find the $\lambda$ that minimises the expected loss, we substitute the equations for the norms, take the partial derivative with respect to $\lambda$ and set it to zero

$$\begin{aligned}
&\frac{\partial \langle \mathcal{L} \rangle_{\xi_{\mathbf{W}_1}, \xi_{\mathbf{W}_2}}}{\partial \lambda} \overset{!}{=} 0 \\
\Leftrightarrow &\frac{1}{4} N_i \sigma_\mathbf{W}^2 \frac{\partial}{\partial \lambda}\left(\sum_{i=1}^{N_h} \sqrt{\lambda^2 + 4\tilde{\mathbf{S}}_i^2} - \lambda\right) + \frac{1}{4} N_o \sigma_\mathbf{W}^2 \frac{\partial}{\partial \lambda}\left(\sum_{i=1}^{N_h} \sqrt{\lambda^2 + 4\tilde{\mathbf{S}}_i^2} + \lambda\right) = 0 \\
\Leftrightarrow &N_i \sum_{i=1}^{N_h} \frac{\lambda}{\sqrt{\lambda^2 + 4\tilde{\mathbf{S}}_i^2}} - N_i N_h + N_o \sum_{i=1}^{N_h} \frac{\lambda}{\sqrt{\lambda^2 + 4\tilde{\mathbf{S}}_i^2}} + N_o N_h = 0 \\
\Leftrightarrow &\sum_{i=1}^{N_h} \frac{\lambda}{\sqrt{\lambda^2 + 4\tilde{\mathbf{S}}_i^2}} = N_h \frac{N_i - N_o}{N_i + N_o}.
\end{aligned} \tag{155}$$

It follows, that under the assumption that $N_i = N_o$, the equation reduces to

$$\sum_{i=1}^{N_h} \frac{\lambda}{\sqrt{\lambda^2 + 4\tilde{\mathbf{S}}_i^2}} = 0. \tag{156}$$

We note, that the denominator is always positive and therefore, that the left-hand side of the equation is always larger zero for any $\lambda > 0$, and smaller than zero for any $\lambda < 0$. The euqation is therefore only solved for $\lambda = 0$.

## D.4 Effect of the architecture from the lazy to the Rich Regime

**Theorem D.6.** *Under the conditions of Theorem C.5, when $\lambda_\perp > 0$, the network enters a regime referred to as the delayed-rich phase. In this phase, the learning rate is determined by two competing exponential factors:*

$$e^{\lambda_\perp \frac{t}{\tau}} e^{-\sqrt{\tilde{S}^2 + \frac{\lambda^2}{4}\mathbf{I}} \frac{t}{\tau}}$$

*and*

$$e^{-\sqrt{\tilde{S}^2 + \frac{\lambda^2}{4}\mathbf{I}}\frac{t}{\tau}}.$$

*As $\lambda$ increases, different parts of the network exhibit distinct learning behaviors: some components adapt quickly and converge exponentially with lambda, while others are constrained by the singular values of the network, resulting in slower adaptation.*

*Proof.* The solution to Theorem C.5 is governed by two time-dependent terms:

$$e^{-\sqrt{\tilde{S}^2 + \frac{\lambda^2 \mathbf{I}}{4}}\frac{t}{\tau}} \quad \text{and} \quad e^{\lambda_\perp \frac{t}{\tau}} e^{-\sqrt{\tilde{S}^2 + \frac{\lambda^2}{4}\mathbf{I}}\frac{t}{\tau}}.$$

The first term exhibits exponential decay with rate $\lambda$, approaching zero as time progresses:

$$\lim_{t \to \infty} e^{-\sqrt{\tilde{S}^2 + \frac{\lambda^2 \mathbf{I}}{4}}\frac{t}{\tau}} = \mathbf{0}.$$

The second term also decays, but at a rate governed by the singular values $\tilde{S}$, as $\lambda$ tends to infinity:

$$\lim_{t \to \infty} e^{\lambda_\perp \frac{t}{\tau}} e^{-\sqrt{\tilde{S}^2 + \frac{\lambda^2}{4}\mathbf{I}}\frac{t}{\tau}} = \mathbf{0}.$$

Since

$$\lambda_\perp - \sqrt{\tilde{S}^2 + \frac{\lambda^2}{4}\mathbf{I}} > 0,$$

we have

$$\lim_{\lambda \to \infty} \left( \lambda_\perp - \sqrt{\tilde{S}^2 + \frac{\lambda^2}{4}\mathbf{I}} \right) = \tilde{S}.$$

Thus, as $\lambda$ increases, the convergence rate slows for certain parts of the network (those governed by larger singular values), while other components continue to learn more quickly. This explains the delay observed in the delayed-rich regime. $\square$

# E  Appendix: Application

## E.1  Appendix: Continual Learning

We build upon the derivation presented in Braun et al. (2022) to incorporate the dynamics of continual learning throughout the entire learning trajectory. Utilizing the assumption of whitened inputs, the entire batch loss for the $i$th task is

$$
\begin{aligned}
\mathcal{L}_i(\mathcal{T}_j) &= \frac{1}{2P} \|\mathbf{W}_2\mathbf{W}_1\mathbf{X}_i - \mathbf{Y}_i\|_F^2 \\
&= \frac{1}{2P} \operatorname{Tr}\left( (\mathbf{W}_2\mathbf{W}_1\mathbf{X}_i - \mathbf{Y}_i\,|)\,(\mathbf{W}_2\mathbf{W}_1\mathbf{X}_i - \mathbf{Y}_i\,|)^T \right) \\
&= \frac{1}{2P} \operatorname{Tr}\left( \mathbf{W}_2\mathbf{W}_1\mathbf{X}_i\mathbf{X}_i^T(\mathbf{W}_2\mathbf{W}_1)^T \right) - \frac{1}{P}\operatorname{Tr}\left( \mathbf{W}_2\mathbf{W}_1\mathbf{X}_i\mathbf{Y}_i^T \right) + \frac{1}{2P}\operatorname{Tr}\left( \mathbf{Y}_i\mathbf{Y}_i^T \right) \\
&= \frac{1}{2}\operatorname{Tr}\left( \mathbf{W}_2\mathbf{W}_1(\mathbf{W}_2\mathbf{W}_1)^T \right) - \operatorname{Tr}\left( \mathbf{W}_2\mathbf{W}_1\tilde{\boldsymbol{\Sigma}}_i^{yx^T} \right) + \frac{1}{2}\operatorname{Tr}\left( \tilde{\boldsymbol{\Sigma}}_i^{yy} \right) \\
&= \frac{1}{2}\operatorname{Tr}\left( \left( \mathbf{W}_2\mathbf{W}_1 - \tilde{\boldsymbol{\Sigma}}_i^{yx} \right)\left( \mathbf{W}_2\mathbf{W}_1 - \tilde{\boldsymbol{\Sigma}}_i^{yx} \right)^T - \tilde{\boldsymbol{\Sigma}}_i^{yx}\tilde{\boldsymbol{\Sigma}}_i^{yx^T} \right) + \frac{1}{2}\left( \tilde{\boldsymbol{\Sigma}}_i^{yy} \right) \\
&= \frac{1}{2}\left\| \mathbf{W}_2\mathbf{W}_1 - \tilde{\boldsymbol{\Sigma}}_i^{yx} \right\|_F^2 \underbrace{-\frac{1}{2}\operatorname{Tr}\left( \tilde{\boldsymbol{\Sigma}}_i^{yx}\tilde{\boldsymbol{\Sigma}}_i^{yx^T} \right) + \frac{1}{2}\left( \tilde{\boldsymbol{\Sigma}}_i^{yy} \right)}_{c}.
\end{aligned}
$$

Hence, the extent of forgetting, denoted as $\mathcal{F}$ for task $\mathcal{T}_i$ during training on task $\mathcal{T}_k$ subsequent to training the network on task $\mathcal{T}_j$, specifically, the relative change in loss, is entirely dictated by the similarity structure among tasks.

$$\mathcal{F}_i\left(\mathcal{T}_j, \mathcal{T}_k\right) = \mathcal{L}_i\left(\mathcal{T}_k\right) - \mathcal{L}_i\left(\mathcal{T}_j\right)$$

$$= \frac{1}{2}\left\|\tilde{\mathbf{\Sigma}}_k^{yx} - \tilde{\mathbf{\Sigma}}_i^{yx}\right\|_F^2 + c - \frac{1}{2}\left\|\mathbf{W}_2\mathbf{W}_1 - \tilde{\mathbf{\Sigma}}_i^{yx}\right\|_F^2 - c$$

$$= \frac{1}{2}\left(\left\|\tilde{\mathbf{\Sigma}}_k^{yx} - \tilde{\mathbf{\Sigma}}_i^{yx}\right\|_F^2 - \left\|\mathbf{W}_2\mathbf{W}_1 - \tilde{\mathbf{\Sigma}}_i^{yx}\right\|_F^2\right).$$

It is important to note that the amount of forgetting is a function of the weight trajectories. Therefore, we have analytical solutions for trajectories of forgetting as well.

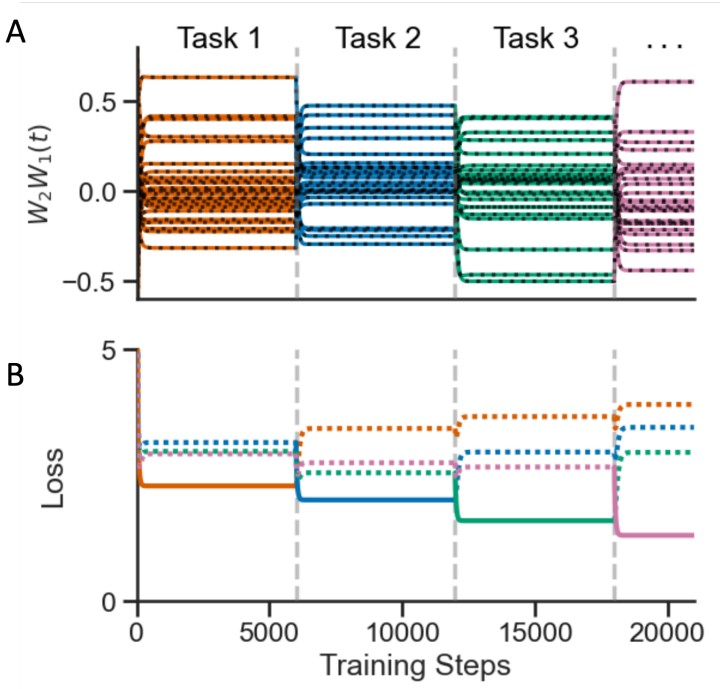

Figure 5: Continual learning. **A** Top: Network training from small zero-balanced weights across a sequence of tasks (colored lines represent simulations, and black dotted lines represent analytical results). Bottom: Evaluation loss for the tasks in the sequence (dotted lines) while training on the current task (solid lines). As the network optimizes its function on the current task, the loss on previously learned tasks increases.

Figure. 5 panel was generated by training a linear network with $N_i = 5$, $N_h = 10$, $N_o = 6$ subsequently on four different random regression tasks with $N = 25$. The learning rate was $\eta = 0.05$ and the initial weights were small ($\sigma = 0.0001$).

### E.2 Appendix: Reversal Learning

As first introduced in Braun et al. (2022), in the following discussion, we assume that the input and output dimensions are equal. We denote the $i$-th columns of the left and right singular vectors as $\mathbf{u}_i$, $\tilde{\mathbf{u}}_i$, and $\mathbf{v}_i$, $\tilde{\mathbf{v}}_i$, respectively.

Reversal learning occurs when both the task and the initial network function share the same left and right singular vectors, i.e., $\mathbf{U} = \tilde{\mathbf{U}}$ and $\mathbf{V} = \hat{\mathbf{V}}$, with the exception of one or more columns of the left singular vectors, where the direction is reversed: $-\mathbf{u}_i = \tilde{\mathbf{u}}_i$.

It is important to note that if a reversal occurs in the right singular vectors, such that $-\mathbf{v}_i = \tilde{\mathbf{v}}_i$, this can be equivalently represented as a reversal in the left singular vectors, as the signs of the right and left singular vectors are interchangeable.

In the reversal learning setting, both $\boldsymbol{B} = \boldsymbol{S}_2 \tilde{U}^T \tilde{U}(\tilde{G} + \tilde{H}\tilde{G}) + \boldsymbol{S}_1 V^T \tilde{V}(\tilde{G} - \tilde{H}\tilde{G})$ and $\boldsymbol{C} = \boldsymbol{S}_2 \tilde{U}^T \tilde{U}(\tilde{G} - \tilde{H}\tilde{G}) - \boldsymbol{S}_1 V^T \tilde{V}(\tilde{G} + \tilde{H}\tilde{G})$ are diagonal matrices.

In the case where lambda is zero, the same argument given in Braun et al. (2022) follows, the diagonal entries of $\mathbf{C}$ are zero if the singular vectors are aligned and non zero if they are reversed. Similarly, diagonal entries of $\mathbf{B}$ are non-zero if the singular vectors are aligned and zero if they are reversed. Therefore, in the case of reversal learning, $\mathbf{B}$ is a diagonal matrix with 0 values and thus is not invertible. As a consequence, the learning dynamics cannot be described by Equation 37. However, as $\mathbf{B}$ and $\mathbf{C}$ are diagonal matrices, the learning dynamics simplify. Let $\mathbf{b}_i$, $\mathbf{c}_i$, $\mathbf{s}_i$ and $\tilde{\mathbf{s}}_i$ denote the $i$-th diagonal entry of $\mathbf{B}$, $\mathbf{C}$, $\mathbf{S}$ and $\tilde{\mathbf{S}}$ respectively, then the network dynamics can be rewritten as

$$
\begin{aligned}
\mathbf{W}_2\mathbf{W}_1(t) = \frac{1}{2}\tilde{\mathbf{U}} & \left[ (\tilde{G} + \tilde{H}\tilde{G})e^{\tilde{S}_\lambda \frac{t}{\tau}}\mathbf{B}^T + (\tilde{G} - \tilde{H}\tilde{G})e^{-\tilde{S}_\lambda \frac{t}{\tau}}\mathbf{C}^T \right) \\
& \left[ \boldsymbol{S}_\lambda^{-1} + \frac{1}{4}\mathbf{B}\left(e^{2\tilde{S}_\lambda \frac{t}{\tau}} - \mathbf{I}\right)\tilde{\mathbf{S}}_\lambda^{-1}\mathbf{B}^T - \frac{1}{4}\mathbf{C}\left(e^{-2\tilde{S}_\lambda \frac{t}{\tau}} - \mathbf{I}\right)\tilde{\mathbf{S}}_\lambda^{-1}\mathbf{C}^T \right]^{-1} \\
& \frac{1}{2}\left( (\tilde{G} - \tilde{H}\tilde{G})e^{\tilde{S}_\lambda \frac{t}{\tau}}\mathbf{B} - (\tilde{G} + \tilde{H}\tilde{G})e^{-\tilde{S}_\lambda \frac{t}{\tau}}\mathbf{C} \right)\tilde{\mathbf{V}}^T
\end{aligned} \tag{157}
$$

$$
= \sum_{i=1}^{N_i} \frac{\mathbf{b}_i^2 e^{2\tilde{\mathbf{s}}_{\lambda i}\frac{t}{\tau}} - \mathbf{c}_i^2 e^{-2\tilde{\mathbf{s}}_{\lambda i}\frac{t}{\tau}}}{4\mathbf{s}_{\lambda i}^{-1} + \mathbf{b}_i^2 e^{2\tilde{\mathbf{s}}_{\lambda i}\frac{t}{\tau}}\tilde{\mathbf{s}}_{\lambda i}^{-1} - \mathbf{b}_i^2\tilde{\mathbf{s}}_{\lambda i}^{-1} - \mathbf{c}_i^2 e^{-2\tilde{\mathbf{s}}_{\lambda i}\frac{t}{\tau}}\tilde{\mathbf{s}}_{\lambda i}^{-1} + \mathbf{c}_i^2\tilde{\mathbf{s}}_{\lambda i}^{-1}} \tilde{\mathbf{u}}_i\tilde{\mathbf{v}}_i^T \tag{158}
$$

$$
= \sum_{i=1}^{N_i} \frac{\mathbf{s}_{\lambda i}\mathbf{b}_i^2\tilde{\mathbf{s}}_{\lambda i} - \mathbf{s}_{\lambda i}\mathbf{c}_i^2\tilde{\mathbf{s}}_i e^{-4\tilde{\mathbf{s}}_i\frac{t}{\tau}}}{4\tilde{\mathbf{s}}_{\lambda i}e^{-2\tilde{\mathbf{s}}_i\frac{t}{\tau}} + \mathbf{s}_{\lambda i}\mathbf{b}_i^2\left(1 - e^{-2\tilde{\mathbf{s}}_{\lambda i}\frac{t}{\tau}}\right) + \mathbf{s}_{\lambda i}\mathbf{c}_i^2\left(e^{-2\tilde{\mathbf{s}}_{\lambda i}\frac{t}{\tau}} - e^{-4\tilde{\mathbf{s}}_{\lambda i}\frac{t}{\tau}}\right)} \tilde{\mathbf{u}}_i\tilde{\mathbf{v}}_i^T
$$

$$\tag{159}$$

It follows, that in the reversal learning case, i.e. $\mathbf{b} = 0$, for each reversed singular vector, the dynamics vanish to zero

$$
\lim_{t\to\infty} \frac{-\mathbf{s}_{\lambda i}\mathbf{c}_i^2\tilde{\mathbf{s}}_i e^{-4\tilde{\mathbf{s}}_{\lambda i}\frac{t}{\tau}}}{4\tilde{\mathbf{s}}_{\lambda,i}e^{-2\tilde{\mathbf{s}}_{\lambda i}\frac{t}{\tau}} + \mathbf{s}_i\mathbf{c}_i^2\left(e^{-2\tilde{\mathbf{s}}_{\lambda i}\frac{t}{\tau}} - e^{-4\tilde{\mathbf{s}}_{\lambda i}\frac{t}{\tau}}\right)} \tilde{\mathbf{u}}_i\tilde{\mathbf{v}}_i^T = 0. \tag{160}
$$

Analytically, the learning dynamics are initialized on and remain along the separatrix of a saddle point until the corresponding singular value of the network function decreases to zero and stays there, indicating convergence to the saddle point. In numerical simulations, however, the learning dynamics can escape the saddle points due to the imprecision of floating-point arithmetic. Despite this, numerical optimization still experiences significant delays, as escaping the saddle point is time-consuming Lee et al. (2022). In contrast, when the singular vectors are aligned ($\mathbf{c} = 0$), the equation governing temporal dynamics, as described in Saxe et al. (2014), is recovered. Under these conditions, training succeeds, with the singular value of the network function converging to its target value.

$$
\lim_{t\to\infty} \sum_{i=1}^{N_i} \frac{\mathbf{s}_{\lambda i}\mathbf{b}_i^2\tilde{\mathbf{s}}_{\lambda i}}{4\tilde{\mathbf{s}}_{\lambda i}e^{-2\tilde{\mathbf{s}}_{\lambda i}\frac{t}{\tau}} + \mathbf{s}_{\lambda i}\mathbf{b}_i^2\left(1 - e^{-2\tilde{\mathbf{s}}_{\lambda i}\frac{t}{\tau}}\right)} \tilde{\mathbf{u}}_i\tilde{\mathbf{v}}_i^T = \frac{\mathbf{s}_{\lambda i}\mathbf{b}_i^2\tilde{\mathbf{s}}_{\lambda i}}{\mathbf{s}_{\lambda i}\mathbf{b}_i^2}\tilde{\mathbf{u}}_i\tilde{\mathbf{v}}_i^T \tag{161}
$$

$$
= \tilde{\mathbf{s}}_{\lambda i}\tilde{\mathbf{u}}_i\tilde{\mathbf{v}}_i^T. \tag{162}
$$

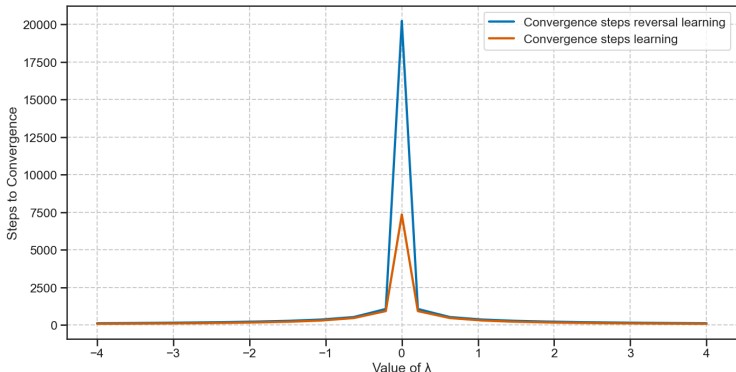

Figure 6: Plot showing the steps to convergence for two tasks: (1) the reversal learning task and (2) a randomly sampled continual learning task across a range of $\lambda$ values. The reversal learning task exhibits catastrophic slowing at $\lambda = 0$.

In summary, in the case of aligned singular vectors, the learning dynamics can be described by the convergence of singular values. However in the case of reversal learning, analytically, training does not succeed. In simulations, the learning dynamics escape the saddle point due to numerical imprecision, but the learning dynamics are catastrophically slowed in the vicinity of the saddle point as shown in figure 6 .

In the case where $\lambda$ is non-zero, the diagonal of $\mathbf{C}$ are also non-zero; this is true regardless of whether they are reversed or aligned. Similarly, the diagonal entries of $\mathbf{B}$ remain non-zero whether the singular vectors are aligned or reversed. Therefore, in the case of reversal learning, $\mathbf{B}$ is a diagonal matrix with elements that are zero. In figure 6

### E.3    Appendix: Generalization and structured learning

We study how the representations learned for different $\lambda$ initializations impact generalization of properties of the data. To do this, we consider the case where a new feature is associated to a learned item in a dataset and how this new feature may then be related to other items based on prior knowledge. In particular, we first train each network (for different values of $-10 \leq \lambda \leq 10$) on the hierarchical semantic learning task in Section 3 and then add a new feature (e.g., 'eats worms') to a single item (e.g., the goldfish) (Fig. 7A), correspondingly increasing the output dimension to represent the novel feature. In order to learn the new feature without affecting prior knowledge, we append a randomly initialized row to $\mathbf{W}_2$ and train it on the single item with the new feature, while keeping the rest of the network frozen. Thus, we only change the weights from the hidden layer to the new feature which may produce different behavior depending on how the hidden layer representations vary based on $\lambda$. After training on the new feature-item association, we query the network with the rest of the data to observe how the new feature is associated with the other items. We find that as $\lambda$ increases positively, the network better transfers the hierarchy such that it projects the feature onto items based on their distance to the trained item (Fig. 7B,C). For example, after learning that a goldfish eats worms, the network can extrapolate the hierarchy to infer that another fish, or birds, may also eat worms; instead, plants are not likely to eat worms. Alternatively, as $\lambda$ becomes more negative, the network ceases to infer any hierarchical structure and only learns to map the new feature to the single item trained on. In this case, after learning that a goldfish eats worms, the network does not infer that other fish, birds, or plants may also eat worms.

Interestingly, this setting highlights how asymmetries in the representations yielded by different $\lambda$ can actually benefit transfer and generalization. This can be shown by observing that the learning of a new feature association only depends on the first layer $\mathbf{W}_1$. Let $\hat{\boldsymbol{y}}_f$ denote the vector of the representation of the new feature $f$ across items $i$ in the dataset. Additionally, let $\boldsymbol{w}_2^{(f)T}$ be the new row of weights appended to $\mathbf{W}_2$ which map the hidden layer to the new feature. Following Saxe et al. (2019b), if $\boldsymbol{w}_2^{(f)T}$ is initialized with small random weights and trained on item $\tilde{\boldsymbol{H}}_i$, it will

converge to

$$\boldsymbol{w}_2^{(f)T} = \tilde{\boldsymbol{H}}_i^T \mathbf{W}_1^T / \|\mathbf{W}_1 \tilde{\boldsymbol{H}}_i\|_2^2 \tag{163}$$

$$\hat{\boldsymbol{y}}_f = (\tilde{\boldsymbol{H}}_i^T \mathbf{W}_1^T \mathbf{W}_1 \tilde{\boldsymbol{H}}) / \|\mathbf{W}_1 \tilde{\boldsymbol{H}}_i\|_2^2 \tag{164}$$

From this we can see that differences in the representations of the new feature across items $\hat{\boldsymbol{y}}_f$ across $\lambda$ are only influenced by $\mathbf{W}_1$.

In the case of the rich learning regime where $\lambda = 0$, the semantic relationship between features and items is distributed across both layers. Instead, when $\lambda > 0$, the second layer $\mathbf{W}_2$ exhibits *lazy* learning, yielding an output representation $\mathbf{W}_2 \mathbf{W}_2^T$ of a weighted identity matrix. However, the first layer $\mathbf{W}_1$ still learns a *rich* representation of the hierarchy, albeit at a smaller scaling. Furthermore, rather than distributing this learning across both layers, in the $\lambda > 0$ case, all learning of the hierarchy occurs in the first layer, allowing it to more readily transfer this structure to the learning of a new feature (which only depends on the first layer). Thus, in this case, the 'shallowing' of the network into the first layer is actually beneficial. Finally, we can also observe the opposite case when $\lambda < 0$. Here, *rich* learning happens in the second layer, while the first layer is *lazy* and learns to represent a weighted identity matrix. As such, these networks do not learn to transfer the hierarchy of different items to the new feature.

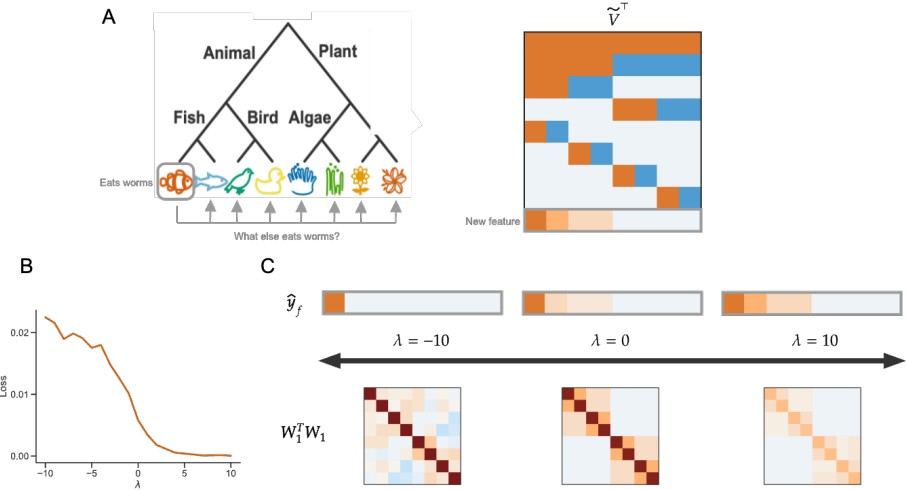

Figure 7: Transfer learning for different $\lambda$. **A** A new feature (such as 'eats worms') is introduced to the dataset after training on the hierarchical semantic learning task (Section 3). A randomly initialized row is added to $\mathbf{W}_2$ and trained on a single item with the new feature (for example, the goldfish), with the rest of the network frozen. The network is then tested on the transfer of the new feature to other items, such that items closer to the goldfish in the hierarchy are more likely to have the same feature. **B** The generalization loss on the untrained items with the new feature decreases as $\lambda$ increases. **C** As $\lambda$ increases positively, networks better transfer the hierarchical structure of the data to the representation of the new feature.

## F   Implementation and Simulations

The details of the simulation studies are described as follows. Specifically, $N_i$, $N_h$, and $N_o$ represent the dimensions of the input, hidden layer, and output (target), respectively. The total number of training samples is denoted by $N$, and the learning rate is defined as $\eta = \frac{1}{\tau}$.

### F.1   Lambda-balanced weight initialization

In practice, to initialize the network with lambda-balanced weights, we use Algorithm F.1. In this algorithm, $\alpha$ serves as a scaling factor that controls the variance of the weights, allowing for adjustments between smaller and larger weight initializations.

**Algorithm 1** Get $\lambda$-*balanced*

---

1: **function** GET_LAMBDA_BALANCED($\lambda$, $in\_dim$, $hidden\_dim$, $out\_dim$, $\sigma = 1$)
2:     **if** $out\_dim > in\_dim$ and $\lambda < 0$ **then**
3:         **raise** Exception('Lambda must be positive if out_dim ¿ in_dim')
4:     **end if**
5:     **if** $in\_dim > out\_dim$ and $\lambda > 0$ **then**
6:         **raise** Exception('Lambda must be positive if in_dim ¿ out_dim')
7:     **end if**
8:     **if** $hidden\_dim < \min(in\_dim, out\_dim)$ **then**
9:         **raise** Exception('Network cannot be bottlenecked')
10:     **end if**
11:     **if** $hidden\_dim > \max(in\_dim, out\_dim)$ and $\lambda \neq 0$ **then**
12:         **raise** Exception('hidden_dim cannot be the largest dimension if lambda is not 0')
13:     **end if**
14:     $W_1 \leftarrow \sigma \cdot$ random normal matrix($hidden\_dim, in\_dim$)
15:     $W_2 \leftarrow \sigma \cdot$ random normal matrix($out\_dim, hidden\_dim$)
16:     $[U, S, Vt] \leftarrow \text{SVD}(W_2 \cdot W_1)$
17:     $R \leftarrow$ random orthonormal matrix($hidden\_dim$)
18:     $S2_{equal\_dim} \leftarrow \sqrt{\left(\sqrt{\lambda^2 + 4 \cdot S^2} + \lambda\right)/2}$
19:     $S1_{equal\_dim} \leftarrow \sqrt{\left(\sqrt{\lambda^2 + 4 \cdot S^2} - \lambda\right)/2}$
20:     **if** $out\_dim > in\_dim$ **then**
21:         $S2 \leftarrow \begin{bmatrix} S2_{equal\_dim} & 0 \\ 0 & 0_{hidden\_dim - in\_dim} \end{bmatrix}$
22:         $S1 \leftarrow \begin{bmatrix} S1_{equal\_dim} \\ 0 \end{bmatrix}$
23:     **else if** $in\_dim > out\_dim$ **then**
24:         $S1 \leftarrow \begin{bmatrix} S1_{equal\_dim} & 0 \\ 0 & 0_{hidden\_dim - out\_dim} \end{bmatrix}$
25:         $S2 \leftarrow \begin{bmatrix} S2_{equal\_dim} & 0 \end{bmatrix}$
26:     **end if**
27:     $init\_W_2 \leftarrow U \cdot S2 \cdot R^T$
28:     $init\_W_1 \leftarrow R \cdot S1 \cdot Vt$
29:     **return** $(init\_W_1, init\_W_2)$
30: **end function**

---

### F.2 Tasks

In the following, we describe the different tasks that are used throughout the simulation studies.

### F.2.1 Random regression task

In the random regression task, the inputs $\mathbf{X} \in \mathbb{R}^{N_i \times N}$ are generated from a standard normal distribution, $\mathbf{X} \sim \mathcal{N}(\mu = 0, \sigma = 1)$. The input data $\mathbf{X}$ is then whitened to satisfy $\frac{1}{N}\mathbf{X}\mathbf{X}^T = \mathbf{I}$. The target values $\mathbf{Y} \in \mathbb{R}^{N_o \times N}$ are independently sampled from a normal distribution with variance scaled according to the number of output nodes, $\mathbf{Y} \sim \mathcal{N}(\mu = 0, \alpha = \frac{1}{\sqrt{N_o}})$. Consequently, the network inputs and target values are uncorrelated Gaussian noise, implying that a linear solution may not always exist.

### F.2.2 Semantic hierarchy

We use the same task as in Braun et al. (2022) and modify it to match the theoretical dynamics. The modification ensures that the inputs are whitened. In the semantic hierarchy task, input items are represented as one-hot vectors, i.e., $\mathbf{X} = \frac{\mathbf{I}}{8}$. The corresponding target vectors, $\mathbf{y}_i$, encode the item's position within the hierarchical tree. Specifically, a value of $1$ indicates that the item is a left child of a node, $-1$ denotes a right child, and $0$ indicates that the item is not a child of that node. For example, consider the blue fish: it is a blue fish, a left child of the root node, a left child of the

animal node, not part of the plant branch, a right child of the fish node, and not part of the bird, algae, or flower branches, resulting in the label $[1, 1, 1, 0, -1, 0, 0, 0]$. The labels for all objects in the semantic tree, as shown in Figure 2 A, are given by:

$$
\mathbf{Y} = 8 * \begin{bmatrix}
1 & 1 & 1 & 1 & 1 & 1 & 1 & 1 \\
1 & 1 & 1 & 1 & -1 & -1 & -1 & -1 \\
1 & 1 & -1 & -1 & 0 & 0 & 0 & 0 \\
0 & 0 & 0 & 0 & 1 & 1 & -1 & -1 \\
1 & -1 & 0 & 0 & 0 & 0 & 0 & 0 \\
0 & 0 & 1 & -1 & 0 & 0 & 0 & 0 \\
0 & 0 & 0 & 0 & 1 & -1 & 0 & 0 \\
0 & 0 & 0 & 0 & 0 & 0 & 1 & -1
\end{bmatrix}. \tag{165}
$$

The singular value decomposition (SVD) of the corresponding correlation matrix, $\tilde{\boldsymbol{\Sigma}}^{yx}$, is not unique due to identical singular values: the first two, the third and fourth, and the last four values are the same. To align the numerical and analytical solutions, this permutation invariance is addressed by adding a small perturbation to each column $\mathbf{y}_i$, for $i \in 1, ..., N$, of the labels:

$$
\mathbf{y}_i = \mathbf{y}_i \cdot \left(1 + \frac{0.1}{i}\right), \tag{166}
$$

resulting in singular values that are nearly, but not exactly, identical.

### F.3 Figure 1

Panels B illustrates three simulations conducted on the same task with varying initial $\lambda$-balanced weights respectively $\lambda = -2$, $\lambda = 0$, $\lambda = 2$. The regression task parameters were set with ($\sigma = \sqrt{10}$). The network architecture consisted of $N_i = 3$, $N_h = 2$, $N_o = 2$, with a learning rate of $\eta = 0.0002$. The batch size is $N = 10$. The zero-balanced weights are initialized with variance $\sigma = 0.00001$. The lambda-balanced network are initialized with $sigmaxy = \sqrt{1}$ of a random regression task with same architecture.

On Panel C , we plot the ballancedness $\mathbf{W}_2(0)^T \mathbf{W}_2(0) - \mathbf{W}_1(0) \mathbf{W}_1(0)^T$ for a two layer network initialised with Lecun initialization with dimension $N_i = 40$ ,$N_h = 120$ ,$N_o = 250$

### F.4 Figure 2

Panel A, B, C illustrates three simulations conducted on the same task with varying initial $\lambda$-balanced weights respectively $\lambda = -2$, $\lambda = 0$, $\lambda = 2$ according to the initialization scheme described in F.7. The regression task parameters were set with ($\sigma = \sqrt{10}$). The network architecture consisted of $N_i = 3$, $N_h = 2$, $N_o = 2$ with a learning rate of $\eta = 0.0002$. The batch size is $N = 10$. The zero-balanced weights are initialized with variance $\sigma = 0.00001$. The lambda-balanced network are initialized with $sigmaxy = \sqrt{1}$ of a random regression task with same architecture.

### F.5 Figure 3

Panel A, B, C illustrates three simulations conducted on the same task with varying initial $\lambda$-balanced weights respectively $\lambda = -2$, $\lambda = 0$, $\lambda = 2$ according to the initialization scheme described in F.7. The regression task parameters were set with ($\sigma = \sqrt{12}$). The network architecture consisted of $N_i = 3$, $N_h = 3$, $N_o = 3$ with a learning rate of $\eta = 0.0002$. The batch size is $N = 5$. The zero-balanced weights are initialized with variance $\sigma = 0.0009$. The lambda-balanced network are initialized with $sigmaxy = \sqrt{12}$ of a random regression task with same architecture.

### F.6 Figure 4

In Panel A presents a semantic learning task with the SVD of the input-output correlation matrix of the task. $U$ and $V$ represent the singular vectors, and $S$ contains the singular values. This

decomposition allows us to compute the respective RSMs as $USU^\top$ for the input and $VSV^\top$ for the output task. The rows and columns in the SVD and RSMs are ordered identically to the items in the hierarchical tree.

The results in Panel B display simulation outcomes, while Panel C presents theoretical input and output representation matrices at convergence for a network trained on the semantic task described in Braun et al. (2022); Saxe et al. (2013),. These matrices are generated using varying initial $\lambda$-balanced weights set at $\lambda = -2$, $\lambda = 0$, and $\lambda = 2$, following the initialization scheme outlined in F.7. The network architecture includes $N_i = 8$, $N_h = 8$, and $N_o = 8$ with a learning rate of $\eta = 0.001$ and a batch size of $N = 8$. Zero-balanced weights are initialized with a variance of $\sigma = 0.00001$, while $\lambda$-balanced networks are initialized with $\sigma_{xy} = \sqrt{1}$ based on a random regression task with the same architecture.

Panel D illustrates results from running the same task and network configuration but initialized with randomly large weights having a variance of $\sigma = 1$.

In panel E, we trained a two-layer linear network with $N_i = N_h = N_o = 4$ on a random regression task for $\lambda \in [-5, -4, -3, -2, -1, 0, 1, 2, 3, 4, 5]$ to convergence. Subsequently, we added Gaussian noise with $\mu = 0, \sigma \in [0, 0.5, 1]$ to the inputs (top panel) or synaptic weights (bottom panel) and calculated the expected mean squared error.

### F.7   Figure 5

Panel A illustrates schematic representations of the network architectures considered: from left to right, a funnel network ($N_i = 4$, $N_h = 2$, $N_o = 2$), a square network ($N_i = 4$, $N_h = 4$, $N_o = 4$), and an inverted-funnel network ($N_i = 2$, $N_h = 2$, $N_o = 4$).

Panel B shows the Neural Tangent Kernel (NTK) distance from initialization, as defined in Fort et al. (2020), across the three architectures shown schematically. The kernel distance is calculated as:

$$S(t) = 1 - \frac{\langle K_0, K_t \rangle}{\|K_0\|_F \|K_t\|_F}.$$

The simulations conducted on the same task with eleven varying initial $\lambda$-balanced weights in $[-9, 9]$. The regression task parameters were set with ($\sigma = \sqrt{3}$). The task has batch size $N = 10$. The network has with a learning rate of $\eta = 0.01$. The lambda-balanced network are initialized with $\sigma xy = \sqrt{1}$ of a random regression task.

Panel C shows the Neural Tangent Kernel (NTK) distance from initialization for the funnel architectures shown schematically with dimensions $N_i = 3$, $N_h = 2$, and $N_o = 2$. The simulations conducted on the same task with twenty one varying initial $\lambda$-balanced weights in $[-9, 9]$. The regression task parameters were set with ($\sigma = \sqrt{3}$). The task has batch size $N = 30$. The network has with a learning rate of $\eta = 0.002$. The lambda-balanced network are initialized with $\sigma xy = \sqrt{1}$ of a random regression task.

