# OpenReview forum: "From Lazy to Rich: Exact Learning Dynamics in Deep Linear Networks"
_NeurIPS.cc/2024/Workshop/UniReps — UniReps_

### Official Review · Reviewer_nwhL · 2024-10-04
**meaningful extension for learning dynamics..**

**Rating:** 9
**Confidence:** 4

**Review:**

Summary: This work derived exact solutions for gradient flow dynamics of a few important statistics in deep linear networks under λ-balanced initializations extending the results from Braun & Dominé 2022. The authors examined learning dynamics different from lazy to rich regimes, showing how this transition is influenced by  λ, architecture, noise, and extended the discussion of the behaviors of the solution in different learning scenarios.

Minor points:
- Repeated reference entries;
- Some lemma and theorem links are broken.

Major points:
- It is definitely looks like rigorously body of work, especially consider it is an abstract’s volume. I trust the authors’ math on finding the matrix Riccati equation for QQT, as it seems non trivial to introduce non-zero λ.
- The funnel v.s. squared network result is interesting, as this kind of breaks the symmetry in the squared network and shows the possibility of the delayed transition to rich. The magnitude of how far the NTK moved from initialization is interesting/surprising, compared to the square network.
- I wonder how the different types of layer imbalance, e.g. λ-balanced v.s. layer specific learning rate, relates to one another.

---

### Official Review · Reviewer_7hwJ · 2024-10-06
**Good early stage work, but hampered by low quality/effort writing**

**Rating:** 6
**Confidence:** 4

**Review:**

**Summary**
This paper builds on the work of (largely) Braun et al., 2022 which derive exact learning dynamics for deep linear networks (DLNs) with unequal input-output dimensions using the Riccati matrix equation. Prior work demonstrated that a zero-balanced DLN can exhibit rich regime solutions; this work examines a broader class of $\lambda$-balanced networks. They replicate previous findings while also introducing new dynamics specific to $\lambda$-balanced networks.

**Strengths**
- The extension to $\lambda$-balanced networks extends our understanding of DLN dynamics to a much broader class of initializations
- Evaluation is thorough and covers all settings discussed in prior works, highlighting how $\lambda$-balanced initializations provide new behavior
- The lazy to delayed rich learning highlighted in Fig. 3C may be useful for understanding grokking phenomenon discussed in other works
- Proofs appear to be mostly correct with no major technical errors

**Weaknesses**
- The technical contributions are clear, but the implications are not well articulated.
- While new behavior for $\lambda$-balanced networks is mentioned, it is not discussed in detail. This is acceptable for early-stage results in the Extended Abstract track, but a future version of this work would strongly benefit from spending more time investigating the delayed rich behavior and reversal learning behavior when $\lambda \neq 0$.
- Section 3, "The dynamics of the representations" introduces a new regime, _semi-structured lazy_ but does not explain this in detail or discuss its implications.
- The quality is quite poor with numerous writing and mathematical errors throughout (most are minor and do not affect the overall conclusions):
    - Lines 92-93 appears to have fragments of a sentence
    - Fig 2E appears to have the titles of the subplots switched (it does not match the caption nor the conclusions reached in the corresponding appendix section)
    - The reversal learning section introduces a "spectrum" of behaviors controlled by $\lambda$, and even suggests it relates to animal behavior, but it points the reader to Appendix E.2 for details. Appendix E.2 abruptly cuts off mid-sentence before explaining any of these details.
    - Proof of Theorem B.2 is missing transpose operators (likely typo?)
    - Lemma C.2 missing factor of $\tau$
    - Cross-references (especially in appendix) are wrong or missing

**Questions**
1. What is the _semi-structured lazy regime_? What are the implications for initializing networks?
2. Prior work discusses how two-layer DLN results extend to multiple layers. Would that be feasible for the type of results in Fig. 3?
3. What are the animal behaviors referred to in the reversal learning section? What are the implications of this work on animal behavior?

**Limitations**
Should have a discussion about what initializations are not covered by the $\lambda$-balanced assumption.

---

### Official Review · Reviewer_MeQz · 2024-10-07
**Solid and perhaps overqualified**

**Rating:** 8
**Confidence:** 3

**Review:**

The authors study the effect of relative initialization scale and layer dimension on the rich/lazy learning dichotomy. They refine some previous results showing that small initialization scale (i.e. variance of random initial weights) can promote rich learning, to study the effect of imbalances between the scale of different layers. They also see how this interacts with imbalances in the dimensionality across layers. All of this is done with a full dynamical picture that allows them to highlight a hybrid "delayed rich" regime.

While it is not really relevant for a future poster, I found this abstract hard to evaluate at times because some key information was missing or deep in the 30 page appendix. For example, one of the central variables, the "relative scale", is never defined. In the first Appendix, it is said to be the difference in weight magnitude between layers, but that leaves open the question of sign. If positive means that inputs have more weight, then it seems like, in figure 3, $\lambda$ is kind of compensating for the imbalanced dimensionality of layer layers (which would trivially produce higher magnitude depending on how weights are initialized) since funnel nets are rich for negative $\lambda$ and _vice versa_. Obviously that's not the case if $\lambda$ means the opposite. This is perhaps due to my fairly shallow familiarity with the related work, but it should still be defined. There are other rough spots presumably due to the trauma of copying from one overleaf to another, but those were mostly stylistic.

One question I had about the interesting "delayed rich" regime, is whether networks in this regime have already saturated in their loss curves. If so, it seems potentially related to the "grokking" phenomenon seen in non-linear networks (to the extent that rich representations are more robust, as briefly mentioned in the paper). I would have liked more exploration of this regime in general, seeing as it got the dignity of an italicised name. It was not clear, for example, which architecture is shown in Fig. 3c (which is otherwise a figure mostly about architecture).

My overall impression is that it is solid and interesting, but a lot of crucial detail was left out of its current presentation. In fact, this seems like fairly mature work that could have benefited from being in the proceedings track.

---

### Official Review · Reviewer_UTVa · 2024-10-07
**Exploring λ-balanced Initialization, Learning Dynamics, Generalization Impact, RSM Evolution, Empirical Validation, and Transfer Learning in Deep Linear Networks**

**Rating:** 6
**Confidence:** 5

**Review:**

Comment 1: In Section 2, the author explains the supervised learning setup and the use of deep linear networks for modeling. While the explanation of weight matrices and input-output mappings is clear, there should be more detail on how the λ-balanced initialization specifically affects the convergence behavior.

Comment 2: The author introduces the QQT matrix to track key statistics of the learning process. This approach is insightful, but there should be a clearer explanation of how the choice of QQT over other methods improves the tracking of learning dynamics.

Comment 3: In Section 3, the author explores the dynamics of singular values under different initializations. While the transition from sigmoidal to exponential dynamics is well-described, there is limited discussion on how these transitions impact generalization.

Comment 4: The paper introduces representational similarity matrices (RSMs) to study the internal representations of weights W1 and W2. However, the theoretical explanation could be strengthened by adding empirical results showing how RSMs evolve during training. P

Comment 5: The results in Section 4 discuss continual and reversal learning, but the paper lacks empirical validation of these findings. The author should present detailed simulation results that show how continual learning dynamics differ across values of λ, particularly when catastrophic forgetting is more likely to occur.

Comment 6: In the discussion of transfer learning, the author makes a compelling case for how task-specific structure can persist across tasks depending on the λ value. However, more quantitative results should be provided to support the claim that smaller λ values lead to better transfer performance.

---

### Decision · Program_Chairs · 2024-10-10

**Decision:**

Accept

**Comment:**

In light of the positive reviewers' feedback and relevancy of the submission, we are pleased to accept this paper for presentation at UniReps 2024. We kindly ask the authors to incorporate the reviewers' suggestions and feedback in the final camera-ready version of the manuscript.